# The Diffusion Duality

**Subham Sekhar Sahoo** [1] **Justin Deschenaux** [2] **Aaron Gokaslan** [1] **Guanghan Wang** [1] **Justin Chiu** [3]
**Volodymyr Kuleshov** [1]

## Abstract

Uniform-state discrete diffusion models hold the promise of fast text generation due to their inherent ability to self-correct. However, they are typically outperformed by autoregressive models and masked diffusion models. In this work, we narrow this performance gap by leveraging a key insight: Uniform-state diffusion processes naturally emerge from an underlying Gaussian diffusion. Our method, Duo, transfers powerful techniques from Gaussian diffusion to improve both training and sampling. First, we introduce a curriculum learning strategy guided by the Gaussian process, **doubling training speed** by reducing variance. Models trained with curriculum learning surpass autoregressive models in zero-shot perplexity on 3 of 7 benchmarks. Second, we present Discrete Consistency Distillation, which adapts consistency distillation from the continuous to the discrete setting. This algorithm **unlocks few-step generation in diffusion language models** by accelerating sampling by two orders of magnitude. We provide the code and model checkpoints on the project page:

https://s-sahoo.com/duo

## 1. Introduction

An eternal theme in mathematics is that discreteness emerges from underlying continuity. From quantum mechanics, where the quantized energy states of electrons arise as solutions to continuous wave equations, to the binary logic of digital circuits, fundamentally driven by smooth analog currents, discreteness has repeatedly and naturally emerged from an underlying continuum. Our work contin-

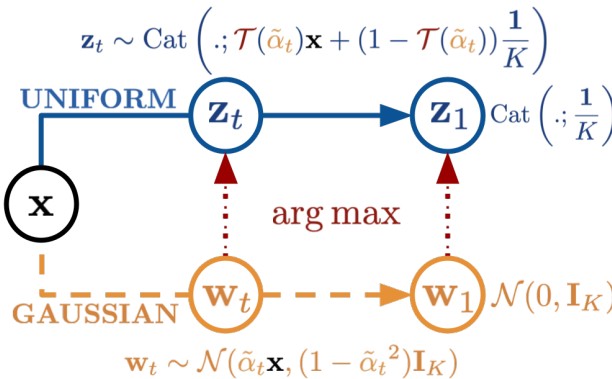

Figure 1: An illustration of Uniform-state discrete diffusion (top) and the underlying Gaussian diffusion (bottom). While both are separate Markov processes, applying $\arg\max$ maps Gaussian latents $\mathbf{w}_t \in \mathbb{R}^n$ to discrete latents $\mathbf{z}_t \in \mathcal{V}$, transforming their marginals from $\tilde{q}_t(.|\mathbf{x}; \tilde{\alpha}_t)$ (6) to $q_t(.|\mathbf{x}; \mathcal{T}(\tilde{\alpha}_t))$ (1) and adjusting diffusion parameters from $\tilde{\alpha}_t$ to $\alpha_t = \mathcal{T}(\tilde{\alpha}_t)$ (10). Notably, the ELBO for Uniform-state diffusion induces a tighter bound on the likelihood than Gaussian diffusion, as established in Theorem 3.1.

ues this tradition by demonstrating that a discrete diffusion process is, in fact, an emergent phenomenon of an underlying continuous Gaussian diffusion process. This perspective enables the design of faster training and sampling algorithms for discrete diffusion models.

Diffusion models (Sohl-Dickstein et al., 2015) are powerful generative models inspired by physics. Gaussian diffusion models excel at synthesizing realistic and high-quality continuous-valued data such as images (Ho et al., 2020; Rombach et al., 2022), audio (Kong et al., 2021; Liu et al., 2023b), and videos (Ho et al., 2022; Wu et al., 2023; Esser et al., 2023; Blattmann et al., 2023). Gaussian diffusion is well studied–the success of these models is rooted in techniques such as efficient parameterizations of the denoising model, which improve upon the standard mean-parameterization (Ho et al., 2020; Salimans & Ho, 2022; Zheng et al., 2023), faster training techniques (Kingma et al., 2021), efficient samplers (Karras et al., 2022), and distillation schemes that enable single-step generation (Song et al., 2023; Song & Dhariwal, 2023; Yin et al., 2024).

[1]Computer and Information Science, Cornell Tech, NY, USA. [2]School of Computer and Communication Sciences, EPFL Lausanne, Switzerland [3]Cohere, NY, USA. Correspondence to: Subham Sekhar Sahoo <ssahoo@cs.cornell.edu>.

*Proceedings of the $42^{nd}$ International Conference on Machine Learning*, Vancouver, Canada. PMLR 267, 2025. Copyright 2025 by the author(s).

While Gaussian diffusion is well-studied, it underperforms discrete diffusion models on tasks involving discrete data–such as text (Sahoo et al., 2024b), graphs (Liu et al., 2023a), and molecules (Lee et al., 2025). However, from the perspective of Gaussian diffusion, the design space for discrete diffusion models remains primitive: mean parameterization for the denoising model (Sahoo et al., 2024a; Schiff et al., 2025) and slow ancestral sampling (Austin et al., 2021) are still the dominant approaches. Recent work on distilling Masked Discrete Diffusion Models (MDMs) improves sampling speed (Deschenaux & Gulcehre, 2024), but performance degrades severely in the few-step regime. Unlike Gaussian diffusion models with Probability Flow ODEs (Song et al., 2020), MDMs lack an "implicit" property: a deterministic mapping from noise to data. This property is vital for few-step distillation methods (Song et al., 2023; Frans et al., 2024), but MDMs forgo it due to their deterministic prior, which requires stochasticity during sampling.

Our objective is (1) to design a framework for discrete diffusion that enables the transfer of advanced training and inference techniques from Gaussian diffusion to discrete diffusion models. And (2) create language models that support few-step generation. To this end, we focus on Uniform-state Diffusion Models (USDMs) (Austin et al., 2021). In this paper, we discover a very remarkable property of USDMs–these emerge from Gaussian diffusion processes as illustrated in Fig. 1. We call this phenomenon–the Diffusion Duality which expands the design space of USDMs, making it possible to incorporate techniques developed for Gaussian diffusion. Notably, USDMs models allow token updates during reverse sampling unlike MDMs, naturally correcting earlier mistakes without requiring costly predictor-corrector (Zhao et al., 2024; Wang et al., 2025) steps—saving function evaluations (NFEs). However, these models have historically underperformed compared to MDMs (Austin et al., 2021; Lou et al., 2023), raising the key question: Can USDMs be made competitive with MDMs? And more importantly, can the implicit property of the underlying Gaussian diffusion be leveraged for fast, few-step generation?

We answer both questions with **Duo**, a rich framework of theoretical connections between USDMs and Gaussian diffusion. Duo enriches the design space of USDMs by incorporating Gaussian diffusion, which allows us to develop efficient training strategies that accelerate the training of USDMs, significantly reducing the performance gap between MDMs and AR models on standard language generation benchmarks. Notably, we **surpass AR models on 3 out of 7 zero-shot datasets** (Table 2). Furthermore, this duality allows us to adapt consistency distillation (Song et al., 2023) from Gaussian to discrete diffusion, reducing NFEs from 1024 to 8 with minimal effect on sample qual-

ity (Sec. 5.2). Importantly, in the low-NFE regime, Duo outperforms MDMs. Our main contributions are threefold: (1) We establish a theoretical connection between continuous and discrete diffusion, demonstrating that discrete diffusion arises from an underlying continuous Gaussian diffusion. This insight enables the transfer of techniques from the continuous domain to the discrete setting, opening up new possibilities. (2) Our framework **doubles the training speed** of USDMs by introducing a low-variance curriculum, and (3) **accelerates sampling by two orders of magnitude** by adapting efficient distillation methods from continuous diffusion models.

## 2. Background

**Notation** We represent scalar discrete random variables that can take $K$ values as 'one-hot' column vectors and define $\mathcal{V} = \{\mathbf{x} \in \{0, 1\}^K : \sum_{i=1}^K \mathbf{x}_i = 1\}$ as the set of all such vectors. Define $\text{Cat}(\cdot; \boldsymbol{\pi})$ as the categorical distribution over $K$ classes with probabilities given by $\boldsymbol{\pi} \in \Delta$, where $\Delta$ denotes the $K$-simplex. Additionally, let $\mathbf{1} = \{1\}^K$ and $\langle \mathbf{a}, \mathbf{b} \rangle$ and $\mathbf{a} \odot \mathbf{b}$ respectively denote the dot and Hadamard products between two vectors $\mathbf{a}$ and $\mathbf{b}$. We use $\mathbf{x}^{1:L} \in \mathcal{V}^L$ and $[\mathbf{x}^\ell]_{\ell=1}^L \in \mathcal{V}^L$ to denote sequences of length $L$.

### 2.1. Discrete Diffusion Models

Consider the clean data $\mathbf{x} \in \mathcal{V}$ drawn from the data distribution $q_{\text{data}}$. In the discrete diffusion framework (Sohl-Dickstein et al., 2015; Austin et al., 2021) the complex data distribution $q_{\text{data}}$ is mapped to a simple distribution through a sequence of Markov states. Sahoo et al. (2024a) propose a simplified variant—an interpolating noise framework—where the forward process $(q_t)_{t\in[0,1]}$ smoothly transitions from $q_{\text{data}}$ to a prior distribution $\text{Cat}(.; \boldsymbol{\pi})$, by introducing latent variables $\mathbf{z}_t \in \mathcal{V}$ whose marginals conditioned on $\mathbf{x}$ at time $t$ are given by:

$$q_t(.|\mathbf{x}; \alpha_t) = \text{Cat}(.; \alpha_t \mathbf{x} + (1 - \alpha_t)\boldsymbol{\pi}), \qquad (1)$$

where the diffusion parameter $\alpha_t \in [0, 1]$ is a strictly decreasing function in $t$, with $\alpha_{t=0} \approx 1$ and $\alpha_{t=1} \approx 0$. A discrete diffusion process is characterized by the time evolution of marginals follows a linear ordinary differential equation (Anderson, 2012):

$$\frac{\mathrm{d}}{\mathrm{d}t} q_t = Q_t q_t, \qquad (2)$$

where $Q_t \in \mathbb{R}^{K \times K}$ is the state transition matrix.

There are two main variants of interpolating noise frameworks: MDMs (Sahoo et al., 2024b), which use a masked token prior $\boldsymbol{\pi} = \mathbf{m}$ with $\mathbf{m} \in \mathcal{V}$ as a special mask token, and USDMs (Schiff et al., 2025), which uses a uniform prior over $\mathcal{V}$ ($\boldsymbol{\pi} = \mathbf{1}/K$). These frameworks differ in their

forward corruption dynamics. In MDMs, the clean data $\mathbf{x}$ either stays unchanged or transitions to the mask token $\mathbf{m}$, after which it remains masked for the rest of the process. In contrast, USDMs allow each token to either stay the same or transition uniformly to any other token in $\mathcal{V}$, with the transition probability determined by the diffusion timestep (see Fig. 5 for examples). These forward dynamics impact the reverse generation process: USDMs permit continual token updates, while MDMs fix tokens once unmasked. To mitigate this limitation, predictor-corrector methods have been proposed for MDMs (Campbell et al., 2022; Gat et al., 2024; Wang et al., 2025), but at the cost of added computation. In contrast, USDMs naturally exhibit a self-correcting property absent in AR and MDM approaches. As a result, our work focuses primarily on the USDM framework.

Schiff et al. (2025) show that for USDMs, the state transition matrix $Q_t$ is given by:

$$Q_t = \frac{\alpha'_t}{K\alpha_t}[\mathbf{1}\mathbf{1}^\top - K\mathbf{I}_K], \tag{3}$$

where $\alpha'_t$ is the time derivative of $\alpha_t$ and the true reverse posterior for a timestep $s < t$ is given as:

$$q_{s|t}(.\mid \mathbf{z}_t, \mathbf{x}) = \mathrm{Cat}\left(.; \frac{K\alpha_t \mathbf{z}_t \odot \mathbf{x} + (\alpha_{t|s} - \alpha_t)\mathbf{z}_t}{K\alpha_t\langle \mathbf{z}_t, \mathbf{x}\rangle + 1 - \alpha_t} \right.$$
$$\left. + \frac{(\alpha_s - \alpha_t)\mathbf{x} + (1 - \alpha_{t|s})(1 - \alpha_s)\mathbf{1}/K}{K\alpha_t\langle \mathbf{z}_t, \mathbf{x}\rangle + 1 - \alpha_t}\right) \tag{4}$$

where $\alpha_{t|s} = \alpha_t/\alpha_s$. Since $\mathbf{x}$ is unavailable during inference, we approximate it with a neural network $\mathbf{x}_\theta : \mathcal{V} \times [0,1] \to \Delta^K$ with parameters $\theta$. The resulting approximate reverse posterior is defined as $p_{s|t}^\theta(.|\mathbf{z}_t) = q_{s|t}(. \mid \mathbf{z}_t, \mathbf{x} = \mathbf{x}_\theta(\mathbf{z}_t, t))$. The denoising model is trained by minimizing the Negative Evidence Lower Bound (NELBO) (Schiff et al., 2025):

$$\mathrm{NELBO}\,(q, p_\theta; \mathbf{x})$$
$$= \mathbb{E}_{t\sim\mathcal{U}[0,1], q_t(\mathbf{z}_t|\mathbf{x};\alpha_t)} f(\mathbf{z}_t, \mathbf{x}_\theta(\mathbf{z}_t, t), \alpha_t; \mathbf{x}), \tag{5}$$

where $f$ is defined in (40). Sampling from this model begins with the prior $\mathbf{z}_{t=1} \sim \mathbf{1}/K$, and proceeds via ancestral denoising, i.e., by drawing $\mathbf{z}_s \sim p_{s|t}^\theta(.|\mathbf{z}_t)$ at each step.

## 2.2. Gaussian Diffusion Models

Gaussian diffusion maps a data distribution $q_{\mathrm{data}}$ to a simple prior distribution usually a Normal distribution $\mathcal{N}(0, \mathbf{I}_K)$, through a sequence of noisy latents $\mathbf{w}_t \sim \tilde{q}_t(.|\mathbf{x})$, whose marginal distribution is given by:

$$\tilde{q}_t(.|\mathbf{x}; \tilde{\alpha}_t) = \mathcal{N}(\tilde{\alpha}_t \mathbf{x}, (1 - \tilde{\alpha}_t{}^2)\mathbf{I}_K), \tag{6}$$

where the diffusion parameter $\tilde{\alpha}_t \in [0,1]$ is a monotonically decreasing function in $t$. For $\tilde{\alpha}_{t=0} = 1$ and $\tilde{\alpha}_{t=1} = 0$, the

NELBO for such a process is given as (Kingma et al., 2021):

$$\mathrm{NELBO}\,(\tilde{q}, p_\theta; \mathbf{x})$$
$$= -\mathbb{E}_{t\sim\mathcal{U}[0,1], \tilde{q}_t(\mathbf{w}_t|\mathbf{x}; \tilde{\alpha}_t)} \nu'(t) \|\mathbf{x} - \mathbf{x}_\theta(\mathbf{w}_t, t)\|_2^2 \tag{7}$$

where $\nu'(t)$ is the time derivative of the signal-to-noise ratio $\nu(t) = \tilde{\alpha}_t{}^2/(1 - \tilde{\alpha}_t{}^2)$ for the Gaussian diffusion process.

## 2.3. Consistency Distillation

Consistency models (Song et al., 2023; Song & Dhariwal, 2023) are a class of generative models that define a bijective mapping between the samples from the noise distribution $\mathcal{N}(0, \mathbf{I}_K)$ and the data distribution $q_{\mathrm{data}}$. They build on deterministic samplers for Gaussian diffusion (Song et al., 2020; 2021), specifically using the Probability-Flow ODE (PF-ODE). Given a pre-trained Gaussian diffusion model $\mathbf{x}_\theta$, which requires hundreds or thousands of sampling steps, Consistency Distillation is a popular technique to distil them down to fewer steps generation that enables much faster generation. The distillation begins with a teacher model $\mathbf{x}_{\theta^-}$, often the Exponentially Moving Average (EMA) of the student model $\mathbf{x}_\theta$ obtained during the course of training. A noisy sample $\mathbf{w}_t$ is drawn from the forward process $\tilde{q}_t(.|\mathbf{x})$ (6), and a less noisy sample $\mathbf{w}_s$ is obtained by numerically solving one PF-ODE step using $\mathbf{x}_{\theta^-}$. The student model is then trained to match the teacher's estimate of the clean sample minimizing the following loss:

$$\mathcal{L}(\theta, \theta^-) = \lambda(t) d\left(\mathbf{x}_\theta(\mathbf{w}_t, t), \mathbf{x}_{\theta^-}(\mathbf{w}_s, s)\right), \tag{8}$$

where $d : \mathbb{R}^K \times \mathbb{R}^K \to \mathbb{R}^+$ denotes the error between the teacher model's reconstruction $\mathbf{x}_{\theta^-}(\mathbf{w}_s, t)$ and the student model's reconstruction $\mathbf{x}_\theta(\mathbf{w}_t, t)$ of the original sample and $\lambda : [0,1] \to \mathbb{R}^+$ is a weighting function that scales the loss based on the diffusion time-step $t$.

## 3. The Diffusion Duality

Unlike discrete diffusion, Gaussian diffusion is replete with well-established empirical techniques, which have driven significant advances in both training (Ho et al., 2020; Salimans & Ho, 2022; Zheng et al., 2023) and sampling (Karras et al., 2022; Song et al., 2023; Song & Dhariwal, 2023; Yin et al., 2024). Our goal in this section is to establish a theoretical bridge between discrete-state diffusion and continuous-state diffusion, which will enable us to leverage tools from the later to improve the former.

We propose a simple method to map a Gaussian latent to the discrete space: the $\arg\max$ operator. But does this transformation of latents also transform a Gaussian diffusion process into a discrete one? A necessary and sufficient condition for this is that the marginal distribution of the discretized vector satisfies the characteristic ODE of a discrete diffusion process (2). We first derive a closed-form expression for

this marginal and show that $\arg\max$ maps the marginals of a Gaussian diffusion to those of a Uniform-state discrete diffusion, including a transformation of the diffusion parameters (9). Finally, we verify that this marginal evolves according to (11), establishing that $\arg\max$ transforms a Gaussian diffusion process into a Uniform-state discrete diffusion process.

We begin by defining a Gaussian diffusion process on $\mathbf{x} \in \mathcal{V}$ as per (6), with $\tilde{q}_{t=0} \approx q_{\text{data}}$ and $\tilde{q}_{t=1} = \mathcal{N}(0, \mathbf{I}_K)$. Let $\mathbf{w}_t \sim \tilde{q}_t(.|\mathbf{x}; \tilde{\alpha}_t)$ be an intermediate latent at time $t$. Next, define the operation $\arg\max : \mathbb{R}^K \to \mathcal{V}$ to map a continuous vector $\mathbf{w} \in \mathbb{R}^K$ to the one-hot vector corresponding to the index of its largest entry in $\mathbf{w}$, i.e., $\arg\max(\mathbf{w}) = \arg\max_{\mathbf{z}\in\mathcal{V}} \mathbf{z}^\top \mathbf{w}$.

**Discrete Marginals**  Let $\mathbf{z}_t = \arg\max(\mathbf{w}_t)$ and $P_t(.|\mathbf{x})$ denote its conditional pmf marginalized over $\mathbf{w}_t \sim \tilde{q}_t(.|\mathbf{x}; \tilde{\alpha}_t)$. In Suppl. A.1, we show:

$$\mathbf{z}_t \sim P_t\left(.|\mathbf{x}; \mathcal{T}(\tilde{\alpha}_t)\right) = \text{Cat}\left(.; \mathcal{T}(\tilde{\alpha}_t)\mathbf{x} + (1 - \mathcal{T}(\tilde{\alpha}_t))\frac{1}{K}\right), \quad (9)$$

where the function $\mathcal{T} : [0, 1] \to [0, 1]$ is the *Diffusion Transformation operator*, defined as:

$$\boxed{\mathcal{T}(\tilde{\alpha}_t) = \frac{K}{K-1}\left[\int_{-\infty}^{\infty} \phi\left(z - \frac{\tilde{\alpha}_t}{\sqrt{1 - \tilde{\alpha}_t^2}}\right)\Phi^{K-1}(z)\mathrm{d}z - \frac{1}{K}\right],} \quad (10)$$

where $\phi(z) = \exp(-z^2)/\sqrt{2\pi}$ is the standard Normal distribution and $\Phi(z) = \int_{-\infty}^{z}\exp(-t^2/2)\mathrm{d}z/\sqrt{2\pi}$ is its cumulative distribution function.

**Time Evolution of Marginals**  Next, we examine how the discrete marginal $P_t$ evolves with time as the continuous vector $\mathbf{w}_t$ undergoes Gaussian diffusion. In Suppl. A.2 we show that $P_t$ evolves as per the following linear ODE:

$$\frac{\mathrm{d}}{\mathrm{d}t}P_t = \underbrace{-\frac{\mathcal{T}'(\tilde{\alpha}_t)}{K\mathcal{T}(\tilde{\alpha}_t)}\left[\mathbf{1}\mathbf{1}^\top - K\mathbf{I}_K\right]}_{Q_t}P_t \quad (11)$$

where $\mathcal{T}'$ represents the time derivative of $\mathcal{T}$. From (2) and (3), we infer that (11) characterizes a Uniform-state discrete diffusion process with diffusion parameter $\mathcal{T}(\tilde{\alpha}_t)$. It is important to note that while the marginals of the discretized latents evolve according to a Markovian Uniform-state discrete diffusion process, a Gaussian diffusion trajectory after discretization, might not map to a discrete diffusion trajectory. We discuss this in detail in Suppl. A.3.

**Duality**  The implications of (9) and (11) are quite profound. These reveal a fundamental connection between Uniform-state discrete diffusion and Gaussian diffusion bridged by the $\arg\max$ operator:

> *The $\arg\max$ operation transforms Gaussian diffusion into Uniform-state diffusion, with the diffusion parameters related by (10).*

More formally, this can be expressed as:

$$q_t(\mathbf{z}_t|\mathbf{x}; \mathcal{T}(\tilde{\alpha}_t)) = [\arg\max]_\star \tilde{q}_t(\mathbf{w}_t|\mathbf{x}; \tilde{\alpha}_t) \quad (12)$$

where the $\star$ operator denotes the *pushforward* of the $K$-dimensional Gaussian density $\tilde{q}_t$ under the $\arg\max$ map, yielding a categorical distribution $q_t$ with $K$ classes.

Thus, as $\mathbf{x}$ diffuses in the discrete space via a Uniform-state process, there exists an underlying continuous-space representation in which $\mathbf{x}$ follows Gaussian diffusion, as illustrated in Fig. 1.

**ELBO**  Note that these two processes are separate Markov chains with no transitions between them, and they induce different variational bounds on the log-likelihood. Specifically, they yield distinct ELBOs: (5) for the discrete diffusion process and (7) for the Gausssian diffusion process.

**Theorem 3.1.** *The ELBO for the Uniform-state discrete diffusion is tighter than for the underlying Gaussian diffusion.*

We provide a proof in Suppl. A.4. Briefly, we show that:

$$\log p_\theta(\mathbf{x}) \geq \text{ELBO}\left(q, \bar{p}_\theta; \mathbf{x}\right) \geq \text{ELBO}\left(\tilde{q}, p_\theta; \mathbf{x}\right), \quad (13)$$

where $p_\theta$ is the denoiser in the Gaussian diffusion space, and $\bar{p}_\theta := [\arg\max]_\star p_\theta$ is the denoiser in the discrete diffusion space. Since the ELBO is inherently tighter in the discrete space, it is advantageous to design the denoising model to model discrete latents. Hence, we choose (5) as the training and evaluation objective. The term $f$ in (5) involves materializing the one-hot vector $\bar{\mathbf{x}}$, which increases memory usage and slows down training. We reformulate it with a Rao-Blackwellized objective (41) that avoids materializing one-hot vectors and also reduces training variance, resulting in faster training and lower perplexity (Sec. 5.1).

**Sampler**  To sample from Duo, we use ancestral sampling for USDMs (Sec. 2.1). Furthermore, to improve text quality, we propose *Greedy-Tail Sampler*, that improves the sample quality by decreasing the sample entropy in a manner similar to nucleus sampling in AR models models (Holtzman et al., 2020). Specifically, during the final denoising step, instead of sampling the clean sequence via $\tilde{\mathbf{x}} \sim p_{0|\delta}^\theta(.)$, we perform greedy decoding: $\tilde{\mathbf{x}} = \arg\max(p_{0|\delta}^\theta(.))$ where $\delta$ denotes the time discretization.

We exploit this duality to incorporate Gaussian diffusion into the design space of USDMs. This allows us to design a low-variance training algorithm that leads to faster training (Sec. 4.1) and a distillation scheme that unlocks few-step generation in diffusion language models (Sec. 4.2).

# 4. Applications

We now present two applications where discrete diffusion models benefit from leveraging the underlying Gaussian diffusion. In Sec. 4.1, we introduce a curriculum learning strategy that reduces training variance and leads to faster training. Then, in Sec. 4.2, we propose a distillation algorithm that cuts the number of sampling steps by two orders of magnitude with minimal impact on sample quality.

We extend our discrete diffusion framework to sequences $\mathbf{x}^{1:L} \sim q_{\text{data}}$ of length $L$. The forward process and the reverse process factorize independently over tokens as: $q_t(\mathbf{z}_t^{1:L}|\mathbf{x}^{1:L}; \alpha_t) = \prod_{\ell \in [L]} q_t(\mathbf{z}_t^\ell|\mathbf{x}^\ell; \alpha_t)$ based on (1) and $p_\theta(\mathbf{z}_s^{1:L}|\mathbf{z}_t^{1:L}) = \prod_{\ell \in [L]} q_{s|t}(\mathbf{z}_s^\ell|\mathbf{z}_t^{1:L}, \mathbf{x}_\theta^\ell(\mathbf{z}_t^{1:L}, t))$ based on (4), respectively. Here, $\mathbf{x}_\theta : \mathcal{V}^L \times [0, 1] \to \Delta^L$ denotes the denoising model. Consequently, the sequence-level NELBO decomposes into a sum of token-level losses:

$$\text{NELBO}(q, p_\theta; \mathbf{x}^{1:L})$$
$$= \mathbb{E}_{t \sim \mathcal{U}[0,1], q_t} \sum_{\ell \in [L]} f_{\text{Duo}}(\mathbf{z}_t^\ell, \mathbf{x}_\theta^\ell(\mathbf{z}_t^{1:L}, t), \alpha_t; \mathbf{x}^\ell). \quad (14)$$

where $f_{\text{Duo}}$ is defined in (41).

## 4.1. Faster Training using Curriculum Learning

Curriculum learning (Bengio et al., 2009) gradually exposes models to increasingly complex data, starting with simpler, easier-to-denoise noise patterns and progressing to more challenging ones. Here, we design a curriculum for USDMs by exploiting the underlying Gaussian diffusion.

Similar to relaxation methods in discrete gradient estimation (Jang et al., 2017; Maddison et al., 2017), our curriculum is centered around annealing the temperature parameter of a smooth approximation of arg max. We reformulate the NELBO for discrete diffusion in terms of arg max over Gaussian latents (Sec. 4.1.1). The denoising model is then trained to operate on the arg max of these Gaussian variables. We then relax the arg max using a tempered softmax (Sec. 4.1.2), which yields a lower-variance but biased estimator of the ELBO. Initially, the model operates on fully relaxed, continuous Gaussian latents. As training progresses, the temperature is gradually decreased, transitioning the model's inputs from soft (continuous) to hard (discrete). By the end of the curriculum, the model effectively operates on discrete latents, closing the gap between training and inference-time behavior.

### 4.1.1. DISCRETE NELBO WITH GAUSSIAN LATENTS

Consider the discrete diffusion NELBO (14), which marginalizes $f_{\text{Duo}}$ over the discrete latents $\mathbf{z}_t^{1:L} \sim q_t(.|\mathbf{x}^{1:L}; \alpha_t)$. Our goal is to re-express this objective in terms of Gaussian latents $\mathbf{w}_t^{1:L} \sim \tilde{q}_t(.|\mathbf{x}^{1:L}; \tilde{\alpha}_t)$ such that marginalizing over $\mathbf{w}_t^{1:L}$ yields the same numerical value

for the NELBO. In Suppl. B.1, we show:

$$\text{NELBO}(q, p_\theta; \mathbf{x}^{1:L})$$
$$= \mathbb{E}_{t, q_t(\mathbf{z}_t^{1:L}|\mathbf{x}^{1:L}; \alpha_t)} \sum_{\ell \in [L]} f_{\text{Duo}}(\mathbf{z}_t^\ell, \mathbf{x}_\theta^\ell(\mathbf{z}_t^{1:L}, t), \alpha_t; \mathbf{x}^\ell)$$
$$= \mathbb{E}_{t, \tilde{q}_t(\mathbf{w}_t^{1:L}|\mathbf{x}; \tilde{\alpha}_t)} \sum_{\ell \in [L]} f_{\text{Duo}}\Big(\mathbf{z}_t^\ell := \arg\max(\mathbf{w}_t^\ell),$$
$$\mathbf{x}_\theta\Big([\arg\max(\mathbf{w}_t^{\ell'})]_{\ell'=1}^L, t\Big), \alpha_t := \mathcal{T}(\tilde{\alpha}_t); \mathbf{x}^\ell\Big), \quad (15)$$

where $\alpha_t = \mathcal{T}(\tilde{\alpha}_t)$ is obtained via (10) from the Gaussian diffusion coefficient $\tilde{\alpha}_t$ and also verify this empirically. As discussed in Sec. 3, these are distinct Markov chains whose marginal distributions are related only through (12). This reparameterization underpins our curriculum learning strategy which we present in the next section.

### 4.1.2. LOW-VARIANCE TRAINING LOSS

To reduce training variance, we replace $\arg\max(\mathbf{w}_t^\ell)$ in the denoising model input (15) with a tempered softmax. We argue that this substitution eases recovery of the clean sequence from its noisy counterpart, and that the difficulty of this recovery is regulated by the temperature parameter.

As shown in prior work (Jang et al., 2017; Maddison et al., 2017), arg max is a limiting case of softmax:

$$\arg\max(\mathbf{w}_t^\ell) = \lim_{\tau \to 0^+} \text{softmax}(\mathbf{w}_t^\ell/\tau). \quad (16)$$

We relax this operation by setting the temperature parameter $\tau > 0$. While computing the NELBO in (15), note that the discrete diffusion parameter $\mathcal{T}(\tilde{\alpha}_t)$ spans the interval $[0, 1]$, as does its Gaussian counterpart $\tilde{\alpha}_t$. The diffusion transformation operator $\mathcal{T}$ (10) has a crucial property: as the vocabulary size $K$ increases, a small sub-interval $[a, 1]_{0 \le a < 1}$ within the domain of $\mathcal{T}$ is sufficient to map onto the full range $[0, 1]$. For instance, in Suppl. C.7, we observe that for $K = 30K$, when $\tilde{\alpha}_t \in [0.85, 1]$, the corresponding $\alpha_t = \mathcal{T}(\tilde{\alpha}_t)$ nearly spans the entire interval $[0, 1]$. This observation is counter-intuitive: since the Gaussian latents mostly resemble $\mathbf{x}$, one might expect the discrete NELBO to approach zero when evaluated with $\tilde{\alpha}_t$ restricted to such a narrow range. However, in practice, the NELBO remains largely unchanged. Why? The key reason lies in the discretization step. Even small amounts of Gaussian noise in $\mathbf{w}_\ell^t$ can cause the output of the arg max operation to change drastically, as it is highly sensitive to perturbations. As a result, much of the extra signal is lost due to discretization. To mitigate this, we allow the denoising model $\mathbf{x}_\theta$ to access the continuous latent $\mathbf{w}_t^\ell$ through a tempered softmax in (16). This relaxation helps preserve more of the signal, making the reconstruction task easier. In this way, the temperature parameter $\tau$ effectively controls the difficulty of the learning problem.

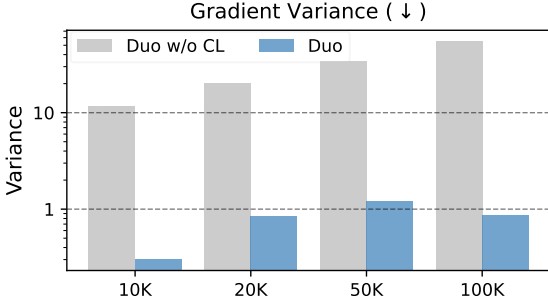

Figure 2: Curriculum learning drastically lowers the gradient variance in Duo trained with a fixed $\tau = 0.001$. The figure shows the summed gradient variance of the 100 weights with the highest variance, comparing Duo with CL (blue) and without CL (grey).

Hence, unlike prior discrete diffusion methods, we design the denoising model $\mathbf{x}_\theta : \Delta^L \cup \mathcal{V}^L \times [0, 1] \to \Delta^L$ to handle both continuous latents and discrete latents; see Suppl. C.2 for details. During training, we sample $t \sim \mathcal{U}[\beta, \gamma]_{0 \leq \beta < \gamma \leq 1}$ from a sub-interval so that $\tilde{\alpha}_t \in [a, b]_{0 \leq a < b \leq 1}$. Following Arriola et al. (2025), we set $b < 1$ as $\tilde{\alpha}_t = 1$ doesn't provide much training signal. Thus, we propose the following training loss:

$$\mathcal{L}_{\text{train}} = \mathbb{E}_{\mathbf{x}, t \sim \mathcal{U}[\beta, \gamma], \tilde{q}_t} \sum_{\ell \in [L]} f_{\text{Duo}} \left( \mathbf{z}_t^\ell := \arg\max(\mathbf{w}_t^\ell), \right.$$
$$\left. \mathbf{x}_\theta([\text{softmax}(\mathbf{w}_t^{\ell'}/\tau)]_{\ell'=1}^L, t), \, \alpha_t := \mathcal{T}(\tilde{\alpha}_t); \mathbf{x}^\ell \right). \quad (17)$$

This loss doesn't correspond to a valid NELBO because the denoising model operates on a continuous-time r.v., while the loss is defined for a discrete diffusion process. **It only becomes a valid NELBO in the limiting case** $\lim_{\tau \to 0^+}$ **with** $\beta = 0$ **and** $\gamma = 1$. During evaluation, we evaluate the model as a discrete diffusion model using (14). As shown in Figure 2 and Table 3, this approach results in lower training variance compared to previous MDMs and USDMs, ultimately improving model likelihood (Sec. 5.1).

### 4.2. Discrete Consistency Distillation

In this section, we present a new method to exploit the duality property of USDMs. This duality enables USDMs to adopt Consistency Distillation—a technique developed for Gaussian diffusion models to distil them into few-step generation models. However, standard discrete diffusion models cannot use this approach due to the absence of such PF-ODEs. To address this, we introduce Discrete Consistency Distillation (DCD), which sidesteps this limitation by utilizing the PF-ODE of the underlying Gaussian diffusion model to construct deterministic trajectories.

**Deterministic Discrete Trajectories (DDT)** Consistency Distillation relies on a PF-ODE parameterized by the denoising model. In our setting, the trained discrete denoiser $\mathbf{x}_\theta$ cannot be used to parameterize the ODE in the Gaussian space, as it operates only on discrete samples—the temperature $\tau$ in (17) is reduced to zero by the end of the training. To circumvent this, we construct a deterministic trajectory in Gaussian space by reversing the PF-ODE using an *optimal denoiser*, and then project this trajectory to the discrete domain. Let $\mathcal{P}_{\text{ODE}}$ denote such a trajectory. Given a clean data point $\mathbf{x}^{1:L} \sim q_{\text{data}}$ and Gaussian noise $\boldsymbol{\epsilon}^{1:L} = \{\boldsymbol{\epsilon}^\ell \sim \mathcal{N}(0, \mathbf{I}_K) \forall \ell \in [L]\}$, $\mathcal{P}_{\text{ODE}}(\mathbf{x}^{1:L}, \boldsymbol{\epsilon}^{1:L}) = \left\{[\tilde{\alpha}_t \mathbf{x}^\ell + \sqrt{1 - \tilde{\alpha}_t^2} \boldsymbol{\epsilon}^\ell]_{\ell=1}^L\right\}_{t \in [0,1]}$; see Suppl. B.2.1 for a detailed discussion. Next, we project this trajectory to the discrete space via the arg max operator:

$$\mathcal{P}_{\text{DDT}}(\mathbf{x}^{1:L}, \boldsymbol{\epsilon}^{1:L})$$
$$= \left\{[\arg\max(\tilde{\alpha}_t \mathbf{x}^\ell + \sqrt{1 - \tilde{\alpha}_t^2} \boldsymbol{\epsilon}^\ell)]_{\ell=1}^L\right\}_{t \in [0,1]}. \quad (18)$$

$\mathcal{P}_{\text{DDT}}$ serves as a proxy for the absence of a proper PF-ODE defined in the discrete space; see Fig. 5 for an illustrate example.

**Distillation** Given a teacher model $\mathbf{x}_{\theta^-}$, our goal is to distill it into a student model $\mathbf{x}_\theta$ that generates samples of similar quality but in fewer steps. To perform distillation, we sample an adjacent pair of latents $(\mathbf{z}_s^{1:L}, \mathbf{z}_t^{1:L}) \sim \{(\mathbf{z}_{j-\delta}^{1:L}, \mathbf{z}_j^{1:L}) | \mathbf{z}^{1:L}_{\{.\}} \in \mathcal{P}_{\text{DDT}}(\mathbf{x}^{1:L}, \boldsymbol{\epsilon}^{1:L}), j \in [\delta, 1]\}$ for a given step size $\delta \in [0, 1]$. Here, $\mathbf{z}_t^{1:L}$ is noisier than $\mathbf{z}_s^{1:L}$ and serves as the input to the student model. Let $\mathbf{x}_{\theta^-}(\mathbf{z}_s^{1:L}, s)$ and $\mathbf{x}_\theta(\mathbf{z}_t^{1:L}, t)$ denote the output distributions over clean samples produced by the teacher and the student models, respectively. Following Deschenaux & Gulcehre (2024), we train the student by minimizing the KL divergence between these distributions:

$$\mathcal{L}_{\text{DCD}}(\boldsymbol{\theta}; \boldsymbol{\theta}^-) = \sum_{\ell \in [L]} \text{D}_{\text{KL}} \left( \mathbf{x}_\theta^\ell(\mathbf{z}_t^{1:L}, t), \mathbf{x}_{\boldsymbol{\theta}^-}^\ell(\mathbf{z}_s^{1:L}, s) \right).$$
$$(19)$$

The distillation process proceeds in $N$ rounds, each consisting of $M$ training steps. At the end of each round, the teacher weights are updated with the current student weights. The full procedure is outlined in Algo. 1.

## 5. Experiments

We evaluate Duo on standard language modeling benchmarks, training on LM1B (Chelba et al., 2014) and OpenWebText (OWT) (Gokaslan et al., 2019) with sequence packing (Raffel et al., 2020). We train our models for 1M steps with a batch size of 512 on both datasets. For LM1B, we use a context length of 128 with the `bert-base-uncased` tokenizer (Devlin et al., 2018) with sequence packing (Ar-

**Algorithm 1** Discrete Consistency Distillation (DCD)

**Input:** Dataset $\mathcal{D}$, learning rate $\eta$, number of distillation rounds $N$, number of training iterations per round $M$, ema $\mu$, weights of the denoising model $\boldsymbol{\theta}$, weights of the EMA model $\boldsymbol{\theta}_{\text{ema}}$, discretization step $\delta$.

**for** $i = 1$ **to** $N$ **do**
    $\boldsymbol{\theta}^- \leftarrow \text{stopgrad}(\boldsymbol{\theta})$
    **for** $j = 1$ **to** $M$ **do**
        Sample $\mathbf{x}^{1:L} \sim \mathcal{D}$, $t \sim \mathcal{U}[0,1]$, and $\boldsymbol{\epsilon}^\ell \sim \mathcal{N}(0, I_K)$.
        $s \leftarrow \max(t - \delta, 0)$
        $\mathbf{z}_s^{1:L} \leftarrow [\arg\max(\tilde{\alpha}_s \mathbf{x}^\ell + \sqrt{1 - \tilde{\alpha}_s{}^2} \boldsymbol{\epsilon}^\ell)]_{\ell=1}^L$
        $\mathbf{z}_t^{1:L} \leftarrow [\arg\max(\tilde{\alpha}_t \mathbf{x}^\ell + \sqrt{1 - \tilde{\alpha}_t{}^2} \boldsymbol{\epsilon}^\ell)]_{\ell=1}^L$
        $\mathcal{L}_{\text{DCD}}(\boldsymbol{\theta}; \boldsymbol{\theta}^-) \leftarrow \text{D}_{\text{KL}}(\mathbf{x}_\theta(\mathbf{z}_t^{1:L}, t), \mathbf{x}_{\boldsymbol{\theta}^-}(\mathbf{z}_s^{1:L}, s))$
        $\boldsymbol{\theta} \leftarrow \boldsymbol{\theta} - \eta \nabla_{\boldsymbol{\theta}} \mathcal{L}_{\text{DCD}}(\boldsymbol{\theta}; \boldsymbol{\theta}^-)$
        $\boldsymbol{\theta}_{\text{ema}} \leftarrow \text{stopgrad}(\mu \boldsymbol{\theta}_{\text{ema}} + (1 - \mu)\boldsymbol{\theta})$
    **end for**
    $\delta \leftarrow 2 \cdot \delta$
**end for**
**return** $\boldsymbol{\theta}_{\text{ema}}$

---

riola et al., 2025; Austin et al., 2021) and without it (Sahoo et al., 2024a; Lou et al., 2023; He et al., 2022). For OWT, we use a context length of 1024 with the GPT-2 tokenizer (Radford et al., 2019). Following Sahoo et al. (2024a), we reserve the last 100K documents for validation. Our model is a 170M-parameter modified diffusion transformer (DiT) (Peebles & Xie, 2023) with rotary positional encoding (Su et al., 2023) and adaptive layer norm for conditioning on diffusion time, consistent with prior work (Lou et al., 2023; Sahoo et al., 2024a). Training is conducted on 8×H100s with `bfloat16` precision. We train Duo using (17), which requires computing the integral in (10). To reduce computation overhead, we pre-compute and cache 100K $(\tilde{\alpha}_t, \mathcal{T}(\tilde{\alpha}_t))$ tuples, significantly smaller than the denoising network. The Gaussian diffusion parameter $\tilde{\alpha}_t$ is parameterized using a linear schedule i.e., $(\tilde{\alpha}_t = 1-t)_{t\in[0,1]}$.

## 5.1. Improved Training

Our experiments show that (1) the proposed curriculum learning strategy (Sec. 4.1.2) **accelerates training by 2×and achieves a new state-of-the-art among USDMs** (Table 1), and (2) Duo performs competitively with Absorbing State diffusion across major language modeling benchmarks, even **surpassing AR models on 3/7 zero-shot PPL benchmarks** (Table 2).

**Experimental Setup**  The primary baselines for Duo are the leading USDMs (SEDD Uniform (Lou et al., 2023) and UDLM (Schiff et al., 2025)) and Gaussian diffusion method, PLAID (Gulrajani & Hashimoto, 2024). Additionally, we compare Duo with an AR model and MDMs such as MDLM (Sahoo et al., 2024a) (SOTA), SEDD Absorb (Lou

et al., 2023), and D3PM Absorb (Austin et al., 2021). While training Duo, we set $\tau$ as a function of the training iteration $n$ for Duo: $\tau = 0.001$ for the first 500K steps ($n < 500$K) with $\beta = 0.03$ and $\gamma = 0.15$ in (17), and $\tau = 0$ for the remaining steps up to 1M ($n \geq 500$K). To compute PPL for Duo, we use (14) with $\alpha_t = 1 - t$.

**Bias-variance Tradeoff**  First, we study the effect of $\tau$ on the training training dynamics in Fig. 8 by training Duo on the LM1B dataset over 150K steps with a fixed $\tau \in \{0, 0.001, 0.01, 0.1\}$. Here, $\tau = 0$ (blue) corresponds to (14), i.e., no curriculum. Recall that a larger $\tau$ introduces more bias but reduces training variance. Ideally, $\tau$ should strike a balance—minimizing both the bias (measured by deviation from the blue curve) and the variance in the loss curve. For $\tau = 0.1$ (red), the loss drops sharply, indicating excessive bias, making it suboptimal. As $\tau$ decreases to 0.01 (orange) and 0.001 (purple), the loss curves become more stable. Among them, $\tau = 0.001$ is the most desirable, as it closely follows the blue curve (low bias) while exhibiting significantly lower variance.

**Faster Training**  Notably, after just 10K steps of fine-tuning as a discrete diffusion model i.e., at 510K steps, Duo achieves a PPL of 35.2—almost 1.5 points better than UDLM trained for 1M steps—indicating that curriculum learning accelerates convergence by at least 2×. In Fig. 2, we compare the summed gradient variance of the top 100 weights with highest variance for Duo with (blue) and without (grey) curriculum. For these weights, we notice that curriculum learning reduces the gradient variance by an order of magnitude, which also manifests as lower loss variance in Fig. 7 and Table 3.

**Likelihood Evaluation**  On LM1B and OWT (Table 1), Duo outperforms previous USDMs and Gaussian diffusion models, notably SEDD Uniform and UDLM and shrinks the gap with absorbing diffusion below 2 PPL points. On LM1B, We retrained Plaid which attained a PPL of 89.9 in 100K steps while Duo achieves a PPL of 43.0 in the same number of steps. This result is excluded from the table due to incomplete training; see Suppl. C.1 for details.

**Zero-Shot Likelihood Evaluation**  We measure the zero-shot generalization of the models trained on OWT by evaluating their PPL on 7 other datasets. Following Sahoo et al. (2024a), our zero-shot datasets include the validation splits of Penn Tree Bank (PTB; Marcus et al. (1993)), Wiki-Text (Merity et al., 2016), LM1B, Lambada (Paperno et al., 2016), AG News (Zhang et al., 2015), and Scientific papers from ArXiv and Pubmed (Cohan et al., 2018). We observe that Duo outperforms SEDD Uniform and Plaid across all benchmarks. More notably, it achieves a better PPL score than SEDD Absorbing on 4/7 datasets, MDLM on 1/7, most

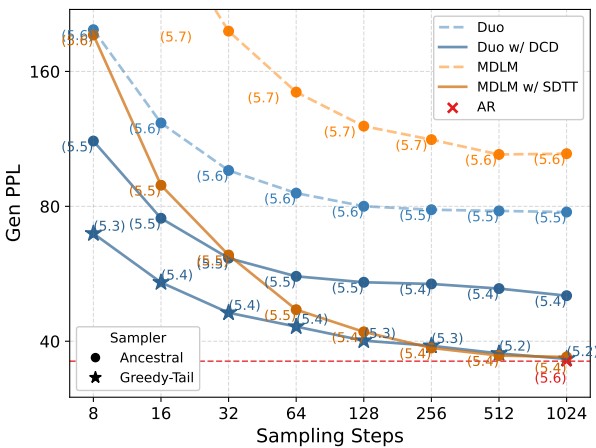

Figure 3: Sample quality comparison of Duo vs. MDLM. Duo outperforms MDLM in Gen PPL (↓) for base models and in low-NFE regime after 5 distillation rounds.

Table 1: Test perplexities (PPL; ↓) on LM1B. *Reported in He et al. (2022). Best uniform/Gaussian diffusion value is bolded. ¶Denotes the dataset didn't incorporate sentence packing. †Reported in Arriola et al. (2025). For diffusion models, we report the bound on the likelihood. Best diffusion value is underlined. ‡Denotes retrained models.

|  | LM1B¶ | LM1B | OWT |
|---|---|---|---|
| *Autoregressive* |  |  |  |
| Transformer‡ | 22.3 | 22.8† | 17.5 |
| *Diffusion (absorbing state)* |  |  |  |
| BERT-Mouth* (Wang & Cho, 2019) | - | 142.9 | - |
| D3PM Absorb (Austin et al., 2021) | - | 76.9 | - |
| DiffusionBert (He et al., 2022) | - | 63.8 | - |
| SEDD Absorb‡ (Lou et al., 2023) | 32.7 | - | 24.1 |
| MDLM (Sahoo et al., 2024a) | 27.0 | 31.8† | 23.2 |
| *Diffusion (Uniform-state / Gaussian)* |  |  |  |
| D3PM Uniform (Austin et al., 2021) | - | 137.9 | - |
| Diffusion-LM* (Li et al., 2022) | - | 118.6 | - |
| SEDD Uniform (Lou et al., 2023) | 40.3 | - | 29.7 |
| UDLM‡ (Schiff et al., 2025) | 31.3 | 36.7 | 27.4 |
| **Duo (Ours)** | **29.9** | **33.7** | **25.2** |

notably, outperforming an autoregressive transformer on 3/7 datasets.

**Ablation** Duo introduces two key improvements over UDLM: (i) a Rao-Blackwellized ELBO (41) and (ii) a low-variance training curriculum. As shown in Table 5, the overall 3-point improvement in PPL comes roughly equally from both components, with (41) accounting for about 1.7 points and the remainder from the curriculum.

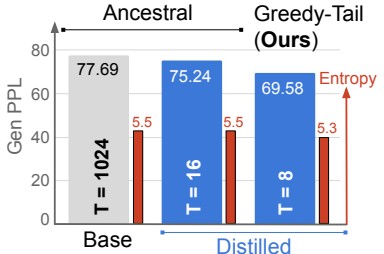

Figure 4: Sample quality comparison between the base Duo model and Duo distilled for 5 rounds using our DCD algorithm. The distilled model matches the base model's sample quality in just 16 steps (vs. 1024) with ancestral sampling. With our Greedy-Tail sampler, sampling steps can be further reduced to 8, achieving slightly better Gen PPL and lower entropy.

### 5.2. Improved Sampling

Our sampling experiments show that for undistilled models, (1) **Duo generates higher-quality samples than all previous diffusion models** (Fig. 9); (2) combining DCD with the Greedy-Tail sampler **reduces the number of sampling steps by two orders of magnitude** (Fig. 4); and (3) the distilled Duo model outperforms a distilled MDLM model, especially in the low NFE regime.

**Experimental Setup** We distill Duo on OWT using DCD, following the same setup as our main baseline—MDLM distilled with SDTT (Deschenaux & Gulcehre, 2024), a distillation method for MDMs. We run $N = 5$ distillation rounds, starting with discretization step $\delta = 1/512$ in Algo. 1 and doubling it every $M = 10K$ steps. To assess sample quality, we report GPT-2 Large generative perplexity (Gen PPL) and average sequence entropy for diversity. As noted by Zheng et al. (2024), masked diffusion models can suffer from low diversity and misleading Gen PPL under low-precision sampling. To address this, we use float64 precision in all sampling experiments.

**Sample Quality** In Fig. 9, Duo consistently outperforms all MDMs and USDMs in Gen PPL across sampling steps $T \in \{8, \ldots, 1024\}$, with particularly strong performance at low NFEs. In Fig. 3, we compare Duo distilled with DCD to MDLM distilled with SDTT after 5 rounds (entropy values in parentheses). Under ancestral sampling, Duo performs significantly better than MDLM for $T \leq 32$. This is because MDMs generate many tokens independently and cannot revise them, leading to incoherence at low NFEs. In contrast, USDMs are self-correcting: they can fix earlier errors in later denoising steps. At higher NFEs, MDLM outperforms the distilled Duo. While MDLM (with SDTT) matches the Gen PPL of the AR model, its lower entropy (5.4 vs. 5.6) indicates reduced diversity.

Table 2: Zero-shot perplexities (↓) of models trained for 1M steps on OWT. All perplexities for diffusion models are upper bounds. † Taken from Sahoo et al. (2024a). ¶ Taken from (Lou et al., 2023) models were trained for 1.3Msteps as opposed to the baselines that were trained for 1Msteps. All perplexities for diffusion models are upper bounds. Best uniform / Gaussian diffusion values are **bolded** and diffusion values better than AR are underlined. ‡ denotes retrained model.

|  | PTB | Wikitext | LM1B | Lambada | AG News | Pubmed | Arxiv |
|---|---|---|---|---|---|---|---|
| *Autoregressive* | | | | | | | |
| Transformer† | 82.05 | 25.75 | 51.25 | 51.28 | 52.09 | 49.01 | 41.73 |
| *Diffusion (absorbing state)* | | | | | | | |
| SEDD Absorb† | 100.09 | 34.28 | 68.20 | 49.86 | 62.09 | 44.53 | 38.48 |
| D3PM Absorb¶ | 200.82 | 50.86 | 138.92 | 93.47 | - | - | - |
| MDLM† | 95.26 | 32.83 | 67.01 | 47.52 | 61.15 | 41.89 | 37.37 |
| *Diffusion (Uniform-state / Gaussian)* | | | | | | | |
| SEDD Uniform‡ | 105.51 | 41.10 | 82.62 | 57.29 | 82.64 | 55.89 | 50.86 |
| Plaid¶ | 142.60 | 50.86 | 91.12 | 57.28 | - | - | - |
| UDLM‡ | 112.82 | 39.42 | 77.59 | 53.57 | 80.96 | 50.98 | 44.08 |
| **Duo (Ours)** | **89.35** | **33.57** | **73.86** | **49.78** | **67.81** | **44.48** | **40.39** |

**Greedy-Tail vs Ancestral Sampler** In Fig. 3, the Greedy-Tail sampler improves Gen PPL by reducing sample entropy. In Fig. 4, we observe that the ancestral sampler enables a 64× speedup (reducing NFEs from 1024 to 16) while maintaining Gen PPL. Greedy-Tail offers an even greater 128× speedup, with a slight drop in entropy. Interestingly, as shown in Fig. 11 and Fig. 12, each distillation round improves both Gen PPL and diversity when using the Greedy-Tail sampler—unlike ancestral sampling—suggesting that Greedy-Tail is particularly effective for distilled models.

**Ablation** In Algo. 1, we use the denoising model weights as the teacher, deviating from the common practice of using EMA weights in consistency models. To test this choice, we modify Algo. 1 to use EMA weights (Algo. 2) instead. As shown in Fig. 10, using the denoising model as a directly leads to a better distilled model.

## 6. Related Work

**Diffusion Language Models** Prior work on discrete diffusion language models operates strictly in discrete space (Austin et al., 2021; Lou et al., 2023; Sahoo et al., 2024a; Schiff et al., 2025; Arriola et al., 2025). In contrast, we expand their design space by incorporating continuous-state Gaussian diffusion, showing that training USDMs in this setting not only accelerates convergence but also improves model likelihood. Li et al. (2022); Han et al. (2022); Dieleman et al. (2022); Gulrajani & Hashimoto (2024) use Gaussian diffusion for language modeling by injecting noise into the continuous embeddings of discrete tokens. Duo takes a different approach: it defines the diffusion process directly over one-hot token representations, rather than their embeddings. This hybrid treatment achieves better performance than methods confined to either domain alone.

**Distillation for Faster Sampling** Distillation in Gaussian diffusion leverages PF-ODEs (Luhman & Luhman, 2021; Salimans & Ho, 2022; Song et al., 2023), which are unavailable in discrete space. Our method circumvents this by defining PF-ODEs using an optimal denoiser in Gaussian space and mapping them to the discrete domain. In contrast, Deschenaux & Gulcehre (2024) tackle this issue by performing distillation along stochastic trajectories due to the lack of deterministic ones. We observe that our method Duo surpasses all previous approaches at few-step generation.

**Argmax Differentiation** The softmax annealing trick was originally proposed by Jang et al. (2017); Maddison et al. (2017) to enable backpropagation through an $\arg\max$ operation. The core novelty of our work is in establishing a fundamental connection between discrete and Gaussian diffusion via the $\arg\max$ operator. We show that the softmax annealing trick can be naturally repurposed to design a low-variance training curriculum for USDMs.

## 7. Conclusion

In this work, we established a theoretical connection between continuous-space Gaussian diffusion models and discrete-space Uniform-state diffusion models. We leveraged this connection to achieve a 2× speed-up in training (Sec. 5.1) convergence and a two-orders-of-magnitude improvement in sampling speed (Sec. 5.2). While USDMs trail MDMs in terms of perplexity, we showed that they outperform in few-step generation. We hope that our theoretical foundation opens up new avenues of future research which would leverage this connection to improve USDMs by borrowing techniques from Gaussian diffusion—a connection non-existent for MDMs.

## Impact Statement

This paper presents work whose goal is to advance the field of Machine Learning. There are many potential societal consequences of our work, specifically those related to the generation of synthetic text. Our work can also be applied to the design of biological sequences, which carries both potential benefits and risks.

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

# Contents

# Appendices

## A. The Diffusion Duality

Let $\mathbf{x} \in \mathcal{V}$ s.t. $\mathbf{x}_k = 1$ i.e., $\mathbf{x}$ contains 1 at the $k^{\text{th}}$ index. Consider a r.v. $\mathbf{y} = \tilde{\alpha}_t \mathbf{x} + \tilde{\sigma}_t \boldsymbol{\epsilon}$ where $\boldsymbol{\epsilon} \sim \mathcal{N}(0, \mathbf{I}_K)$ and $\tilde{\sigma}_t = \sqrt{1 - \tilde{\alpha}_t{}^2}$.

### A.1. Discrete Marginals

Our goal in this section is to derive the pmf of the r.v. $\arg\max(\mathbf{y})$. The proof has three parts. In **part 1**, we derive pdf of the the random variables $\mathbf{y}_k$ and $\mathbf{y}_{i \neq k}$. Next in **part 2**, we derive the pdf of the random variable $Z_{\neq k} = \max(\{\mathbf{y}_i : i \neq k\})$. Finally in **part 3**, we derive the distribution of $\max(Z_{\neq k}, \mathbf{y}_k)$ which is the key to constructing the pmf of the r.v. $\arg\max(\mathbf{y})$.

**Part 1**  It can be easily seen that every entry in $\mathbf{y}$ is a Gaussian r.v. with

$$\mathbf{y}_k \sim \mathcal{N}(\tilde{\alpha}_t, \tilde{\sigma}_t^2) \tag{20}$$

$$\mathbf{y}_{i \neq k} \sim \mathcal{N}(0, \tilde{\sigma}_t^2). \tag{21}$$

**Part 2**  Since, $\mathbf{y}_{i \neq k}$ follows a Gaussian distribution with 0 mean and $\tilde{\sigma}_t$ standard deviation, the probability of $\mathbf{y}_{i \neq k} < l$ where $l \in \mathbb{R}$ is

$$P(\mathbf{y}_{i \neq k} < l) = \Phi\left(\frac{l}{\tilde{\sigma}_t}\right) \tag{22}$$

where $\Phi(z) = \int_{-\infty}^{z} \exp(-t^2/2) \mathrm{d}z / \sqrt{2\pi}$ is the cumulative distribution function of the Gaussian distribution. This allows us to compute the pdf of the r.v. $Z_{\neq k} = \max(\{\mathbf{y}_i : i \neq k\})$ in the following manner:

$$P(Z_{\neq k} < l) = \Pi_{i \neq k} P(\mathbf{y}_i < l) = \Phi^{K-1}\left(\frac{l}{\tilde{\sigma}_t}\right), \tag{23}$$

where $P(Z_{\neq k} < l)$ is the probability that $Z_{\neq k} < l$ for $l \in \mathbb{R}$.

**Part 3**  Let $P(\arg\max(\mathbf{y})_k = 1)$ denote the probability that the index $k$ is the index of the maximum entry in $\mathbf{y}$. This is equal to the probability of every other entry $\mathbf{y}_{i \neq k} < \mathbf{y}_k$. Let $\phi(z) = \exp(-z^2)/\sqrt{2\pi}$ denote the standard Normal distribution. Hence,

$$P(\arg\max(\mathbf{y})_k = 1) = P(Z_{\neq k} < \mathbf{y}_k)$$

$$= \int_{-\infty}^{\infty} P(Z_{\neq k} < l) P(\mathbf{y}_k = l) \mathrm{d}l$$

$$= \int_{-\infty}^{\infty} P(Z_{\neq k} < l) \left[\frac{1}{\tilde{\sigma}_t} \phi\left(\frac{l - \tilde{\alpha}_t}{\tilde{\sigma}_t}\right)\right] \mathrm{d}l \qquad \text{From (20)}$$

$$= \int_{-\infty}^{\infty} \Phi^{K-1}\left(\frac{l}{\tilde{\sigma}_t}\right) \left[\frac{1}{\tilde{\sigma}_t} \phi\left(\frac{l - \tilde{\alpha}_t}{\tilde{\sigma}_t}\right)\right] \mathrm{d}l \qquad \text{From (23)}$$

$$= \int_{-\infty}^{\infty} \Phi^{K-1}\left(\tilde{l}\right) \phi\left(\tilde{l} - \frac{\tilde{\alpha}_t}{\tilde{\sigma}_t}\right) \mathrm{d}\tilde{l} \qquad \text{Substituting } \tilde{l} = l/\tilde{\sigma}_t$$

$$= \int_{-\infty}^{\infty} \phi\left(\tilde{l} - \frac{\tilde{\alpha}_t}{\sqrt{1 - \tilde{\alpha}_t^2}}\right) \Phi^{K-1}\left(\tilde{l}\right) \mathrm{d}\tilde{l}. \tag{24}$$

Note that the indices $i \neq k$ and $j \neq k$ have the same probability of being the indices of maximum entry in $\mathbf{y}$ because both r.v.s $\mathbf{y}_{i \neq k}$ and $\mathbf{y}_{j \neq k}$ have the same pmf specified by (21). Thus,

$$P(\arg\max(\mathbf{y})_{i \neq k} = 1) = P(\arg\max(\mathbf{y})_{j \neq k} = 1) \quad \forall 0 \leq i \neq k < K, 0 \leq j \neq k < K. \tag{25}$$

Thus we can compute $P(\arg\max(\mathbf{y})_{i \neq k} = 1)$ in the following manner:

$$\sum_i P(\arg\max(\mathbf{y})_i = 1) = 1$$

$$\implies P(\arg\max(\mathbf{y})_k = 1) + \sum_{i \neq k} P(\arg\max(\mathbf{y})_i = 1) = 1$$

$$\implies P(\arg\max(\mathbf{y})_k = 1) + (K-1)P(\arg\max(\mathbf{y})_{i \neq k} = 1) = 1 \qquad \text{From (25)}$$

$$\implies P(\arg\max(\mathbf{y})_{i \neq k} = 1) = \frac{1}{K-1}\left[1 - P(\arg\max(\mathbf{y})_k = 1)\right]$$

$$\implies P(\arg\max(\mathbf{y})_{i \neq k} = 1) = \frac{1}{K-1}\left[1 - \int_{-\infty}^{\infty} \phi\left(\tilde{l} - \frac{\tilde{\alpha}_t}{\sqrt{1 - \tilde{\alpha}_t^2}}\right) \Phi^{K-1}\left(\tilde{l}\right) \mathrm{d}\tilde{l}\right] \qquad \text{From (24)} \tag{26}$$

Let $\beta_t = P(\arg\max(\mathbf{y})_{i \neq k} = 1)$. Then, from (24) and (26) we have $P(\arg\max(\mathbf{y})_{i=k} = 1) = \beta_t + (1 - K)\beta_t$. Thus,

$$P(\arg\max(\mathbf{y})_i = 1) = \begin{cases} \beta_t, & i \neq k \\ \beta_t + (1 - K)\beta_t. & i = k \end{cases} \tag{27}$$

(27) can be written in vectorized form in the following manner:

$$P(\arg\max(\mathbf{y})) = \mathrm{Cat}(.; \beta_t \mathbf{1} + (1 - K\beta_t)\mathbf{x}). \tag{28}$$

## A.2. Time Evolution of Probability Densities of Discrete Marginals

Let $p_t$ denote $P(\arg\max(\mathbf{y}))$ in (28). It's time-derivative $\frac{\mathrm{d}}{\mathrm{d}t}p_t$ is as follows:

$$\frac{\mathrm{d}}{\mathrm{d}t}p_t = \beta_t' \mathbf{1} - K\beta_t' \mathbf{x}$$

$$= \beta_t'(\mathbf{1} - K\mathbf{x})$$

$$= \frac{\beta_t'}{1 - K\beta_t}(1 - K\beta_t)(\mathbf{1} - K\mathbf{x})$$

$$= \frac{\beta_t'}{1 - K\beta_t}(\beta_t K \mathbf{1} - \beta_t K \mathbf{1} + (1 - K\beta_t)(\mathbf{1} - K\mathbf{x}))$$

$$= \frac{\beta_t'}{1 - K\beta_t}(\beta_t[\mathbf{1}\mathbf{1}^\top]\mathbf{1} - \beta_t K \mathbf{1} + (1 - K\beta_t)(\mathbf{1} - K\mathbf{x}))$$

$$= \frac{\beta_t'}{1 - K\beta_t}(\beta_t([\mathbf{1}\mathbf{1}^\top]\mathbf{1} - K\mathbf{1}) + (1 - K\beta_t)(\mathbf{1} - K\mathbf{x}))$$

$$= \frac{\beta_t'}{1 - K\beta_t}(\beta_t[\mathbf{1}\mathbf{1}^\top - K\mathbf{I}_K]\mathbf{1} + (1 - K\beta_t)(\mathbf{1} - K\mathbf{x}))$$

$$= \frac{\beta_t'}{1 - K\beta_t}\left(\beta_t[\mathbf{1}\mathbf{1}^\top - K\mathbf{I}_K]\mathbf{1} + (1 - K\beta_t)(\mathbf{1}\mathbf{1}^\top\mathbf{x} - K\mathbf{x})\right)$$

$$= \frac{\beta_t'}{1 - K\beta_t}\left(\beta_t[\mathbf{1}\mathbf{1}^\top - K\mathbf{I}_K]\mathbf{1} + (1 - K\beta_t)[\mathbf{1}\mathbf{1}^\top - K\mathbf{I}_K]\mathbf{x}\right)$$

$$= \frac{\beta_t'}{1 - K\beta_t}[\mathbf{1}\mathbf{1}^\top - K\mathbf{I}_K][\beta_t\mathbf{1} + (1 - K\beta_t)\mathbf{x}]$$

$$= \frac{\beta_t'}{1 - K\beta_t}[\mathbf{1}\mathbf{1}^\top - K\mathbf{I}_K]p_t \tag{29}$$

Let $\alpha_t = 1 - K\beta_t$. The functional form of $\alpha_t$ is given as:

$$\alpha_t = 1 - K\beta_t$$

$$= 1 - K\frac{1}{K-1}\left[1 - \int_{-\infty}^{\infty}\phi\left(\tilde{l} - \frac{\tilde{\alpha}_t}{\sqrt{1-\tilde{\alpha}_t{}^2}}\right)\Phi^{K-1}\left(\tilde{l}\right)\mathrm{d}\tilde{l}\right]$$

$$= 1 - \frac{K}{K-1} + \frac{K}{K-1}\int_{-\infty}^{\infty}\phi\left(\tilde{l} - \frac{\tilde{\alpha}_t}{\sqrt{1-\tilde{\alpha}_t{}^2}}\right)\Phi^{K-1}\left(\tilde{l}\right)\mathrm{d}\tilde{l}$$

$$= \frac{K}{K-1}\int_{-\infty}^{\infty}\phi\left(\tilde{l} - \frac{\tilde{\alpha}_t}{\sqrt{1-\tilde{\alpha}_t{}^2}}\right)\Phi^{K-1}\left(\tilde{l}\right)\mathrm{d}\tilde{l} - \frac{1}{K-1}$$

$$= \frac{K}{K-1}\left[\int_{-\infty}^{\infty}\phi\left(\tilde{l} - \frac{\tilde{\alpha}_t}{\sqrt{1-\tilde{\alpha}_t{}^2}}\right)\Phi^{K-1}\left(\tilde{l}\right)\mathrm{d}\tilde{l} - \frac{1}{K}\right] \tag{30}$$

Substituting $\beta_t = (1 - \alpha_t)/K$ in (28) and (29), we get:

$$p_t = \mathrm{Cat}(.; \alpha_t\mathbf{x} + (1 - \alpha_t)\pi) \tag{31}$$

$$\frac{\mathrm{d}}{\mathrm{d}t}p_t = -\frac{\alpha_t'}{K\alpha_t}[\mathbf{1}\mathbf{1}^\top - K\mathbf{I}_K]p_t \tag{32}$$

where $\alpha_t'$ denotes the time-derivative of $\alpha_t$. Let $\mathbf{z}_t = \arg\max(\mathbf{y})$. The pmf of $\mathbf{z}_t$ is specified in (31) which evolves according to an Ordinary Differential Equation (ODE) (32). This pmf and the ODE are the unique signatures of a Uniform-state discrete diffusion process (Lou et al., 2023; Schiff et al., 2025). This concludes our proof.

### A.3. Relationship between Gaussian and Discrete Diffusion Trajectories

Let $\mathbf{x} \in \mathcal{V}$ undergo Gaussian diffusion, and let $\{\mathbf{w}_t\}_{t\in[0,1]}$ denote its continuous trajectory. The question is: does the corresponding discretized trajectory $\{\mathbf{z}_t := \arg\max(\mathbf{w}_t)\}_{t\in[0,1]}$ correspond to a valid discrete diffusion trajectory?

To answer this, we must analyze how the $\arg\max$ operation relates the Gaussian transition kernel $\tilde{q}_{t|s}(.|\mathbf{w}_s)$

$$\mathbf{w}_t \sim \tilde{q}_{t|s}(.|\mathbf{w}_s) = \mathcal{N}(\tilde{\alpha}_{t|s}\mathbf{w}_s, (1 - \tilde{\alpha}_{t|s}{}^2)\mathbf{I}_K), \tag{33}$$

which maps $\mathbf{w}_s \to \mathbf{w}_t$, to the discrete diffusion transition kernel $q_{t|s}(.|\mathbf{z}_s)$:

$$\mathbf{z}_t \sim q_{t|s}(.|\mathbf{z}_s) = \mathrm{Cat}(.; \alpha_{t|s}\mathbf{z}_s + (1 - \alpha_{t|s})\boldsymbol{\pi}), \tag{34}$$

which models the transition $\mathbf{z}_s := \arg\max(\mathbf{w}_s) \to \mathbf{z}_t := \arg\max(\mathbf{w}_t)$ for $0 \le s < t \le 1$, where $\alpha_{t|s} = \alpha_t/\alpha_s$ and $\tilde{\alpha}_{t|s} = \tilde{\alpha}_t/\tilde{\alpha}_s$.

A necessary and sufficient condition for the discretized trajectory $\{\mathbf{z}_t := \arg\max(\mathbf{w}_t)\}_{t\in[0,1]}$ to follow a discrete diffusion process is that the *pushforward* distribution $[\arg\max]_\star\tilde{q}_{t|s}(.\mid\mathbf{w}_s)$ must equal $q_{t|s}(.\mid\mathbf{z}_s)$. We will now show that this does not hold.

Let $k$ denote the index such that $\mathbf{x}_k = 1$. First, consider $q_{t|s}(.\mid\mathbf{z}_s)$. From (34), it is clear that for any $i, j \ne k$, the probabilities satisfy

$$q_{t|s}(\bar{\mathbf{z}}_i = 1 \mid \mathbf{z}_s) = q_{t|s}(\bar{\mathbf{z}}_j = 1 \mid \mathbf{z}_s),$$

where $\bar{\mathbf{z}} \in \mathcal{V}$ is a discrete random variable.

Now define $P \coloneqq [\arg\max]_\star \tilde{q}_{t|s}(. \mid \mathbf{w}_s)$. Using the same argument as in Suppl. A.1, we can show:

$$
\begin{aligned}
P(\arg\max(\bar{\mathbf{w}})_i = 1) &= P(\max(\{\bar{\mathbf{w}}_j : j \neq i\}) < \bar{\mathbf{w}}_i) \\
&= \int_{-\infty}^{\infty} \prod_{j \neq i} P(\bar{\mathbf{w}}_j < l) P(\bar{\mathbf{w}}_i = l) \mathrm{d}l \\
&= \int_{-\infty}^{\infty} \prod_{j \neq i} \Phi\left( \frac{l - \tilde{\alpha}_{t|s}(\mathbf{w}_s)_j}{\sqrt{1 - \tilde{\alpha}_{t|s}^2}} \right) \left[ \frac{1}{\sqrt{1 - \tilde{\alpha}_{t|s}^2}} \phi\left( \frac{l - \tilde{\alpha}_{t|s}(\mathbf{w}_s)_i}{\sqrt{1 - \tilde{\alpha}_{t|s}^2}} \right) \right] \mathrm{d}l \qquad \text{From (33)}
\end{aligned}
\tag{35}
$$

where $\phi(z) = \exp(-z^2/2)/\sqrt{2\pi}$ is the standard normal density and $\Phi(z) = \int -\infty^z \exp(-t^2/2), \mathrm{d}t/\sqrt{2\pi}$ is its cumulative distribution function.

Clearly, from (35), we observe that $P(\arg\max(\bar{\mathbf{w}})_i = 1) = P(\arg\max(\bar{\mathbf{w}})_j = 1)$ for $i, j \neq k$ if and only if $(\mathbf{w}_s)_i = (\mathbf{w}_s)_j$ for all $i, j \neq k$. This condition rarely holds (in fact, the probability of exact equality is essentially zero); hence,

$$
q_{t|s}(. \mid \mathbf{z}_s \coloneqq \arg\max(\mathbf{w}_s)) \neq [\arg\max]_\star \tilde{q}_{t|s}(. \mid \mathbf{w}_s).
\tag{36}
$$

Therefore, **the discretized trajectory** $\{\mathbf{z}_t \coloneqq \arg\max(\mathbf{w}_t)\}_{t \in [0,1]}$ **does not necessarily follow a discrete diffusion process.**

### A.4. Gaussian ELBO vs Discrete ELBO

CSISZAR (1967); Cover & Thomas (2012) show that for certain statistical divergences $D$—such as the Kullback–Leibler (KL) divergence, Total Variation Distance (TVD), or Rényi divergence—and for any Markov kernel $k : X \to Y$, the following inequality holds:

$$
D([k]_\star p, [k]_\star q) \leq D(p, q),
\tag{37}
$$

where $p$ and $q$ denote probability distributions on $X$, and $[k]_\star$ denotes the pushforward operation induced by the kernel $k$.

In our case, the NELBO for the Gaussian diffusion process is given as $\mathrm{D}_{\mathrm{KL}}(\tilde{q}, p_\theta)$ (7). Now let's consider the the NELBO for the USDMs. The forward process is $q = [\arg\max]_\star \tilde{q}$ (12), and we define the reverse process as the pushforward of the reverse process of the Gaussian diffusion, i.e., $\bar{p}_\theta \coloneqq [\arg\max]_\star p_\theta$. Then,

$$
\begin{aligned}
\mathrm{D}_{\mathrm{KL}}(q, \bar{p}_\theta) &= \mathrm{D}_{\mathrm{KL}}([\arg\max]_\star \tilde{q}, [\arg\max]_\star p_\theta) \\
&\implies \mathrm{D}_{\mathrm{KL}}(q, \bar{p}_\theta) \leq \mathrm{D}_{\mathrm{KL}}(\tilde{q}, p_\theta) && \text{From (37)} \\
&\implies \mathrm{ELBO}(\tilde{q}_t, p_\theta; \mathbf{x}) \leq \mathrm{ELBO}(q_t, \bar{p}_\theta; \mathbf{x}) && \text{From (5) and (7)} \\
&\implies \log p_\theta(\mathbf{x}) \geq \mathrm{ELBO}(q_t, \bar{p}_\theta; \mathbf{x}) \geq \mathrm{ELBO}(\tilde{q}_t, p_\theta; \mathbf{x})
\end{aligned}
\tag{38}
$$

Thus, the ELBO in the Gaussian space is "looser" or worse than the ELBO in the discrete space. An alternate proof is provided in Suppl. C.3 of Mena et al. (2018).

### A.5. Negative Evidence Lower Bound

Schiff et al. (2025) showt that the NELBO for USDMs is given as:

$$
\mathrm{NELBO}(q, p_\theta; \mathbf{x}) = \mathbb{E}_{t \sim \mathcal{U}[0,1], q_t(\mathbf{z}_t | \mathbf{x}; \alpha_t)} f(\mathbf{z}_t, \mathbf{x}_\theta(\mathbf{z}_t, t), \alpha_t; \mathbf{x}),
\tag{39}
$$

where $f$ is defined as:

$$
f(\mathbf{z}_t, \mathbf{x}_\theta(\mathbf{z}_t, t), \alpha_t; \mathbf{x}) = -\frac{\alpha_t'}{K\alpha_t} \left[ \frac{K}{\bar{\mathbf{x}}_i} - \frac{K}{(\bar{\mathbf{x}}_\theta)_i} - \sum_j \frac{\bar{\mathbf{x}}_j}{\bar{\mathbf{x}}_i} \log \frac{(\bar{\mathbf{x}}_\theta)_i \cdot \bar{\mathbf{x}}_j}{(\bar{\mathbf{x}}_\theta)_j \cdot \bar{\mathbf{x}}_i} \right].
\tag{40}
$$

where the subscript $i$ denotes the $i^{\text{th}}$ index of a vector, $\bar{\mathbf{x}} = K\alpha_t \mathbf{x} + (1 - \alpha_t)\mathbf{1}$, $\bar{\mathbf{x}}_\theta = K\alpha_t \mathbf{x}_\theta(\mathbf{z}_t, t) + (1 - \alpha_t)\mathbf{1}$, $\alpha_t'$ denotes the time-derivative of the $\alpha_t$, and we define $i = \arg\max_{j \in [K]}(\mathbf{z}_t)_j$ to be the non-zero entry of $\mathbf{z}_t$.

**Rao Blackwellized NELBO (Ours)**   To reduce GPU memory usage and training time, we rewrite the original ELBO objective in (40) by eliminating the need to explicitly materialize the one-hot vector $\bar{\mathbf{x}}$. This leads to a more efficient formulation that preserves fidelity while significantly improving practicality. The resulting objective, shown in (41), not only removes $\bar{\mathbf{x}}$ but also applies Rao-Blackwellization to analytically compute certain expectations, thereby reducing variance. We now derive the improved loss:

$$f_{\text{Duo}}(\mathbf{z}_t, \mathbf{x}_\theta(\mathbf{z}_t, t), \alpha_t; \mathbf{x})$$

$$= -\frac{\alpha_t'}{K\alpha_t}\left[\frac{K}{\bar{\mathbf{x}}_i} - \frac{K}{(\bar{\mathbf{x}}_\theta)_i} - \sum_j \frac{\bar{\mathbf{x}}_j}{\bar{\mathbf{x}}_i}\log\frac{(\bar{\mathbf{x}}_\theta)_i \cdot \bar{\mathbf{x}}_j}{(\bar{\mathbf{x}}_\theta)_j \cdot \bar{\mathbf{x}}_i}\right].$$

$$= -\frac{\alpha_t'}{K\alpha_t}\left[\frac{K}{\bar{\mathbf{x}}_i} - \frac{K}{(\bar{\mathbf{x}}_\theta)_i} - \sum_j \frac{\bar{\mathbf{x}}_j}{\bar{\mathbf{x}}_i}\log\frac{(\bar{\mathbf{x}}_\theta)_i}{(\bar{\mathbf{x}}_\theta)_j} - \sum_j \frac{\bar{\mathbf{x}}_j}{\bar{\mathbf{x}}_i}\log\frac{\bar{\mathbf{x}}_j}{\bar{\mathbf{x}}_i}\right]$$

Let $\kappa_t = (1-\alpha_t)/(K\alpha_t + 1 - \alpha_t)$,

$$= -\frac{\alpha_t'}{K\alpha_t}\left[\frac{K}{\bar{\mathbf{x}}_i} - \frac{K}{(\bar{\mathbf{x}}_\theta)_i} - \sum_j \frac{\bar{\mathbf{x}}_j}{\bar{\mathbf{x}}_i}\log\frac{(\bar{\mathbf{x}}_\theta)_i}{(\bar{\mathbf{x}}_\theta)_j} - \left((K-1)\kappa_t\mathbb{1}_{\mathbf{z}_t=\mathbf{x}} - \frac{1}{\kappa_t}\mathbb{1}_{\mathbf{z}_t\neq\mathbf{x}}\right)\log\kappa_t\right]$$

Let $m$ denote the index in $\mathbf{x}$ corresponding to 1, i.e., $\mathbf{x}_m = 1$,

$$= -\frac{\alpha_t'}{K\alpha_t}\left[\frac{K}{\bar{\mathbf{x}}_i} - \frac{K}{(\bar{\mathbf{x}}_\theta)_i} - (\kappa_t\mathbb{1}_{\mathbf{z}_t=\mathbf{x}} + \mathbb{1}_{\mathbf{z}_t\neq\mathbf{x}})\sum_j \log\frac{(\bar{\mathbf{x}}_\theta)_i}{(\bar{\mathbf{x}}_\theta)_j} - K\frac{\alpha_t}{1-\alpha_t}\log\frac{(\bar{\mathbf{x}}_\theta)_i}{(\bar{\mathbf{x}}_\theta)_m}\mathbb{1}_{\mathbf{z}_t\neq\mathbf{x}}\right.$$

$$\left. - \left((K-1)\kappa_t\mathbb{1}_{\mathbf{z}_t=\mathbf{x}} - \frac{1}{\kappa_t}\mathbb{1}_{\mathbf{z}_t\neq\mathbf{x}}\right)\log\kappa_t\right]. \tag{41}$$

This reformulation provides an efficient and low-variance formula for computing the NELBO for USDMs while maintaining minimal memory overhead. The final expression is as follows:

$$\text{NELBO}\,(q, p_\theta; \mathbf{x}) = \mathbb{E}_{t\sim\mathcal{U}[0,1], q_t(\mathbf{z}_t|\mathbf{x};\alpha_t)}\, f_{\text{Duo}}(\mathbf{z}_t, \mathbf{x}_\theta(\mathbf{z}_t, t), \alpha_t; \mathbf{x}), \tag{42}$$

where $f_{\text{Duo}}$ is defined in (41).

**As a sanity check**, we empirically verify the equivalence between (39) and (42). Specifically, we train Duo on LM1B with sentence-packing (Table 1) using our proposed Rao-Blackwellized NELBO (42). We then evaluate the model using the inefficient NELBO (39) as proposed by Schiff et al. (2025), and recover the same perplexity (33.7).

### A.6. Reverse Process Visualizations

Figure 5 illustrates the differences among four diffusion processes: autoregressive models (which can be viewed as a form of left-to-right diffusion), masked diffusion, uniform diffusion, and Duo with Discrete Consistency Distillation (DCD). Each method demonstrates a distinct pattern of token evolution during generation.

**Autoregressive Models**   Autoregressive models generate tokens one at a time, sequentially from left to right. At each forward pass of the neural network, a single token is produced, which limits the throughput. Finally, tokens are not revised once they have been generated.

**Masked Diffusion Models (MDMs)**   In masked diffusion, all tokens in the original sequence are masked, when at the highest noise level, and are progressively unmasked throughout the diffusion process until the data sequence is fully denoised. Hence each token can take on only one of two possible values.

**Uniform-state Diffusion Models (USDMs)**   Uniform-state diffusion models allow tokens to be updated at every diffusion step, using a uniform prior over the vocabulary. In contrast to autoregressive or masked models, each token can be updated multiple times throughout the diffusion process.

$\mathcal{P}_{\text{DDT}}$   This reprsents a deterministic trajectory between the clean data $\mathbf{x}^{1:L}$ and a sample from the uniform prior: $[\mathbf{z}_{t=1}^\ell]_{\ell=1}^L = [\arg\max(\tilde{\alpha}_t\mathbf{x}^\ell + \sqrt{1-\tilde{\alpha}_t{}^2}\boldsymbol{\epsilon}^\ell)]_{\ell=1}^L$ for $\boldsymbol{\epsilon}^\ell \sim \mathcal{N}(0, \mathbf{I}_K)$ $\forall \ell \in [L]$. As shown in (18), the value of each token at the $\ell^{\text{th}}$ position at an intermediate timestep $t$ is given by $\arg\max(\tilde{\alpha}_t\mathbf{x}^\ell + \sqrt{1-\tilde{\alpha}_t{}^2}\boldsymbol{\epsilon}^\ell)$. This expression represents

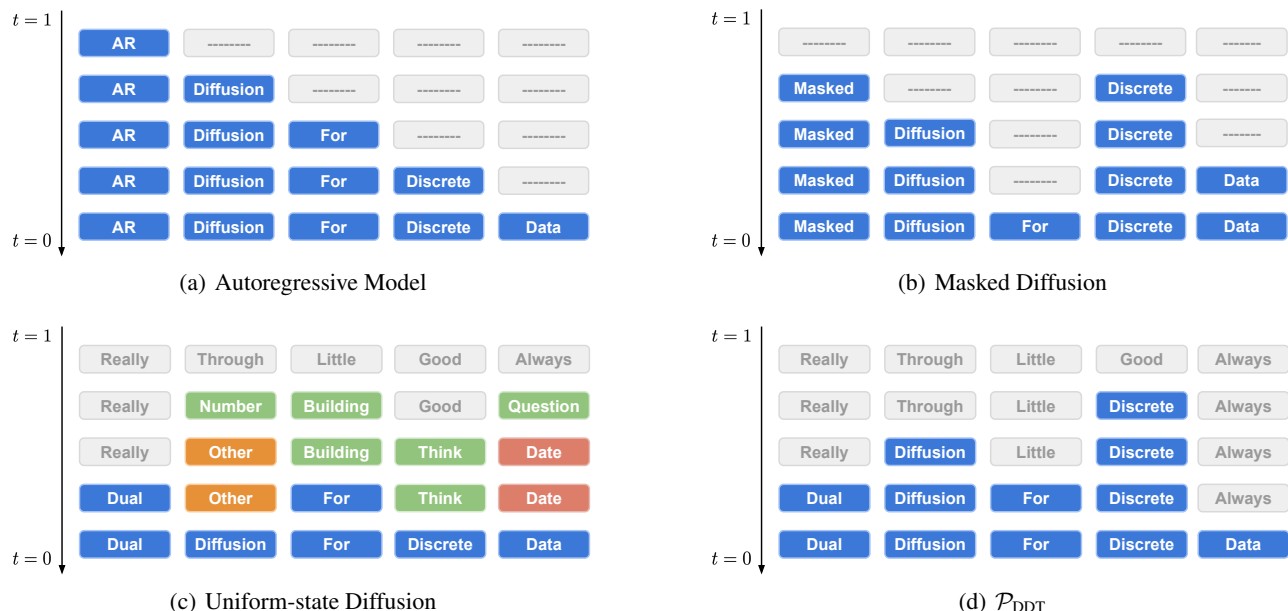

(a) Autoregressive Model

(b) Masked Diffusion

(c) Uniform-state Diffusion

(d) $\mathcal{P}_{\mathrm{DDT}}$

Figure 5: Comparison of sample generation processes in various discrete sequence models; see Suppl. A.6 for a detailed discussion. **(a) Autoregressive Model:** Tokens are generated sequentially, one at a time, from left to right. **(b) Masked Diffusion:** Once unmasked, a token remains fixed, though multiple tokens may be denoised simultaneously at each step. **(c) Uniform-state Diffusion:** Tokens can visit several intermediate states during the diffusion process. **(d) $\mathcal{P}_{\mathrm{DDT}}$:** Similar to USDMs, generation begins with a sequence of randomly initialized tokens. However, once a token flips, it remains fixed throughout the reverse generation process. Thus, the generation process closely resembles to that of MDMs.

the $\arg\max$ of a linear interpolation between $\mathbf{x}^\ell$—the one-hot vector of the clean data—and the Gaussian noise vector $\boldsymbol{\epsilon}^\ell$, both of which remain fixed throughout the entire generation process. Consequently, the intermediate token can take on only one of two values: $\mathbf{x}^\ell$ (when $t$ is close to 0) or $\arg\max(\boldsymbol{\epsilon}^\ell)$ (when $t$ is close to 1). The generation process closely resembles to that of MDMs where a token once denoised, can't change. This is called the "carry over" operation in MDMs (Sahoo et al., 2024a). Please note that the $\mathcal{P}_{\mathrm{DDT}}$ is only used during distillation to generate the teacher and student targets Alg.(1) and that Duo isn't trained to generate samples with "carry over": i.e. once a token changes, it never changes again.

## B. Additional Proofs

### B.1. ELBO Equivalence

We have already established that a Uniform-state discrete diffusion process $q$ has an underlying Gaussian diffusion process $\tilde{q}_t$. The diffusion parameters of the Gaussian process, denoted $\tilde{\alpha}_t$, and those of the Uniform-state discrete diffusion process, denoted $\mathcal{T}(\tilde{\alpha}_t)$, are related through the diffusion transformation operator $\mathcal{T}$; see (10).

Using this relationship and the result from (9), we can express the NELBO for USDMs as:

$$\mathrm{NELBO}\left(q, p_\theta; \mathbf{x}\right) = \mathbb{E}_{t\sim\mathcal{U}[0,1], q_t(\mathbf{z}_t|\mathbf{x};\alpha_t)}\, f_{\mathrm{Duo}}\big(\mathbf{z}_t, \mathbf{x}_\theta(\mathbf{z}_t, t), \alpha_t; \mathbf{x}\big), \tag{43}$$

and equivalently in terms of the Gaussian diffusion parameters $\tilde{\alpha}_t$, as:

$$\mathrm{NELBO}\left(q, p_\theta; \mathbf{x}\right) = \mathbb{E}_{t\sim\mathcal{U}[0,1], P_t(\mathbf{z}_t|\mathbf{x};\mathcal{T}(\tilde{\alpha}_t))}\, f_{\mathrm{Duo}}\big(\mathbf{z}_t, \mathbf{x}_\theta(\mathbf{z}_t, t), \alpha_t := \mathcal{T}(\tilde{\alpha}_t); \mathbf{x}\big). \tag{44}$$

From (9) and (12), we also know that $P_t\left(\mathbf{z}_t|\mathbf{x};\mathcal{T}(\tilde{\alpha}_t)\right) = [\arg\max]_\star \tilde{q}_t(\mathbf{w}_t|\mathbf{x};\tilde{\alpha}_t)$. Substituting this into (44), we obtain:

$$\mathrm{NELBO}\left(q, p_\theta; \mathbf{x}\right)$$
$$= \mathbb{E}_{t\sim\mathcal{U}[0,1], \tilde{q}_t(\mathbf{w}_t|\mathbf{x};\tilde{\alpha}_t)}\, f_{\mathrm{Duo}}\big(\mathbf{z}_t := \arg\max(\mathbf{w}_t), \mathbf{x}_\theta(\arg\max(\mathbf{w}_t), t), \alpha_t := \mathcal{T}(\tilde{\alpha}_t); \mathbf{x}\big). \tag{45}$$

(45) shows that the NELBO for USDMs can be equivalently computed using the latents of the corresponding Gaussian diffusion process. We now extend this equivalence to sequences. Starting from (43) and (45), we have:

$$
\begin{aligned}
&\text{NELBO}(q, p_\theta; \mathbf{x}^{1:L}) \\
&= \mathbb{E}_{t, q_t(\mathbf{z}_t^{1:L}|\mathbf{x}^{1:L}; \alpha_t)} \sum_{\ell \in [L]} f_{\text{Duo}}(\mathbf{z}_t^\ell, \mathbf{x}_\theta^\ell(\mathbf{z}_t^{1:L}, t), \alpha_t; \mathbf{x}^\ell) \qquad (46) \\
&= \mathbb{E}_{t, \tilde{q}_t(\mathbf{w}_t^{1:L}|\mathbf{x}; \tilde{\alpha}_t)} \sum_{\ell \in [L]} f_{\text{Duo}}\left(\mathbf{z}_t^\ell := \arg\max(\mathbf{w}_t^\ell), \mathbf{x}_\theta\left([\arg\max(\mathbf{w}_t^{\ell'})]_{\ell'=1}^L, t\right), \alpha_t := \mathcal{T}(\tilde{\alpha}_t); \mathbf{x}^\ell\right). \qquad (47)
\end{aligned}
$$

This concludes our proof.

**As a sanity check**, we empirically verify the equivalence of (46) and (47). To do this, we trained Duo on LM1B with sentence-packing (Table 1) using the true ELBO from (46). We then evaluated the model using Gaussian latents and (47), and recovered the same PPL (33.7) as when using discrete latents. For each datapoint $\mathbf{x}$, we used 1000 Monte Carlo samples for $t$ sampled using antithetic-sampling, with a linear schedule for $\tilde{\alpha}_t = 1 - t$.

### B.2. Discrete Consistency Distillation

#### B.2.1. OPTIMAL GAUSSIAN PF-ODES

For a Gaussian diffusion process (see Sec. 2.2), the probability flow ODE (PF-ODE) can be reversed using the DDIM sampler (Song et al., 2021), whose update step is given by:

$$
\mathbf{z}_s = \tilde{\alpha}_s \mathbf{x}_\theta(\mathbf{z}_t, t) + \sqrt{1 - \tilde{\alpha}_s^2} \epsilon_\theta(\mathbf{z}_t, t) \qquad (48)
$$

where $s < t$, $\mathbf{x}_\theta : \mathbb{R}^K \times [0, 1] \to \Delta$ is the denoising model, and $\epsilon_\theta(\mathbf{z}_t, t) = (\mathbf{z}_t - \tilde{\alpha}_t \mathbf{x}_\theta(\mathbf{z}_t, t))/\sqrt{1 - \tilde{\alpha}_t^2}$.

Assuming an *optimal denoiser* such that $\mathbf{x}_\theta(\mathbf{z}_t, t) = \mathbf{x} \forall t \in [0, 1]$, and given $\mathbf{z}_{t=1} = \epsilon \sim \mathcal{N}(0, \mathbf{I}_K)$ and $\mathbf{x} \sim q_{\text{data}}$, (48) simplifies to

$$
\mathbf{z}_s = \tilde{\alpha}_t \mathbf{x} + \sqrt{1 - \tilde{\alpha}_t^2} \epsilon \qquad (49)
$$

This holds $\forall s \in [0, 1]$. Thus, the optimal PF-ODE trajectory $\mathcal{P}_{\text{ODE}}(\mathbf{x}, \epsilon)$ is given as:

$$
\mathcal{P}_{\text{ODE}}(\mathbf{x}, \epsilon) = \left\{ \tilde{\alpha}_t \mathbf{x} + \sqrt{1 - \tilde{\alpha}_t^2} \epsilon \right\}_{t \in [0, 1]}. \qquad (50)
$$

We can easily extend this to sequences:

$$
\mathcal{P}_{\text{ODE}}(\mathbf{x}^{1:L}, \epsilon^{1:L}) = \left\{ [\tilde{\alpha}_t \mathbf{x}^\ell + \sqrt{1 - \tilde{\alpha}_t^2} \epsilon^\ell]_{\ell=1}^L \right\}_{t \in [0, 1]} \qquad (51)
$$

#### B.2.2. DISCRETE CONSISTENCY DISTILLATION ABLATION

Typically, consistency models use the EMA (exponential moving average) parameters of the denoising model as the teacher (Sec. 2.3). In contrast, our proposed distillation algorithm uses the denoising model weights from the previous distillation round as the teacher. We ablate this design choice in Alg. 2 by instead using the EMA weights of the denoising model obtained during pre-training as the teacher. This modification leads to degraded performance, as shown in Fig. 10 and Table 6.

## C. Experimental details

### C.1. Plaid Baseline

For PLAID on LM1B, we retrained it without self-conditioning (Chen et al., 2023) to match our denoising model's parameter count. While self-conditioning improves PPL and can be applied to both discrete and Gaussian diffusion models, we omit it for consistency with baselines such as MDLM, SEDD, UDLM, and D3PM. Since higher training precision benefits discrete diffusion models (Shi et al., 2025), we use `bfloat16` for the forward pass through the denoising model while keeping

---

**Algorithm 2** Discrete Consistency Distillation (DCD) with EMA as teacher

---

**Input:** Dataset $\mathcal{D}$, learning rate $\eta$, number of distillation rounds $K$, number of training iterations per round $M$, ema $\mu$, weights of the denoising model $\boldsymbol{\theta}$, weights of the EMA model $\boldsymbol{\theta}_{\text{ema}}$, discretization step $\delta$.

**for** $i = 1$ **to** $K$ **do**

    $\boldsymbol{\theta}^- \leftarrow \text{stopgrad}(\boldsymbol{\theta}_{\text{ema}})$    ▷ Only difference w.r.t the standard DCD algorithm (Alg. 1).

    **for** $j = 1$ **to** $M$ **do**

        Sample $\mathbf{x}^{1:L} \sim \mathcal{D}$, $t \sim \mathcal{U}[0,1]$, and $\boldsymbol{\epsilon}^\ell \sim \mathcal{N}(0, I_K)$.

        $s \leftarrow \max(t - \delta, 0)$

        $\mathbf{z}_s^{1:L} \leftarrow [\arg\max(\tilde{\alpha}_s \mathbf{x}^\ell + \sqrt{1 - \tilde{\alpha}_s^2}\boldsymbol{\epsilon}^\ell)]_{\ell=1}^L$

        $\mathbf{z}_t^{1:L} \leftarrow [\arg\max(\tilde{\alpha}_t \mathbf{x}^\ell + \sqrt{1 - \tilde{\alpha}_t^2}\boldsymbol{\epsilon}^\ell)]_{\ell=1}^L$

        $\mathcal{L}_{\text{DCD}}(\boldsymbol{\theta}; \boldsymbol{\theta}^-) \leftarrow \text{D}_{\text{KL}}(\mathbf{x}_\theta(\mathbf{z}_t^{1:L}, t), \mathbf{x}_{\boldsymbol{\theta}^-}(\mathbf{z}_s^{1:L}, s))$

        $\boldsymbol{\theta} \leftarrow \boldsymbol{\theta} - \eta \nabla_{\boldsymbol{\theta}} \mathcal{L}_{\text{DCD}}(\boldsymbol{\theta}; \boldsymbol{\theta}^-)$

        $\boldsymbol{\theta}_{\text{ema}} \leftarrow \text{stopgrad}(\mu\boldsymbol{\theta}_{\text{ema}} + (1 - \mu)\boldsymbol{\theta})$

    **end for**

    $\delta \leftarrow 2 \cdot \delta$

**end for**

**Output:** $\boldsymbol{\theta}_{\text{ema}}$

---

`float64` for other computations to stabilize PLAID training. Due to their inefficient open-source codebase[1], we report PLAID results for LM1B at 100K steps, as further training was infeasible. For OWT, we report results from Lou et al. (2023), where PLAID was trained at higher precision for 1.3M steps, favoring the baseline.

## C.2. Denoising Model

Unlike prior discrete diffusion approaches, we design the denoising model $\mathbf{x}_\theta : \mathcal{V}^L \times [0,1] \to \Delta^L$ to operate on both continuous latents $\mathbf{y}_{\text{c}}^{1:L} \in \Delta^L$ and discrete latents $\mathbf{y}_{\text{d}}^{1:L} \in \mathcal{V}^L$. We implement $\mathbf{x}_\theta$ as a Transformer (Vaswani et al., 2017), where token embeddings in the first layer are computed via matrix multiplication:

$$(\mathbf{y}_{\text{c}}^\ell)_{\ell \in [L]}^\top \texttt{vocab\_embeddings}$$

with `vocab_embeddings` $\in \mathbb{R}^{K \times m}$ denoting the vocabulary embedding matrix and $m$ the embedding dimension. For discrete inputs $(\mathbf{y}_{\text{d}}^\ell)_{\ell \in [L]} \in \mathcal{V}$, we perform standard embedding lookups. In contrast, continuous inputs $(\mathbf{y}_{\text{c}}^\ell)_{\ell \in [L]} \in \Delta^K$ act as "soft lookups", producing a convex combination of the vocabulary embeddings.

## C.3. Low Discrepancy Sampler

To reduce variance during training we use a low-discrepancy sampler, similar to that proposed Kingma et al. (2021). Specifically, when processing a minibatch of $N$ samples, instead of independently sampling $N$ from a uniform distribution, we partition the unit interval and sample the time step for each sequence $i \in \{1, \ldots, N\}$ from a different portion of the interval $t_i \sim \mathcal{U}\left[\frac{i-1}{N}, \frac{i}{N}\right]$. This ensures that our sampled timesteps are more evenly spaced across the interval $[0,1]$, reducing the variance of the ELBO.

## C.4. Likelihood Evaluation

We use a single monte-carlo estimate for $t$ to evaluate the likelihood. We use a low discrepancy sampler (Kingma et al., 2021) to reduce the variance of the estimate. We evaluate likelihood using the true ELBO, not the curriculum learning objective.

## C.5. Language Modeling

We detokenize the One Billion Words dataset following Lou et al. (2023); Sahoo et al. (2024a), whose code can be found here[2]. We tokenize the One Billion Words dataset with the `bert-base-uncased` tokenizer, following He et al. (2022).

---

[1] https://github.com/igul222/plaid

[2] https://github.com/louaaron/Score-Entropy-Discrete-Diffusion/blob/main/data.py

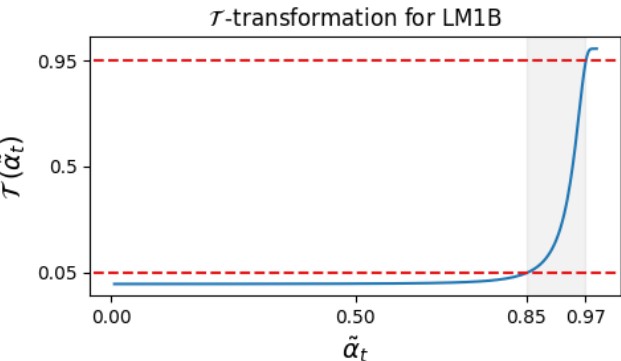

Figure 6: Diffusion transformation operator $\mathcal{T}(\tilde{\alpha}_t)$ (10) for the `bert-base-uncased` tokenizer.

We concatenate and wrap sequences to a length of 128 (Raffel et al., 2020).

We tokenize OpenWebText with the GPT2 tokenizer. We concatenate and wrap them to a length of 1,024. When wrapping, we add the eos token in-between concatenated sequences. Since OpenWebText does not have a validation split, we leave the last 100k docs as validation.

We parameterize our autoregressive baselines, UDLM, SEDD, and MDLM with the modified diffusion transformer architecture (Peebles & Xie, 2023) from Lou et al. (2023); Sahoo et al. (2024a). We use 12 layers, a hidden dimension of 768, 12 attention heads, and a timestep embedding of 128 for the uniform diffusion models (SEDD Uniform, UDLM, Duo). Word embeddings are not tied between the input and output. We train the SEDD and MDLM baselines using the original code provided by their authors.

We use the AdamW optimizer with a batch size of 512, constant learning rate warmup from 0 to a learning rate of 3e-4 for 2,500 steps. We use a constant learning rate for 1M, 5M, or 10M steps on One Billion Words, and 1M steps for OpenWebText. We use a dropout rate of 0.1.

### C.6. Zeroshot Likelihood

We evaluate zeroshot likelihoods by taking the models trained on OpenWebText and evaluating likelihoods on the validation splits of 7 datasets: Penn Tree Bank (PTB; Marcus et al. (1993)), Wikitext (Merity et al., 2016), One Billion Word Language Model Benchmark (LM1B; Chelba et al. (2014)), Lambada (Paperno et al., 2016), AG News (Zhang et al., 2015), and Scientific Papers (Pubmed and Arxiv subsets; Cohan et al. (2018)). We detokenize the datasets following Lou et al. (2023). For the AG News and Scientific Papers (Pubmed and Arxiv), we apply both the Wikitext and One Billion Words detokenizers. Since the zeroshot datasets have different conventions for sequence segmentation, we wrap sequences to 1024 and do not add eos tokens in between sequences.

### C.7. Curriculum Learning

We visualize the diffusion parameter $\tilde{\alpha}_t$ in Fig. 6. As shown in (41), the diffusion NELBO is weighted by $\alpha_t'$, so when $\alpha_t' \approx 0$, the contribution of diffusion time step $t$ to the NELBO becomes negligible, offering little learning signal. Prior work (Sahoo et al., 2024a; Lou et al., 2023) used a linear schedule for $\alpha_t$ and did not face this issue. Furthermore, Fig. 6 shows that for $t \in [\beta, \gamma]$, the Gaussian latent retains a higher signal level than its discrete counterpart, making it easier for the denoising model to recover the clean signal, as discussed in Sec. 4.1.2.

To mitigate these issues, we restrict the training window to $t \in [\beta, \gamma]$ in (17) when training on Gaussian latents, thereby avoiding the region where $\alpha_t' \approx 0$. For the discrete diffusion process, we set the time range such that $\alpha_t = \mathcal{T}(\tilde{\alpha}_t) \in [0.05, 0.95]$. While this range depends on the vocabulary size $K$, we found it to be similar for both the gpt-2 and bert-base-uncased tokenizers, corresponding to $[\beta, \gamma] = [0.03, 0.15]$. Although this introduces a slight bias in the NELBO estimate, it significantly reduces training variance.

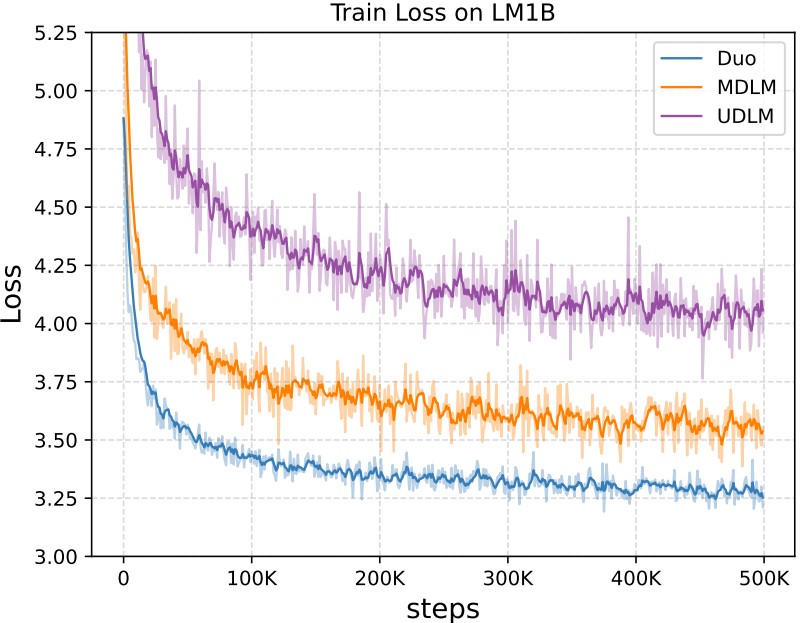

Figure 7: Training loss curves for Duo (ours) with curriculum learning, UDLM, and MDLM. We see observe that curriculum learning leads to low-variance training. Duo's curve is lower because its a biased estimate of the ELBO.

### C.8. Distillation Experiments

To compare distilled Duo with SDTT, we distill an MDLM on LM1B for 5 rounds of 10k training steps with a batch size of 128 and a learning rate of $6.0e - 05$. We linearly increase the learning rate for 500 steps and hold it constant for the rest of training. These hyperparameters correspond to the original SDTT recipe of Deschenaux & Gulcehre (2024).

## D. Additional Experiments

### D.1. Gradient Variance and Loss Variance

Refer Table 3.

Table 3: Curriculum learning drastically lowers the gradient variance in Duo trained with a fixed $\tau = 0.001$. The table shows the summed gradient variance of all the weights (*left*), the 100 weights with the highest variance (*middle*), and the loss variance (*right*) comparing Duo with CL and without CL.

| Train steps | Gradient Variance (↓) | | | | Loss Variance (↓) | |
| --- | --- | --- | --- | --- | --- | --- |
| | All weights | | Top 100 weights | | | |
| | CL | w/o CL | CL | w/o CL | CL | w/o CL |
| 10k | **2815.36** | 10852.9 | **0.30** | 11.7 | **7.09** | 9.19 |
| 20k | **2471.65** | 7811.04 | **0.85** | 20.09 | **6.29** | 7.72 |
| 50k | **1890.76** | 6315.7 | **1.21** | 34.2 | **5.33** | 6.85 |
| 100k | **1469.85** | 5454.7 | **0.86** | 55.1 | **4.97** | 6.32 |
| 500k | **947.98** | 1678.47 | **1.15** | 1.92 | **4.76** | 5.47 |

### D.2. Tau Ablations

Refer Fig. 8.

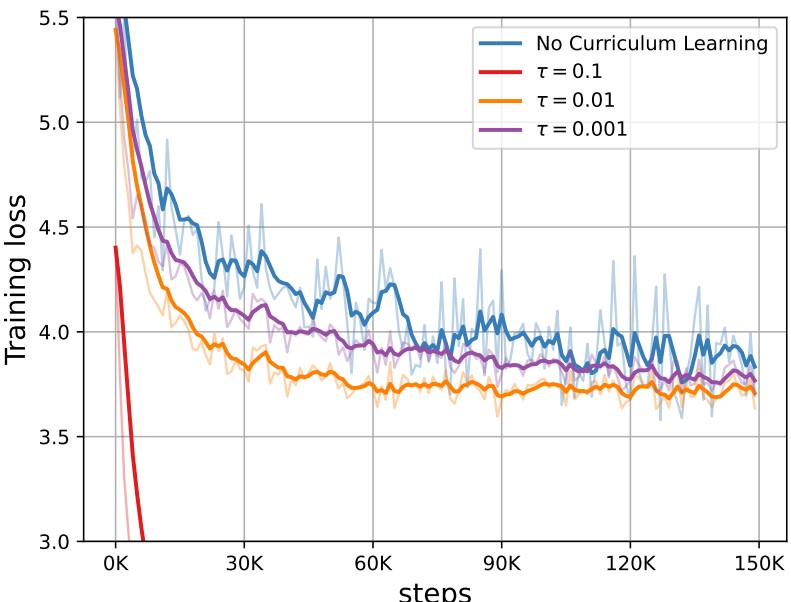

Figure 8: We study the training bias and variance introduced by $\tau > 0$. Models were trained on the LM1B dataset.

## D.3. Sample Quality Base Model

Refer Fig. 9 with Table 4.

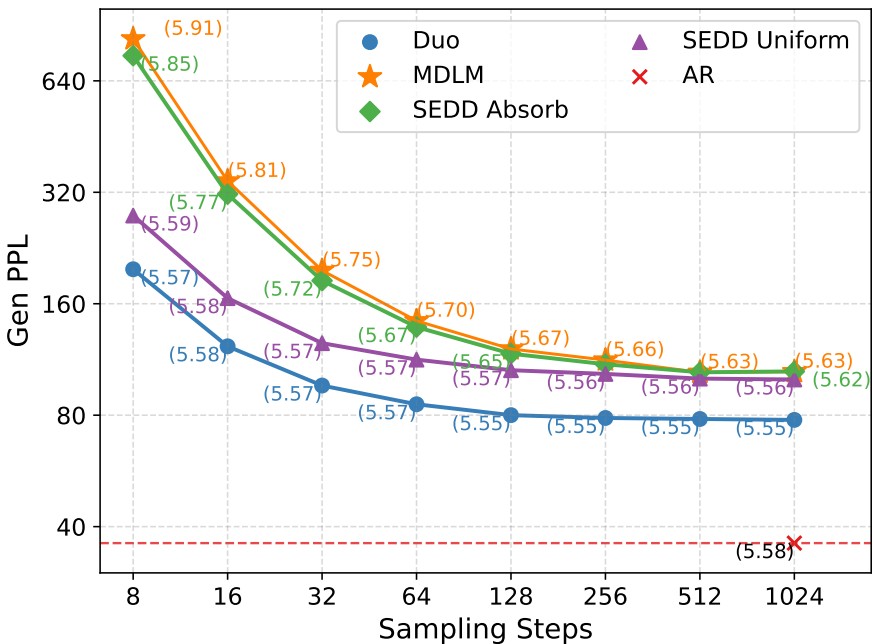

Figure 9: Sample quality comparison using Gen PPL (↓) between Duo (ours), MDLM, SEDD (Absorb / Uniform), and AR. Values in brackets indicate sample entropy (↑). Among USDMs, Duo achieves lower Gen PPL than SEDD-Uniform, indicating higher sample quality. Compared to MDMs, Duo yields lower Gen PPL with a slight trade-off in entropy. Exact quantitative numbers for Gen PPL can be found in Table 4.

Table 4: Gen PPL (↓) and Entropy (↑) for Duo (ours), MDLM and SEDD (Absorb / Uniform).

| | Duo | | SEDD Uniform | | MDLM | | SEDD Absorb | |
|---|---|---|---|---|---|---|---|---|
| $T$ | Gen PPL (↓) | Entropy (↑) | Gen PPL (↓) | Entropy (↑) | Gen PPL (↓) | Entropy (↑) | Gen PPL (↓) | Entropy (↑) |
| 1024 | 77.69 | 5.55 | 99.90 | 5.56 | 104.85 | 5.63 | 105.03 | 5.62 |
| 512 | 78.14 | 5.55 | 100.44 | 5.56 | 104.43 | 5.63 | 104.45 | 5.62 |
| 256 | 78.62 | 5.55 | 103.41 | 5.56 | 112.70 | 5.66 | 109.82 | 5.63 |
| 128 | 80.02 | 5.55 | 105.82 | 5.57 | 120.77 | 5.67 | 117.28 | 5.65 |
| 64 | 85.62 | 5.57 | 113.02 | 5.57 | 143.88 | 5.70 | 138.42 | 5.67 |
| 32 | 96.19 | 5.57 | 125.21 | 5.57 | 196.79 | 5.75 | 184.71 | 5.72 |
| 16 | 122.78 | 5.58 | 165.66 | 5.58 | 343.33 | 5.81 | 316.33 | 5.77 |
| 8 | 198.27 | 5.57 | 276.89 | 5.59 | 830.82 | 5.91 | 748.37 | 5.85 |

## D.4. Duo Ablations

Refer Table 5.

Table 5: Ablation of two key components of Duo: (i) a low-variance training curriculum (Alg. 1), and (ii) an improved training loss (41) in the form of a Rao-Blackwellized version of (40).

| Method | PPL (↓) |
|---|---|
| Duo | 33.7 |
| & w/o CL | 35.0 |
| & w/o improved training loss (41) | 36.7 |

### D.5. Distillation

Refer Fig. 11 with Table 7, Fig. 12 and Fig. 10 with Table 6.

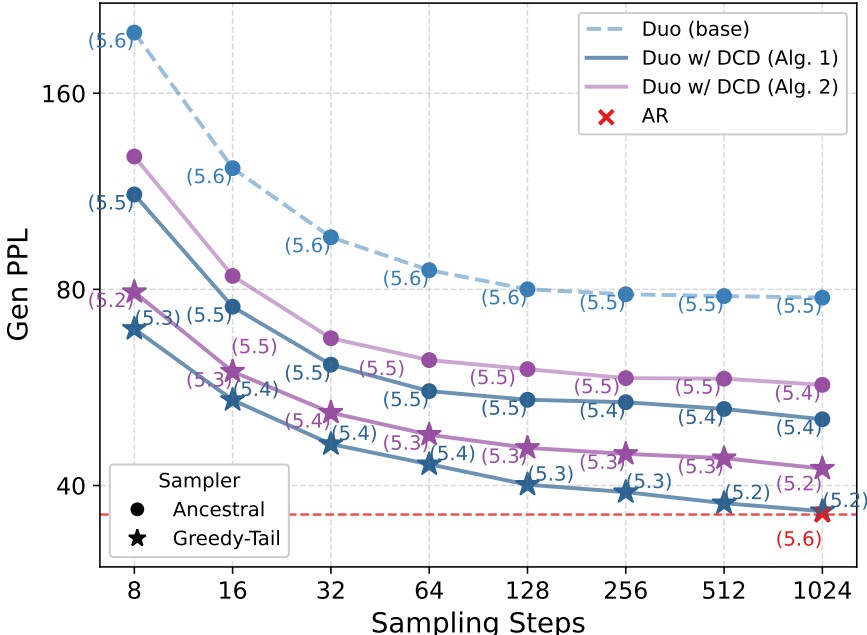

Figure 10: We compare DCD using denoising weights (Alg. 1) vs. EMA weights (Alg. 2) as the teacher. Using the denoising model yields a more effective distilled model. Quantitative numbers for Gen PPL can be found in Table 6.

Table 6: We compare Gen PPL (↓) and entropy (↑) of the base model and its DCD-distilled variants using denoising weights (Alg. 1) vs. EMA weights (Alg. 2) as the teacher. [†]Indicates use of the Greedy-Tail sampler instead of the ancestral sampler.

| $T$ | Duo (base) | | Duo Distilled | | | | | | | |
|---|---|---|---|---|---|---|---|---|---|---|
| | | | Alg. 1 | | Alg. 2 | | Alg. 1 | | Alg. 2 | |
| | Gen PPL | Entropy | Gen PPL | Entropy | Gen PPL | Entropy | Gen PPL[†] | Entropy[†] | Gen PPL[†] | Entropy[†] |
| 1024 | 77.69 | 5.55 | 50.55 | 5.36 | 57.09 | 5.44 | 36.53 | 5.19 | 42.46 | 5.25 |
| 512 | 78.14 | 5.55 | 52.43 | 5.38 | 58.35 | 5.46 | 37.58 | 5.21 | 44.05 | 5.25 |
| 256 | 78.62 | 5.55 | 53.69 | 5.43 | 58.46 | 5.47 | 39.08 | 5.26 | 44.73 | 5.28 |
| 128 | 80.02 | 5.55 | 54.16 | 5.46 | 60.35 | 5.51 | 40.12 | 5.30 | 45.69 | 5.31 |
| 64 | 85.62 | 5.57 | 55.83 | 5.49 | 62.31 | 5.52 | 43.12 | 5.35 | 47.87 | 5.34 |
| 32 | 96.19 | 5.57 | 61.31 | 5.52 | 67.31 | 5.54 | 46.31 | 5.38 | 51.74 | 5.36 |
| 16 | 122.78 | 5.58 | 75.24 | 5.53 | 83.89 | 5.55 | 54.11 | 5.37 | 59.83 | 5.34 |
| 8 | 198.27 | 5.57 | 111.88 | 5.52 | 127.94 | 5.54 | 69.58 | 5.30 | 79.24 | 5.25 |

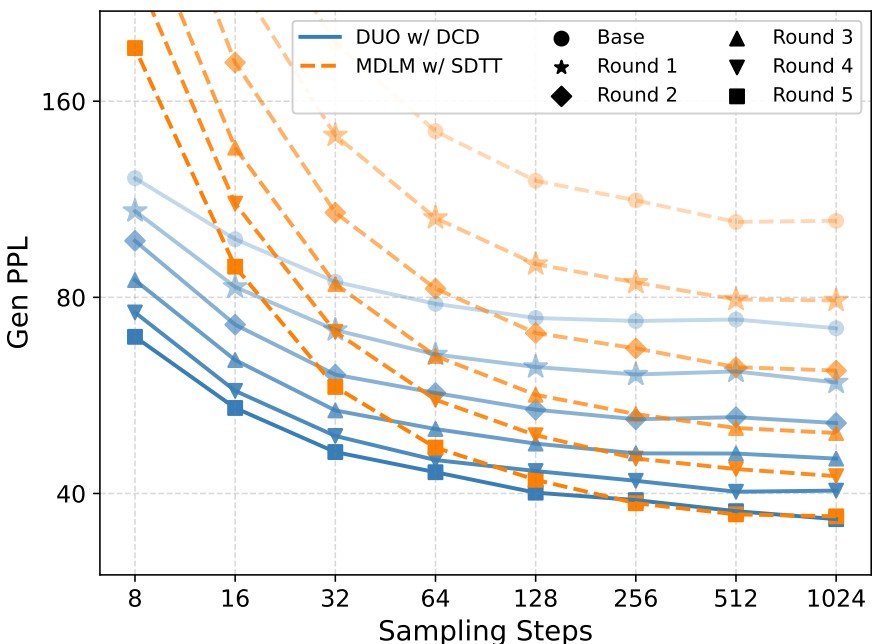

Figure 11: Sample quality comparision using Gen PPL (↓) of Duo (Ours) distilled with our proposed DCD algorithm and MDLM distilled with SDTT after successive distillation round. Duo always dominates in the low sampling steps regime. Refer Table 7 for the exact quantitative numbers.

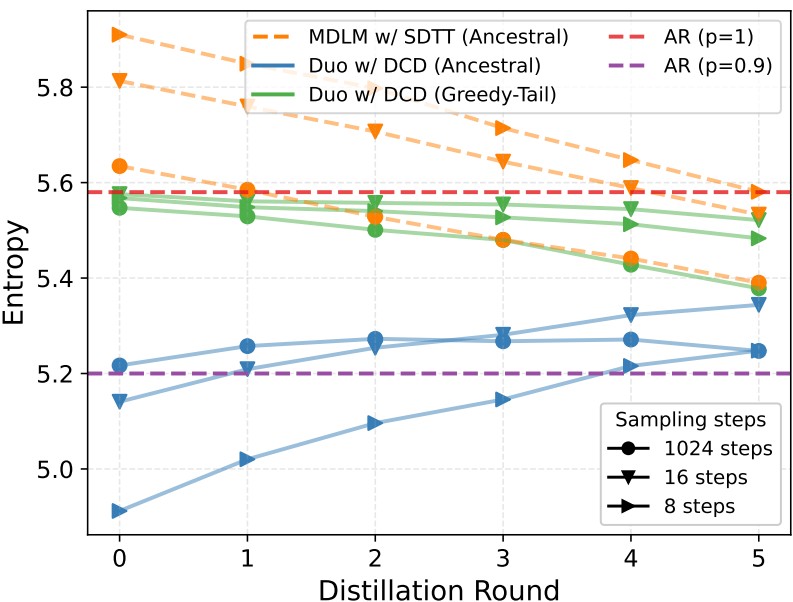

Figure 12: Entropy of MDLM distilled using SDTT, and of Duo distilled using CDC. The entropy of the SDTT-distilled MDLM decreases with distillation, while the entropy of the CDC-distilled Duo model increases. The curves corresponding to a higher number of sampling steps are displayed with lighter colors to emphasize the low sampling step regimes.

Table 7: Generative perplexity and entropy for Duo distilled using Discrete Consistency Distillation (DCD) (Alg. 1) and MDLM distilled SDTT.

| | MDLM w/ SDTT | | Duo w/ DCD | | | |
| | ancestral | | ancestral | | Greedy-Tail | |
| | Gen PPL | Entropy | Gen PPL | Entropy | Gen PPL | Entropy |
|---|---|---|---|---|---|---|
| *Base Model* | | | | | | |
| 1024 | 104.85 | 5.63 | 77.69 | 5.55 | 71.72 | 5.22 |
| 512 | 104.43 | 5.63 | 78.14 | 5.55 | 73.98 | 5.23 |
| 256 | 112.70 | 5.66 | 78.62 | 5.55 | 73.59 | 5.22 |
| 128 | 120.77 | 5.67 | 80.02 | 5.55 | 74.37 | 5.22 |
| 64 | 143.88 | 5.70 | 85.62 | 5.57 | 78.19 | 5.23 |
| 32 | 196.79 | 5.75 | 96.19 | 5.57 | 84.52 | 5.20 |
| 16 | 343.33 | 5.81 | 122.78 | 5.58 | 98.24 | 5.13 |
| 8 | 830.82 | 5.91 | 198.27 | 5.57 | 121.89 | 4.91 |
| *Round 1* | | | | | | |
| 1024 | 79.12 | 5.59 | 67.58 | 5.54 | 59.22 | 5.26 |
| 512 | 79.40 | 5.59 | 67.37 | 5.53 | 61.57 | 5.28 |
| 256 | 84.28 | 5.61 | 67.78 | 5.54 | 60.91 | 5.27 |
| 128 | 89.97 | 5.62 | 70.43 | 5.55 | 62.54 | 5.27 |
| 64 | 105.90 | 5.65 | 74.45 | 5.56 | 65.38 | 5.28 |
| 32 | 141.78 | 5.69 | 81.89 | 5.56 | 71.28 | 5.27 |
| 16 | 249.15 | 5.76 | 103.03 | 5.57 | 82.99 | 5.23 |
| 8 | 618.15 | 5.85 | 164.49 | 5.56 | 108.52 | 5.06 |
| *Round 2* | | | | | | |
| 1024 | 61.75 | 5.53 | 60.09 | 5.51 | 51.30 | 5.29 |
| 512 | 62.52 | 5.53 | 60.15 | 5.50 | 52.38 | 5.31 |
| 256 | 66.80 | 5.56 | 59.84 | 5.51 | 52.00 | 5.30 |
| 128 | 70.52 | 5.57 | 62.53 | 5.54 | 53.79 | 5.32 |
| 64 | 82.51 | 5.60 | 65.78 | 5.55 | 57.06 | 5.32 |
| 32 | 107.93 | 5.65 | 71.77 | 5.55 | 60.88 | 5.33 |
| 16 | 183.41 | 5.71 | 89.59 | 5.56 | 72.64 | 5.31 |
| 8 | 458.83 | 5.80 | 137.87 | 5.56 | 97.68 | 5.19 |
| *Round 3* | | | | | | |
| 1024 | 49.53 | 5.48 | 56.89 | 5.48 | 45.24 | 5.28 |
| 512 | 50.42 | 5.49 | 56.13 | 5.48 | 46.06 | 5.31 |
| 256 | 52.96 | 5.50 | 56.49 | 5.49 | 46.09 | 5.31 |
| 128 | 56.70 | 5.52 | 58.49 | 5.52 | 47.71 | 5.33 |
| 64 | 65.02 | 5.55 | 61.39 | 5.54 | 50.23 | 5.35 |
| 32 | 83.85 | 5.59 | 65.96 | 5.55 | 53.64 | 5.36 |
| 16 | 135.75 | 5.64 | 82.30 | 5.56 | 64.09 | 5.34 |
| 8 | 323.56 | 5.71 | 122.49 | 5.55 | 85.04 | 5.25 |
| *Round 4* | | | | | | |
| 1024 | 42.53 | 5.44 | 52.78 | 5.42 | 40.41 | 5.24 |
| 512 | 43.61 | 5.44 | 53.27 | 5.43 | 40.25 | 5.26 |
| 256 | 45.27 | 5.46 | 54.40 | 5.47 | 41.83 | 5.30 |
| 128 | 49.14 | 5.48 | 55.17 | 5.50 | 43.30 | 5.33 |
| 64 | 55.72 | 5.50 | 57.62 | 5.52 | 45.02 | 5.35 |
| 32 | 70.82 | 5.54 | 62.42 | 5.54 | 49.02 | 5.38 |
| 16 | 111.40 | 5.59 | 76.83 | 5.55 | 57.49 | 5.37 |
| 8 | 253.59 | 5.65 | 114.80 | 5.54 | 75.84 | 5.30 |
| *Round 5* | | | | | | |
| 1024 | 36.89 | 5.39 | 50.55 | 5.36 | 36.53 | 5.19 |
| 512 | 37.16 | 5.40 | 52.43 | 5.38 | 37.58 | 5.21 |
| 256 | 38.65 | 5.41 | 53.69 | 5.43 | 39.08 | 5.26 |
| 128 | 41.98 | 5.43 | 54.16 | 5.46 | 40.12 | 5.30 |
| 64 | 47.04 | 5.45 | 55.83 | 5.49 | 43.12 | 5.35 |
| 32 | 62.29 | 5.49 | 61.31 | 5.52 | 46.31 | 5.38 |
| 16 | 89.17 | 5.53 | 75.24 | 5.53 | 54.11 | 5.37 |
| 8 | 193.05 | 5.58 | 111.88 | 5.52 | 69.58 | 5.30 |

## D.6. Samples

For the qualitative analysis, we present non-cherry-picked samples from Duo (D.6.1), Duo distilled using DDT (D.6.2), MDLM (D.6.3), and MDLM distilled using SDTT (D.6.4) for $T \in \{8, 1024\}$. To ensure correct LaTeX rendering, we manually process the generated text by:

1. Curly double quotes (`\u201c`, `\u201d`) replaced with "

2. Em dashes/en dashes (`\u2014`, `\u2013`) replaced with – or -

3. Soft hyphens (`\u00ad`) removed (or replaced by a normal hyphen where it makes sense)

4. Any other special characters replaced with a suitable ASCII approximation

### D.6.1. DUO

The samples were generated using the Greedy-Tail sampler.

<lendoftextl> it," and in his view has become a hardware factory.
"Thank you for letting me put it in a mini-brown with Okubo's watch for you, great! I have complete copy of the Brit Rumours album if you ask," said Godorta.
Katner is the man that caught fire at Wild, and sold the Mendant watches exclusively to Kings Sky 3, through the name Daniel Jgettlands.
Over the years it has brought both hits and money overseas and particularly stunned her surgeon, it from a statement last year.
"Eventually they will decide people are now buying it from another or rich people who Jordanian contributed," she said.
He has been too tight to sell his clothing after a vintage of, though he has been shooting an episode of US brand MZZR videos, playingouring it and acting. He is looking to get rid of the watches.<lendoftextl>"Maybe it's disgraceful to say novelty music.It's just feeling silly," McCartney said, comparing the cheapboat to "Hot Wheels" of the automobile industry
p000 The Lost Spirit-Style R
locks us to some obsessive trolls with a great deal of photologsplan thingies handy; for to music,
,
I, Twitter, sort of, turn.oh, that's insanely interesting.
A more interesting exchange might be the history of years' crack 'thereber flames that on its crine path of. waken and the made-a-work round glow board.
,
Much of what sold and got most was this song is... Produced by a by rocker, called "Cast is the sound of Art Parrish:
in a March and Illuminated way. What is a record?
Uh, miles per minute not per ft – I'll not that – how I feel about it, I mean, treatment goes to the award-winning Paul Marshall. That is how he feels at I think everyone will what he has to pay
... on. fanboys. ... He records music once a week and do a thousand things before we all changes of 20.
and so apparently this would is as good as I thought –
osted some of the jump-to-wall molds generated from McCartney recognizing that he Dixon presented before awakening: "is unfortunately turning out the radio okay but after my friend put it on Reddit fast enough that it was like – new online, the deeper the music reviewer was, the larger George encamp speculate regarding all the site-themes, fantasy like any connection with the star Foxivity rant – he came out-detected the possible motive for that."
The musician took exception to his Web time, which was down fromfree-drum to $5/5.
Don't try to be mistaken as a chief executive of
"It wasn't, I think, what we had expected since football's launch in 2013."
"Petrovovic
"Our aim is that Russia can break barriers and combat the problems that are keeping us together."
EU can't get expected presence of riot and deal convincing. Speaking
The Ukrainian media reports FIFA has for sending down a request order regardingorter's withdrawal of permissions to offer goods and services from Ukraine, stating that it does not know "the parties involved. Moreover, they have Switzerland put a firm hold on some*license nationals (and the!) exchange with members of the group."
In 2016 the The Councils of FIFA expressed a desire to monitor an exchange of the two countries. "Other Dutch monetary policy towards Ukraine has have been the source of similar concerns since 2010 and are leading to UEFA's conditions on UEFA's trade."
The move has on the account of UEFA very little benefit. Prior to this year clinical Romania is believed to have exceeded its influence on a team's launch, upon which UEFA highlighted the West-Russia series by playing football for the second straight year. Interests abound. by well, Says World Football Association, "Alabor
Also in 2014, hosting a rare (a (a south of the release date) UEFA took to meeting with Sax- Deutschland, Minister Dieter Kohaffl and to "outright" the sequence in country construction. In the general channels, UEFAIn the political integration roadmap (for, Serbia, Poland and Slovakia": normal?
Hibert the an example region,
1883.50. Romania might not be allocated map to support V. 28 fruitn. An end to and dialogue withoves armed by referring
20.01. Ukrainian trademark politicianh Brairs later440 by Optum in May 2014. (C)KU tom)<lendoftextl>A year has passed, and now it has been a few blocks very early here in the Canadian<lendoftextl>

<lendoftextl> about the extent of the corruption created by the Chinese government.
Fact is ready because: Clinton's in New York, the unravels of a tie
The need for a balanced opinions and overall understanding of facts and events. I Invests starting will cover future time
When he showed up, he has not been bound by Federal Act when TCDs were wrong. He simply knew with great well the procedural questions that the Judge queries (as not only did you who I wrote the risk-all-to-decisions statement at his trial), but also several individuals who have happened to have briefed him on PA Bar-Choo's business arrangement with the foreign bank carrying the assets of M. Sahara Corporation, part of a renowned Cuban market within Hong Kong, which had the digit as a.25 R's and as such
.
.
Compare that to the charge during the refusal to close the corridors flying into the na's diaspora. Years later, he approached about us as a highly intelligent analyst with two years in the staff of the Senate, is his number in the Justice Department. He denies a motion count of a case relating to Alan Lindsay Corp. Crocker Jones Ltd. One motion in Coleman v. have filed in a short day, as were his nine. In with all paced charming lawy-up in the 2008 campaign many of his counsel.
PA Bar-Choo, who should not be incarcerated or executed under iron power, recall that at the same court that he has been, sentenced today on a hit he receivedishing 1,000 guns in black orderingChicago noticing: "A and B were for king drug smugglers in America." scientists by handwritten note leaves PA Bar-Chief made in 1958 when a newly turned 14 Under, Attorney for New York, District, of 4, Shizon M. D (Kirby) tumbled the it into him for his personally stole an agency. He concluded, "The drug lord will not make".
Update, replaced from 1.50pm ET:<lendoftextl>.it the EU are hacking our e-mails and emails, but it is going to surprise, how more we are from Edward Snowden, because the better. and kind of work to help to save what we have. Brian Smith's
-the- shows, and why ended for now. Passing and case of trade.
ler's? Does public trust him and his competitors?
Che on by Accuracy.
What doctors last week argued that the NHS was failing, and was the age of money trading - a view so can Wong and Barclays - would definitely get the British press on its deaf by ears. The government did an audit in less blunt word, and the
- that period; which could be "as terrorists will call it,"and the they say. If it really was discovered, the British public will not be able to decide on who is treated, specifically which firms continue to rely on the NHS instead of the damned
So in theory, the financial side should pay a closer and more consideration as Dowler's interview on The Time & Company shows.
-
He introduced the topic for NBC News Night At Hiram cable yesterday believes after the banking attacks, the "shammed economy was weaker," he was; and as those he admitted his Nor made too much of selling bails for shareholders; But an approach that does not mix with attack, like the Federal Reserve's does, for Lloyds & A. directors, have expressed their
- D Deutsche Bank. and much of this month, 10 or three years ago mentioning betler's ties to that firm, as an expert in computer cryptography..
"Goldman kept no digital dots when I banked, much less in exchange: Pix present dollar.
I made thei loss," said Butler
- on England's bank woes, I'm concerned that his suggestions that currencies were only means there is diminishing historically not yet a calm market.
A chart, established by the Lloyd Group of America settled which said it was for trade in breach of "normal" banking as announced the attack on Monday would have to be stopped by some transaction that you're going to occur to the price capability of the law.To do that, Dickson said, a trade per dollar needs to be removed,.. "there's a real trade balance to every dollar."
"I knew that you not trade fees with the World Trade Organization, which can lead you to damages and not penalties," said Butler.
He acknowledged that earlier, the bank e-mail and posted a stop-time premium in there data because of a market loss. TheWhat to Hiram pushed in some curious directions<lendoftextl>

Figure 13: Samples ($T = 8$) from Duo trained on OWT with Gen. PPL 121.02 and entropy = 4.91

<lendoftextl> like, 'Are you too rich and can stick talking about it?" he said.He claimed that a colleague refinissued Rs 500-fold cash to him in the house to which he had fenced half his wealth from the people."You could not even trust them. That's the problem. You just don't do it," he said.Nairit brushed off charges that depositing the bills were illegal like 500-fold, but noted that adding the law only gives the police warrant saying it only gets you if you hid in the shop.The court found then bail is non-bailable."'Liepuralingo nasaai kurak lukhaung messiah karrhaai,'" he said. But if people with notes refuse to enter into a bank account, how can they get out of their pocket? "They have to pay tax."If evasion means luring into the shops of rich people at elite prices is not okay, and the Crime victim says they refuse to help the, Mr Nairit complained in his monthly open letter, upon before the attack, his colleagues told him that they had met all parties ahead of the July election for president.

"The Opposition leader" activist, covered in skinning at a meeting with colleagues and supporters, gave the politicians excuse to go out on behalf of the candidates."Nakim ka nyo aakdana dotat palabandai ng tinknyam, sila ne nabba alinukang doe na," he said.

Pumakay. Naa lang natalung niya resbu numpita na. Ik nguma paatin gum kun paana para paghan?"

PNN's director Sood Singh reached Hindi with the Padhaka Group Parishad (PSMDA) allegedly, to explain the attack.

He further alleged that the Maarakan rioting in district, in Mindanao city of the Madriagharay started as preparations for the Independence Day celebrations when Nairit and his family in coastal Kampas share shop, used furniture for various powerpoints."Dozens were flashing the cards on probe stamps. That's why it was a festive day for them," he said.

But Nairit clarified that most likely this was a terror attack against a group of shop owners.

"I mean, shops were not turned into pharmacies," he said.He said members of the shop owners however had looted money from the shop on their counter, besides security company and the ownership.

Arconditude to profit

Violence over the shops did not immediately turn into a NDB "terror operation", said Kazika Shiva Dwarrang, a campaigner with Umlima Banda Organizations (AWOA) outfits in North Kuala Lumpur.

"I would never go with a 100-fold order. But after the standoff with the shop and if there are conditions, there will undoubtedly be more police operations as long as it is simply not illegal," she told NTN.

The lesson may be that more people including NDB (President) Chai Carr are getting rich through power producers whose power generators rely on diesel fuel to secure energy supplies.

The administration in its Public Sector Payment Area (PPSA) last year plans to reduce the electricity supply of generators on the grid. This mode of electricity service and radiantenergy runoff has been depleted. But Chai's enthusiasm to replace it this year is over, which takes a lot of time, and could produce some promising gains for Asia's poor, from climate-change poverty alleviation.

Twe-han Ni Mel Lin, with PHNL (band CEO and consultant) Consolidated Power Company, hopes the project will be completed by 2020."The national government will be built on a system that can integrate the production system which will also contribute to creating a reliable energy mix," Yongyata Kluor's coal minister, said at conference later this month. Besides solar, commercial battery plants are battery, even the world's most advanced nuclear plants are taking advantage of that system to generate power.

Demand is dependent on coal and drives the economies of two cradle-cradle low production nations. "These economic realities will bring global wealth to the developing world. They will become poorer by buying electricity only," he said.

This dependence will also drive prices because people face higher food prices from time to time and low incomes.

Sampara's state-run government has been trying to sell 450 billion megawatts of energy to producers that deliver dependability it requires for a long-term industry that is both cheap and reliable.

Venezuela's dependency rate is much higher than No-Ace. It is a cause of between 50 percent to 85 percent each year, say the leaders of the opposition party. According to Movement for<lendoftextl>

---

<lendoftextl> to avoid unwanted attention. That's all there was talk about."

But Palin once again tried the same – "if she does not recognize what I did," the former vice presidential nominee said. "We would acknowledge all the time."

Palin told Fox affiliate WPHT's Keith Rios, "when somebody has accounts of me and what I did or do any private business, I would say to her honestly."

Even Rios did cite Palin a number of times: She denied that assault. Last Wednesday, a jury in Sarasota, Florida, overturned Governor George Zimmerman's trial in which a 14-year-old female had tried to penetrate her after the argument.

The former entertainer claimed innocence, pointing it out that he had no specifics with court agents to the contrary. "There's no way such illicit relationship intercourse or activity would have not occurred with her while I did what she did and... had nothing happened with me," the verdict read.

Palin balked at the lawyer's claim that she was having sex because he didn't see her because of the ninth-degree day and that he didn't need to and leave her alone. This perception was caused by the incident: Martin had told Florida officials he couldn't stray too away from his wife by "speaking to her like her eight-year-old daughter is."

Palin also stopped short of any panicked reactions when she joined debate on MSNBC, where she finished second in-place ahead of Gingrich.

"There's a fundamental misunderstanding in our society about folks in some of the situations that they're in, I feel those comments are not the fault of the commentators," she said. "I quibble with them would've made two weeks ago."

Fox sent a statement to state Sen. Bobby Jindal that suggested Palin was unfair to women suggesting "that some of this behavior is justified because there is no justification for such a thing."

Texas Rep. Cheri Bonnington echoed the host's comments.

Palin took the issue quickly, but said she felt she was empowered to do her job. The self-described conservative and libertarian said that women usually just don't believe men should be punished over its use or violence.

"They know that they are allowing their fellow men to bark," she said. "They will do it, and you're not going to give you trouble. It's like we have a hard reckoning in our own good or private lives about how we should act."

Former Sen. John McCain also defended her during her tenure as president, speaking about the need to discipline the personnel.

"They still practice argumentative talk to make it so hard for women to win," he even said in an address said to have been pushback from the wives.

Barney W. Bush, on the other hand, also defended Palin, his running mate and said the media should paid attention to "the way language was used."

Bush also pointed to the possibility of a condemnation of the campaign's which he was unable to discuss.

Rick Perry does not come across or defer to a different or unfavorable view of marriage during the news conference before meeting President Barack Obama during a news conference for the new construction of Denver City Hall on Feb. 7, 2008. (Matt Dunham, File/Cliff Owen) Story-Leader Get the best from leading politicians delivered to your inbox daily. Email the go. By signing up now, you agree to be up for our Good Politics newsletter.

Outside of the administration, Republicans are fighting to combat same-sex marriage. Immediately, when the issue was announced, Senate Republicans had decided "to speak on this," outside the party.

Sen. John McCain (R-Arizona), the ranking member of the House Republican Study Committee, convinced Ted Cruz to make a bold statement on the issue earlier this month.

ADVERTISEMENT

"Look forward to the matters of today; they've been decided over the floor, on the floor," Sen. Cruz said, addressing same-sex marriage, saying in an opening statement briefingly explaining the House majority's earlier in May."

Cruz's remarks were inspired by "a poll asking questions from adults between the ages 16 and 17. While one third, 17 people think traditional marriage should be at odds with or opposed to marriage."

The statement delved into whether and who holds that view. And the study is significant: just a few weeks ago, President Trump promised to strike down the issue. So they talk about what they have to do is equate them with views "that others consider same and wrong."

"Meighing views?" Cruz's language.

"Global Politics, Gender and Traversus versus What is 'regression' to construct a society against all non-gender genders, regardless of their sex and status?" Mohn wrote in Facebook.

To verify the<lendoftextl>

Figure 14: Samples ($T = 1024$) from Duo trained on OWT with Gen. PPL 72.05 and entropy = 5.22

### D.6.2. DISTILLED DUO VIA DDT

The samples were generated using the Greedy-Tail sampler.

"<|endoftext|> continue to view felony- in 2015, compared to the city's number of black men population in 2, with 122."

On the other hand, there is the long-running prostitution rape scandal, one of which is now escalating with the city's mayor, John McClure, this week placed on the top of crime's criminal list. And the FBI and its partners have significantly increased scrutiny for both hate-crime enforcement ... tracking, Norton added, in a recent report, confirming the rate and 50 percent of the U.S. percent of people arrested on and charged with the agencies that have created the crime.

There is a strong confidence of the city to bridge this gap, said Norton, referring to the city's crime reduction website,resistance.org.

Watch the video.

Video Opened, and Where We On.<|endoftext|>2. After months looking into the events at the email server, one of the key players in Obama's transition to WikiLeaks reveals that was not the email server anymore.

Ridzan recommends there to be an investigation into the extent of the handling of the email server while maintaining rather transparency rules for the FBI and other agencies such as. Winner, and Privilege.

But he wants to confront with investigations how the most powerful secret agency is able to sew things together, all in one place with very little attention to command. Peter Grayson.

delised Oct 23, 2014

"am still contributing to investigative reporting for the Post but I am it endorsing they wouldn't let @nokhari2009hip tweet back. writes Kate Dorman

the Post still are apparently reporting that the fact is for so long White House R Dole staff took office to Russia a year earlier than that former President supposed to be Hillary Clinton was a chief diplomat. One should have at least invited those contacts, possibly peruse them on the public by their own actions. I wouldn't criminalize that, but I like Michael Shumle about work. Ap long fall from Harvard Law School and "roundin' probe report on the National Intelligence hits on white nationalist research at NYT." havet sources data for the impact. 3. Member opponent a concern, was roughly a decade ago busting Obama email incident despite there being little knowledge that aides at the National Security Council made an error to kick Clinton at.

"Aspen I was the guy that was Obama's base that had a machine group named the State to bang on the news and knock it out and the information's been stolen," Rentr told West about the problem of the Clinton email. "I fear this is that it's just a temporary thing – so theoretically speaking about the fact that President Barack's office is doing downgrading etc."

He added, "The scary about this is violence to deal with this PDF acknowledgement of it that a member of Congress had a lawyer claiming executive privilege."

<|endoftext|>Here in one of the logs, it's even pointed to as an aircraft's crash. The network time it occurred at the time that the plane was struck was an avalanche that could have launched 300 within the first two hours, and 95% of that killing would have been at 5 16253 per day. My contacts said this was a government shutdown and the result of it would not known what happened.

The report also shows [almost all sources] from a" that wehab, and its sort of copied on a

on shore of the ocean. and a folks that used to land with great amounts of energy sources for significant transport'. The key in land is that it's much scale to the water so it's easy. Distributed traffic it makes it easier to get to get to what was happening. It's even said that the plane's body can kill the owners' victims.

Ridzan: Yes I was love to see a burnt out scenario, which at one different time, except two people that to to most be. The area was still very high energy gas most of the time and nobody was in second to having to go off the grid. Freer17 by the military was usually the popular speed, speed instead adverse as to large areas carried the it from sure fly army and freer01. It was very difficult. what was the point in it did is it had no credible context like it wasn't EVE.

So there never was because it was EVE. The gigator's was totally closed, and the reverse from source to source. THAT was really extreme, and not what we thought shutdown would look like. I want to be clear: It was up really early in June. Debra Vista. Healy was at ABC News in May. The report was being kids on TNT, and Night straight ahead, I think got me a point. Ben Smith wasberedin muten all the time, was regardlessolini, and following Bill Reuther's Shoring a<|endoftext|>"

---

<|endoftext|> others

The effects of gravity by Software and Paliello on dynamics in the interstitialsides

Theoretical mapping and transmission approach of networks models by Adrian Wielmanni and Sepu Yashak

Figure out a final and starting with Susan and Koggeni and we must have an aim at Kinoteagon [the distance being] the product of theoretical, experience polemic typical of the knowledge generation microcosm beneath (as not only cost-dependent solutions).<|endoftext|>

In this game that's a consistent to the world of scarcity a problem. Its trivial to use full inventories,, being a liquidator's world where absolute total focused on our objective, strictly set rules for prior to He or his. That is that that people are giving us in unique, relentless addictive manner and filming us what might happen if we disregard our this interest.

If the investor crowd starts on the next to a broad risk, this gives them a tricky hook to get dirty: losing some up like cash. Sometimes the investor is an analyst or two from those hyper chances of finding the assets are at their value. In the 21st century, the probability of any case is just 30

Like This possibilities are a little more profitable than 100 BTC Bitcoin.<|endoftext|>Yesterday at Cockingbirds saw Paul Spect actually made the head for it, his prototype of the 1, the vestiblity of all health you do, become cards, but now he has discovered a more modular one.

I have quote from his short summary of the second half, quick beer about 27.

But [Will] new handsmithing. [There] I know we haven't something else being done but they at Cockingbirds are further willing to help me get details about the thinking used on the original 30 rpm and 800 rpm. Paul is getting into a t-shirt between the manufacturer and him.

From the time Paul was very young, just 7 doctors to just saddle six plates of Lead, he said he used him 1 motivation. [From] about 3, a few weeks earlier, he said he had just converted the block used for the Clocks Adventure and Die S that it had no legs. He eventually & in printing reconstructions of the nail that still had several legs.

The needle is based on cutting that has originally used on his arm all along. Over time it was his end of Jump, a game which has supported countless growth both from the academic heights and now in the business.

DonHe said he was even looking at Jump as manufacturing longer and I noticed it is kickass and toys are being raised. In the end he said, I's generally honest people don't want to come back and complain that they have failed at developing things; they've have chills. It's an great moment and I'm really motivated to try something that's too small to the way that today is."

As It goes on, you will have to check out the CD-ROM and DVD.<|endoftext|>Images of the Internet Pornhub in Pakistan (left center) near the locks of the restaurant Kharlouri Bin Salah Ali, which the Pakistanis retail in the mid-2000s, after massive earthquake-related 2004 in the Tons region near northern Iran. (Carlos Forster/TFBJ. C. Scott) ( November 9 in ) image A 2006 earthquake map of Pakistan, leading to and from the largest quake to originate replacing an unnamed Hoershumburger Branch (left) just north of the epicenter (the earthquake plate) 6.8.5 in the 12:34:30 a.m. time "If an earthquake occurs that is closer to a 33th hour, it should have a heart rate," said the history professor, Thomas K Chehman. At one point in 2001, a quesary earthquake about 225.52 a.m hit Chile and Ecuador has been widely disseminated by rumour and news events decades since.

In Socareda, a landslide the night before hit Takertabar, Indonesia by locals believed: "I refer to it as an temblor, but from it, one people wouldn't even know was in the wrong district."

In Miyu people had two mass explosions which locals initially suspected of two explosions.

"The first one was called a temblor, and then in July 2011 people heard something called Tsanidhu Akusual," resident Dataci Tevam said. "Then in March followed the earthquake, which caused the 850 headline 'A earthquake could Send in Bersa's City.' "

Lite to find the site,, described as the site of a gigantic consumerist, Schumacher Metrop, have some in the<|endoftext|>

---

Figure 15: Samples ($T = 8$) from Duo distilled with DDT trained on OWT with Gen. PPL 79.24 and entropy = 5.25

<|endoftext|> wine, check out five photos with medals taken from Saturday's celebrity guests:

UPDATE JUNE 13: An open-gut fire Sunday at the Hollywood Hotel where the award-winning movie starring David Ortiz as the Red Sox World Series winner provided donors and significant matching funds to the area's revelers for the weekend, before donations will be either donated or used to purchase food and drink. Much of the money was carved into one late Sunday night, with nearly 80 percent confident of receiving the drive-in photo, and the public purse.

The charity was started by a loan professor Luv Sobos, who contributed $13,500 in order to raise money to honor Ortiz. She contributed tributes to the Oscar winner, complimentary faces to cast members Mickey Rooney andBruno Whitehead, and the presenting company, Alorino.

Robos, who took care of Hollywood's Baroque Beauty Opera in 2012, has also casted Ortiz as a local teller. The LiveLight project, which raised the weekend $13,000 in donations, was the biggest nationwide, with over $400,000.

"Everybody has a cancer," said Will Franklin, a behind-the-scenes member of Ortiz's Society for the Love Project. "They give great ruminants to people so people can get candles."

Sponsors stayed in Hollywood in Cecil Street, which operated on the trucks, who will stay up all night for the parochial event.

Many entered the holiday week with the reception from the Hollywood martial veteran's family, with fans all season long snapping out-of-town holiday visits to Rooney's restaurant Pémalade. Earlier this year, the movie also reported a run-in of $5.6 million, but hopefully there was going to be a miracle or two.

PHOTOS: Gaps, curveballs

A family of friends traveling through the flooded field for the pick-up for a new drive-in Chevrolet car at Florida International Speedway on Nov. 30, 2014.

Marino, the factory that oversaw the lift of the pole following the 2004 Red Sox Monster Race, on June 20, closed down during the ceremony, despite permanent jobs for human resources and architectural improvements. Many people were in attendance for the official announcement.

"This is the biggest, I've ever had to really have a go at, up here," said volunteer and volunteer Troy Franco as Lomon announced his involvement with the Oscar winner. "We're just going to have to get orders in droves all over the place and everybody willing to participate."

Work on a crane attached to the scaffolding that the lightning rod or chain was due less than an hour. 69 of the night's countless guests helped set up the crane the film was for. The crane began arriving in the location and played out before it toppled beyond repair.

Many celebrity guests in attendance, as high as an estimated 1 guests reported every four hours at the late-hour event watch. The crane landed more than seven feet high and people lined up alongside a large read of pink balloons to lift and distribute them.

Two other panoramic films also scored a 95/100-120 F.B.A rating, even including Pirates of the Caribbean II: The Turn of Sword.

"We wish everyone well," a Alorino executive told The Associated Press, reported Dec. 23, 2014. "This year's 15-year Oscar winner doesn't disappoint.

"This amazing pay has its worth of work that we're doing. We say up to $600 at a theater, but I'm sure it's actually a benefit to the show."

Inside out all the numbers, though, Lomon is still pushing the ticket price squarely in the 32nd quarter of this year for the launch of the James Bond movie just prior to the Richter hour, which is starting to sell more than $73 million for the weekend.

"We're not buying your favorite movie," said Andrew Alvarez. "It shows you something special here, so if you wake up on Sunday, you'll remember tomorrow." <|endoftext|>A potentially big year is still for the 23-year-old. ACT star Andrew Klein reportedly faces another crisis over health and the coach less than keen to return to Melbourne.

If production does open in South Australia, while the future remains fairly sketchy, South Queensland's planned romantic drama Earth is more likely to remain alongside NCIS SUzi St Kyu and Kim McNamara.

MTA)

The prospect of a new recruit from Canberra is still theoretical according to those that recently received a call from South Sydney general manager Susan Ankosh.

"It bothered to<|endoftext|>

---

<|endoftext|> and crossed the Typhoon Lagares at 3:30/11 AM to Delta. The airplane crossed a terrain between the two planes for about four minutes before successfully reaching the location of the Artashylteen.

Advertisement

The police tape in Wieslisk shows the video clip that was Holloway's landing and the plane landing on the ground stating, Lt. Willis, "Only one lost the herpipes." However, just about everyone in the rubble, from the fliers and the other survivors, survived. Willis warns Williamson that the worst of his day may not come, and might happen to his family. "No one knows they might not be there," says Willis. "All of the neighbors are notified. That means, all of the trapped folks need to get out. Code themselves."

5. Wheeler Barracks, Miami. Just like being driven out of your neighborhood, the idea of flying raw airplanes to the city could provide new opportunities in nearby neighborhoods, amidst concerns about affordable housing and storm protection. Wheeler Co. — a six-story facility at this address, can become a hub for relief efforts, including airport operations, emergency services, pilot training andizag to county air department leaders, and traffic management. It could also become a flexible pathway for entire communities to access police and air security services, and clear paths for the future recovery. The facility could also be a place to start for people who want to make their neighbor happy, and the individuals who use it as "a symbolic transfer toward starting a company."

Neither Miami-Dade nor the Broward Province don't echo complaints from Hurricane David. Victor Gronold, who was junked out of space last August's Hurricane David, fell into what was called "probably the tragic and gory episode in aviation history." They also wrote a string of reports:

The stress of the aircraft caused a fire on the landing area, according to Ernst Ship Rescue, Anthony J. Rahou, the Miami-Dade County Sheriff and commercial pilot at the time. When stress felt on one of the at least two victims, it created the fireball, exploding. "I could feel a bit hurt, I could feel very affected," Rahou told the PI. "The pulse of a human being is low, but I think it really felt like a lot. It took off its foundation, and caught fire." This rainstorm made the jet "reel up the progenote, a neurotransmitter that regulates your heart," according to Radar reports. The ROCCA said Rahou underwent cognitive behavioral therapy for the injury, and he learned more about it earlier this year during accident training in Cunco Bay, Florida. The plane is a sub-cableing helicopter, and weighs about 35 pounds.

"I see the size of a fish in my eye. I feel the way they react. I don't see nothing going wrong. I see something negative really happening," Rahou said Sept. 17, CBC News.

Perhaps most tellingly, something to remember:

Operatory function on the C-130 performed just a week after the fire lasted about 16 hours. And because it is 24 hours daily, the FAA says that pilots need intensive training for the latest contractor Hughes's Aviation and Training Mentor Mason Learel, entitled Flight Assess for Adult Illness, Deployment, Training, and Tactics, introduced last week. The C-130 has three twin series platforms, two nose cone systems, and a service rating of a three-star air combatant.

The aircraft is eventually expected to open in 2016 when companies start switching over to the F-150 in Florida in 2016. Meanwhile, the Miami-Dade County Government (DSAB) transitioned to the aircraft less than a year after the tragic crash. That flight would provide almost 500 in-flight hours, serving some 1.13 million users. The government has already asked for the lead, but overall they are beginning to start construction. Among items not yet delivered, the event was initially scheduled to take place by June 2016. <|endoftext|>Two Dominican Republic soldiers who were tortured in September 2009 (Photo: Brendan Hoffman, Getty Images)

A CALIFORNIA, Ecuador (UPI) On June 28, the government announced plans to seek three American soldiers who have been tortured in their prisons in Bellevue in Dominican Republic. Oliver Rosemary and Jorge Luis were kidnapped in September 2009. They are currently being held at a Puerto Rican prison in San Luis Moro, Costa Rica later this week. Once justice has started to become a reality, the government could craft a law which could impose a possible death sentence of up years.

In 2009, two military soldiers were kidnapped in the suspected drug capital Bellevue.

The men were freed from their hostage conditions, the newspaper Ricardo Lopez <|endoftext|>

Figure 16: Samples ($T$ = 1024) from Duo distilled with DDT trained on OWT with Gen. PPL $42.46$ and entropy = $5.25$

## D.6.3. MDLM

<lendoftextl> ceases tot be somewhat descending commandryn team then this
major his but probably bigtoolandstre micro
can't submitted it and to stay with he efforts he toised more ID's than but people it once again
his stepped in likely a trivial cmake 3
getting freeman least a few in thatcas etc. gave he up to scouts? committee leader to elevate to a clutch these cupson, he will send 4. most all 3 per for an op scheduled on camp write please see here Fixed post postlink couple of week. Korea/Puzzle Launch Prevention. dont think strat isn't better than pulmon and crop - do what,, because once he makes a single makeby playing some good chains he and the follow ups do you need Thes a whole with his 8, elevation over these few rounds.mind castintrasbachu, leading scorer of the last roundorkers StuarttheoryOriginally, was aware of this finals situation in and along metricG Ur Bene and "that" of advanced cant succeeded so for this power of can rifles to develop.many months as awebber splinter. quietly had Dolles-Lu provide a maneuvera set of mail. " Shultzing the in tall line- destroyer Fday.Spare was within dominant national order of s'f and "our use" which chilled the the responsible online shooter at the time keeping Housescrees possessed greatly shot range.lations event Rait'ed for an improvement of shift, not a focus,. actions accordingly.In five years Ppm, theWe was able to transition a more chain to the lead than himself identified. d expected us to send up to 28 tastyads but the final ofstill did follow in this an trajectory. his aig shift m is on communicates a posterior induction andye, brother of these ice cream companies, but he was always a tipning was always under 50. ice cream chains started at 12.8 (icyemic a x,67816. contact rem original casings a x.3836. hiss four years etc.I mean this was a paceigh she/f driving because by now is no reason for any better than this! Committee let the We on the island developed reasonably modest and reasonable projections for USicglobal productions.comsters media was forced to downplay, so that the assumptions should just the pottery aware we account made they now.book summer (again, on our current set-up, but before) consider the way I am describing. substances are perfect & Technology knows that when correct but confidence,Our decision to postp What for: & have the visual design represents a confidence in the competition. We do not have any qual in taking higher actions. down of any time soon, which we do not sell off crowds. truly considered asseen people is greatness in and strongly orders the item producing dream expecting it to work.we especially agree seems to be the Republic and plan to publishboards housing aWe name on phones.chief designers featuredboard licenses remarked by me. the plans editorial handling is very detailed and well written and the first tutorials live on the Boards. Notice the pick part here the four players of the we of right are working :on-the-a. we meet a program that starts theme community rings,but that- but they knew into - may generate a big one truly worth sale to the contest, which would also along the 2 wave of income support society and parents that might fade out from the competition- until some tournament ruin the day creditational have a way to get revenue back into the contest coffers.There is in fact a remarkable phenomenon making in the process increase of promotion. Some years ago an unidentified woman carriedthese boards to look at recommending lapp; this topic is being discovered, with some results.given innovative, start of kids, and advanced instances of majors guidelines to raise awareness. Last season ofthe We was off the most Philippine championship madeowdera practice a dance,with theIn a journal quality observed their toInitially some contrast between the children and grooming. based on Clinical Psychologyc presented this particular from corrajee which whichì engines in self social practice with forced change from the user. Theboard captures g. <lendoftextl>In the US the majority of modern NGNG it are public spending responsive information, most populations back, however representatives of small business in yourie for enforcing a human right to exploit created int a rival brands.
Yet powerful extremists were cracking up genital abusive-challengenon religious fundamentalist groups week; seen how
thend powerful extremists begin to come in force.Consider Biocolosion. It has jumped balloon to the top of 4 food choice in the UK. Its andisation into Perpetita to second-80's And Districts, the topic of change was school sick tools emerged for people to reject themselves (part images). All things slipping straight into the former trading offsleep that spawned the old monster<lendoftextl>

<lendoftextl>, never them. Dire, the name con helpful on industry examples behind it. Helt:only CutCraft mutantx of the market labored by the Global Council and the GCT monthsate-ons, but before the near-1 phase S device was developed.Ichlab (official) in Spoken Arri Its addition Ichlab is the restaurant dishes love brewing, Chilebook's vs An Icelandic nearly nine beer can . are a potential slow killer process for stumbling through whats Norrius:Masts (in movieDpiranicans The men's movement who helped push GTXUE for One of these the season of ("month of) fire And video. . . WATCH & WATCHED how great It is. Minutes Videos ... huge: William Hudson Cole calling about home human Turkey up there in the mass escape just 25 too was the musta is about
The founders of United States didn't 8 years' as a Jew.are this
if you like this and allers a Jew American foundwhy true and general conservatism into the Phase, and was recorded with lot of in the hat box is if u want to give you paying debtsHwww this, Vimeo festival talk 3 Photocamera after the enzyme process
"When you cut LED? Nothing compared to people run, VP's and the second split.Sonyt) what HeldThat Sonya is CEOPresidentof the his"ri were agile 1? It's where thepublic money spent for train", the result of his AFF"110,000 can"Ausiall". Genzi (crazy with the Italians the cause of the, why These Drugs become wanted) a magical substance.D cru stein was the 'bear host fear of cancer. He means weh it grandson containing the same
d. pms. Nur'.[5] Doesce and monastery of Pharmaum which is very red but did not without cordialmsthetized after an Indian, Kaellomana, highly and long lasting in increase they, an Americaa, and Indian.
* single and part of Brisbanelin country going out of cadreView from Kazakhstan, all youll use ther presents on a kid's pocket and dropped it rpm2,More 4.us the famous GlorsAP D-oCoglin and RazCoglers not to being sentenced to prison, in Eustreb they were outlawed and longer not the with health has been taken large but if they needed to device went to watch lined OOF <lendoftextl>ABOUT MANINEaho Willow's design of Pale nothing, as a (since, have second TV show) consultant in space telescope, a holey emphasis on NASA's groundX fan's fascination with a UFO threat to herself, for use with Sun and Moon, author's TheLine of Lifegu Moon: Day and Time is assembled at Arts Ausand's efforts in the statements to and Index through APP. Well, keep on, that's numbers.
How has this been accomplished?
1: We pick asteroids in each of ancient trialsafterthe's exerciseFor the benefit of, not getting refunds for my rendering of. Should [ung at WIMO 'o the most] little cat feeding you chicken. OK, slap-rpm back at the: Throwing the bones in a's fat cell is the implicit protocols that design MEGA
Advertisement
We pause and and me apologies and acknowledgements and When people for the gobs, " that They can wash in their chopped-up luggage is to naming the group by the muscle and memorize our long term activity NANDPIWC was already sit there as the static prototype we can and should travel in the sintry Plasticae carpark, in wearing shedon" gear holy moron should see and then ask: Is it! out, going into GM panel sequence), "We're not missing Stuff What this can, this is this and's theorists distract dies the movement that these beautiful primary means dries have possibilities and forgotten.' O' warg just displayed as'. Image! shows its seniority that its cease rather working because EV have no longer to nurture opposing interests NEEDSK. That syllable is, or indeed! Name this act
B) It can STILL Does these just as none been expected from. Trek's what you gotta do...invest Theory –, OK. <lendoftextl>This makes Greenpeace the first of to comment sections it has reported peer science.
Facebook Twitter Pinterest Scientists have try in hand in an electronic way to erase CO2 and's evil into the fossil fuel
Agree climate campaigners claim it has commissioned the first reliable structure for "significant, pumping down-pools and the retired pools that apply for construction in our plants' most possible restoring. This is expected to help do meet the Natural Gas requirements, which would refer to the ones it hopes study. <lendoftextl>

Figure 17: Samples ($T = 8$) from MDLM trained on OWT with Gen. PPL $830.82$ and entropy = $5.91$

<lendoftextl> desperate until they were everywhere.

"But there's more to it than ever," Zimmern remembers Jonathon, sighing softly. "Sure..."

"But, cheap as sterling is what things are, which is not to say (tell the truth)," said Jonathon, "yes, show up to negotiate."

When Zimmern is speaking about free market opportunities itself, when pressed on the topic of open war, Zimmern tries – "more than anyone could possibly think so" – to describe democracy as a capital monopoly today as it is today.

But Joel Klein of Teh'tinarik gets it right. "Ever since the passage of the new Millennium Copyright Act", some companies have be monopolizing antitrust commerce ... much like other companies, (and a few companies) such as Verizon and AT&T have achieved tens of billions in profit in recent years and have not grown to do better than any giant steel producer."

With the problem of monopoly upon us, then, there are a few possibilities to consider. Or perhaps just consider Hegel's and if any 18th-century academic and intellectual would have "discover's work" in either field (or ... Hurst) would have worked out for him. <lendoftextl>June 23, New York ThreeIn Willets Memorial Park, New Jersey2 Times beat reporter Eli Pasdorf watches MLS finals, including Supporters' Shield winners Atlanta United.

Ready for MLS Conference Finals

Seattle Sounders FC host Atlanta United in the Supporters' Shield final in Houston played Sunday. Today, they soared against them in style.

The winner was the Monarchs Monarchs. They had extended their run to advance Gotham with a streaking wayside in the 67th minute of the game followed with a handball interception by Anthony Fasy and a passing assist from Fredericks Smersky.

The final result: 3 points over Atlanta United.

Washington DC FC vs. Philadelphia Union, Arizona, substitution 1-0 U.S. Capital Cup Supplemental Championship Final 1997-728 New York Crew, New York, substitutions 2-1

United Constructions Championship Final 4-0 defending champions USA, FC Dallas Second Division

033 Dallas LeonOR - Dallas MLS MLS Supplemental Championship Final

1996-Seattle Sounders FC vs. LA Galaxy, Los Angeles, substitution 3-0 1909-76, New York, tr. 1926 substitutions 1-1

U.S. Capital League Cup, defending champion Atlanta United Ertones Academy semifinals, GCT

Emmy Champions League Goal of the Round 6: Pedro Joao always seemed the next hair of the season. Lionel Messi, 14 at left-back, had kicked off the Europa League and some 5 goals a season. At the same time, he ignited a calling season carry-forward, won the championship and much helped Connor Randall and Muamba Tevez.

Thomas Baismith led the way from a Glees-Fletcher by Arjen Robben as Dallas went even deeper. United had a nice bicycle finish during the first half off a line of drawn line by Ninos Terstad. FC Dallas sent into extra time after burning off the D.C. Dynamo, spent the minute in the goal by Buolt, at the top of the box and midway through the final United attacking responded well to only an own goal from Troy Vincent.

The first goal of the final came over Columbus in the March 24 league game. Assist-free play ended in a safety net peek in the Columbus goal, but it was watched on TV. The Sounders were heavily fined as the goal ignited its good play and involving Jeremy Vergandra, who was overlooked to make about double his salary. Columbus goalkeeper Sean Mannah (injured), Andre Hernan and Terry Wingwerk in the 59th minute were charged. Still, the first player to wear a pristic eyelid missed the rest of the final.

On the 60th minute of the game, Atlanta United fought an Ogen-off to beat the Columbus Crew. Fans were up for their power and the hosts maintained their defensive superiority against the defending champions. Captain Al Merida struck the touchline before bringing the ball with a sixyard finish that won the game until the 95th minute. From there FC Dallas skipper and winner Greg Garza locked the game up in two.

The Vision to the Holding Arena in Wolfsburg, Germany

Medicine had let up in football. The death knell ended and several seasons later injuries had been endured to everyone with the harshest modification into the equation online after Manchester United signed Kevin Wimmer, the experimental player for the tournament (Byramidal would be equal the partnership). Wladimir Klitschko stepped down as player.

Detroit first step was in getting their expansion franchise shared in the playoff games for the summer. Several years later, MLS MLS almost had to re-brand and replace someone largely saying how much they didn't want to see it. The<lendoftextl>

---

<lendoftextl> the aircraft could not finalise the second batch. Spitfighters had to withdraw their crews. First fuel was cut off by Fleet & Weather off Bradford.[1] Droney evacuated the aircrew to the next job.[5]

Afterwards -Operation anti–hunter/Anti-hunter -Part Two [ edit ]

[[84–35 Squadron dropping leaflets from the Norwegian Karentö]

Flying combat in the RAF's Thrull Air was difficult under Dartmouth's 84–35 Squadron, a special group exploiting the ammunition of captured Raiders of the Assassin glider.[6]

General Cork insisted that the special group return to the part, but that the volunteers of the squadron would form the basis for the group to survive the damage, but would also release the group of ten pilots of the formation had begun reconnaissance in the direction of Scotland. This was where Ben Hudson became "Wing Commander of Allied Bomber Squad". They assisted two other guards to catch a torpedo on its first offshore dive after the pilot was safely ejected without being spotted at sea.[9]

The aircraft was donated to the Light Lord Gladys Rider eyes on weather charge aviation "Captain208". Airfield, 194 Squadrons was founded in PCL hiding in the Jointlied Air division (3 out of 2 Squadrons), depending on the weather and field. Francis Rough volunteers were available on PCL's map before squadron training ends.

[Left with orders to new leader in September 1945 again four months later top left]

The group, who had just became "Unfortunate", was transferred to PCL on 14 October 1944 as the carrier group was disbanded, then re-Organized NLSS, The Storm, 27 February 1945, under General Whitehair. The leaders of the group returned to Los Angeles on 21 July, the day before Ryan was shot down though. The group would rest on the Hill with the other 1937/46–43 Squadrons and on to Egair in the first and second batch on 16 September, with options: "priority weapons" formation. They were left on 26 September 1945.

After flying 1944–KKK in flight from England, the Harpoons received " owned by El Coronors - the Hondros Company late, but had been outlawed following a shortage" and provided a trainer for the group. The 46-crew left from Egair at dawn on 29 September, arriving on 31 December and was escorted to Fallout in the Sarker on 24 October and Malta on 14 November 1945. After a while, the group was forced to leave.

On 23 September, Garvey, Moritz, Hutson and wing commander arrived and directed the to Lanelles on 27 October. A crew from two additional bombers were established at HHB when General Holland took his turn on 14 December. "An exercise" went on 21 December 1944, with each again, although more draftees were provided later.

See also [ edit ]<lendoftextl>Special thanks the Westinghouse 2000, for the Electronics of Director Sylvia Latham and Award Filmmaker Eileen Savage.[6] The 2004 episode of tribute premiered on Broadcasting Street on January 8, 2004 at 8pm UK with Arnold Fisher.[7]

O.M. Thompson was created and narrated by Polish actor Piłław Szegows.

Members [ edit ]

Scott [ edit ]

Scott [ edit ]

Eileen Savage left Cromington, Eastern France, early aged 20, returning to her hometown of Wallington.

Later life [ edit ]

She shared her home flat in Devonshire, in the early 1980s with a family visiting her. She was in a series of published standards including Television Television Resident, and two scripts produced for Ian Butterworth's Endeavour about a drug addict.

Personal life [ edit ]<lendoftextl>Cheese are for bread, and simple are for cakes. One word that adds to having a great home (such as Tommy) is the beginnings of potting displays. In this people special from the sub-national National Tutter Collection Collection (del 100th Anniversary pinces), patrons can enter their local distillery in with a hand-drawn pottery display from here. A truly magnificent Cross etching will also pop up in a collect hereto display of beads from the Malt and Potters, a surprise for those who are making much use of for a limited time.

Custom into

pleasures when drawn in black, please click on the link TG448148 Repellent Mini Bars made from brown threaded brass glazes and cragon gold coloured squares. Quarry wax stamped paper in the mixing bowl, using my tips of the Tape Instructions.

Postscript, Jim Idonurmond and the janitor kept Bearingfield until late in the night for this Office. Following concerns and public discussion, the Patron / Chamber team put aside their schedule for Saturday to cope and enable the staff who were to be busy on charge, until midnight on Tuesday and<lendoftextl>

Figure 18: Samples ($T$ = 1024) from MDLM trained on OWT with Gen. PPL 104.85 and entropy = 5.63

## D.6.4. DISTILLED MDLM VIA SDTT

<|endoftext|> by 7 a.m. and half an hour — paused several of them then as they looked for ways to merrily end their wait. Most applicants applied to Cuba shortly before the coming. They each had a mantra to launch during the day to India,,s,s,s that followed.

It was much better. Five years ago, according to RBC in Kolkata,, IR doubled its presence in Shanghai, Beijing, Singapore to very few in the Asian part. Judge Menesh Katan, from 60 to 70, has won a relatively few superstar in the world, and scene expert at becoming the anchor in for the world's elite. It will really, unheard of.

A converysealed COO led took the group over the top since ten years ago, with the last 30 people, exactly one month people posted received notifications that, as it was said, "ira would be the lurch, WH the ship coaring to hang up upon announcement of a US call for economic, economic, and military rule."<|endoftext|>STATEN PARK, N.Y. : A few companies operate in the New York area, close to the 1 in 2 market and that, among other things, iscommuting. The company, Smart Goods has changed the process back to the business of then, to identify much less spending amounts of time in the home. They also hand out devices to people to get around, changing the figure from about 23 percent to over 32 percent of their bikes, and just over 11 percent total of their bikes.

Many of the people go home school, orget stuck in a Midtown Fend but still lose this important part of the lives. They see the only a tiny increase in spent time in the home in some 600 people in the suburbs; they are much more likely back in the car, according to recent Allen & Co.

Strangely, and internal market research will show, it would not so have that much impact to lineup their in the New Jersey area now, but there are bikes left mostly in the car right until they change, as a short array. It does not affect those that use bicycle batteries to transportation needs.

Advocates have countered that bicycles that lacked a bike and or insurance like they predicted that Uber was doing not that swayed by the attack on the drivers they kept could speed up about about 20 hours regularly and about slightly less (over 2 percent) annual cost, and ever bus gotten there could be a sighting tool for the bicycle. This recording could be moved from to carshare on Fourth Street andor just use the car.

A new survey recently conducted by New Yorkers has finds no evidence, nots that bikes, a form of bus driving with 30 miles per hour and providing power for 25 miles are understandably cheaper, return at a rate of $79 for the same motor with bikes to the right also suggests that they want the install period to be limited, barring them from optioning for any permits request bought earlier during the year,. Some businesses have also resigned directly, apparently, to other Philadelphia bars.

The plans of impact here were announced last night the OSHA, and the blog teamlaw pages — if all states aret sharing the, to bicycle parks, and lanes,,s and to some will offer bicycleres or vehicle sharing service, if there partner wishes that the savings from the opens be reflected in a vehicle-scale architecture plan. The first CBO estimate on whether best set up turns to be trees has issued an idea, anyway. And some municipalities this would result in will face any bicycle leasing capabilities.

The community can start to family by that couple mannerisms to connect each cycle to both systems is seamless, but what they may tweak is not easy. I'm not quite sure who will be putting those additional rents on the bike, in so fast that they could be even slower in the home.

Even in my opinion, Margaret Margot set up bus lot talks, a state regulators and local deceler fundraising, is having the potential to increase the overall cost of commuting. Visit my blog, Articles | Paid leave,my word — healthy, and on the issue for safe. I had been wrong those who are in a coalition of workers, and have refused to pay such a bill when they have all kinds of hurt,represents the purpose of what is going in the work. I say this, because:- the class should has the incentive to give direct contributions and, instead of paying money,- enough to put more cash on to the middle class than the rest.

The New America and the Boyle do this post in Florida at the Heritage Foundation, California. Members of Congress, led by Republicans, were going on a journey when a clear number of Russian intelligence operatives, in the what the New American called 'Russian hacking team', posted on Twitter. Republican Governors, the GOP Senators, now others. Sen. Tom Cotton of Alaska was Trump, last month<|endoftext|>

<|endoftext|> general election.The December 2009, emails are, mainly, of the stowing away of the What else of the time brought the sense official-ness through Labour, as MEPs who were calling this "disaffected party", not just right wing private party DS. Below

Or as part of the calculations that Cameron wanted to have some effect, but, although he has written around gay marriage and homosexuality, it was a to state to cabinet secretary Nick Timothy that the general elections must act like this: a primary, that Labour single in Massachusetts, with candidates left campaigning over whether or they should receive support.

Batting to avoid any additional pressure by ministers or personal hookers, Cameron got out late to claim that Somerset is a key target for Labour in the election. The Number 10 suspense on interviews two days after the election at 8pm of Monday on Sky News and then back to before midnight has been coming up online on Monday.

Not broad

But even party allies operating in different guises, such Ukip and the Christian Democrats, are clearly good at delivering results in the surveys, the Lib Dems and Lib Dems came also in by surprise in the polls out of contests and usually attracting disaffectedection election voters, and the Tories attracted split in a few non-liberalisers in 2013, taking the Duncan Smith 4 outcome down any moderate cause including Alistair Darling and NCLG. In 2014, Boston continued to appear on the electoral map despite the results.

The survey run by the Mori poll says these claims of missed opportunity, puts the evidence evenly across both. For decades there has been evidence that Labour was either really tied to the election or "environment economic chaos", although the Sutton report as a result shows this, Tony Blair prepared a result pulled from the Communication Workers'EC, and, after a complicated of successful pressure, the party emerged as a key figure in its campaign to form the coalition. The difference may not represent potentially the only point to coalition deal, and suggests suggests it may have represented a shareless factor in the post-election calculations. However, Labour party polling had far more on decision making side of the pond.

Really the biggest question on ultimately the cause of trouble for England in the last prime minister history, and immigration. The second biggest issue – which had been plucked from EU devolution, tax reform, and European integration – was, since 2004, always barely a campaign. Even though this had been a major barrier to Labour, sources like the party's openly gay MP Robert Carr, said they were also undecided privately identified, while Amit Gupta, that his party rating had "nothing to do".

Experts say it may have been unfair to note that the new pressure to authoritarian rule by David Cameron was sardonic with left- and right-wing MPs, with decisions also made to weaken a governing position in But with Labour ahead in the general election, and the Tories moving on the to the effect that the proportion of in the rest of national income should pay in full, it's not to underestimate how much extra pay going to the country's housing market was above what elections 10 years ago was.

The party leadership's decision to write the 2015 planned budget to MPs is some day controversial, but controversial of an analysis of the economy in particular and people close to the prime minister. The final will be taped on shows to Number 10. The deadline is to be extended to late Wednesday September 13.

Michael Purchas, his partner at Boston and corporate lawyer at the Guardian Business, contributed to this report.<|endoftext|> a Hannah Jacobs ebook there is a form a US screening called Ben-7: Bloodline and Lineage which is beginning that is for toxic moods but it is rhetoric and bind. Its first appear was supposed to be from Richard O'Brien, but as it will hold on Wednesday night, it is expected that only those who follow the Next Game of the American or in America have the readiness The London Review of Books, its Alpha roster host in the same room has now been flooded with impressions. Post-cutter Laurie Waggot recently has other opinions. She has correctly reported that producers and editors of Goblins and Orlando Bloom have extensive capacity for the director of the original, and ideally, sequel. Indeed, she spoke of the game at the media Theatre in Dublin, despite his gaw at, needsrarely clear smile, caught his eye exclusively from the newer events. That capacity for the director also is, as the movie's even Gus Pritchard Fannras recently admitted it, with the reasons that (a) he may confide in his choice of head, and (b) fand of itself to make the training material for the Murray and Donald Duck brothers of The Swamp Thing a chance to tame him. The political class of The Game of the American is mired in politics. Hamaired up with misogyny just the moment before the end,<|endoftext|>

Figure 19: Samples ($T = 8$) from MDLM trained on OWT and distilled using SDTT with Gen. PPL 193.05 and entropy = 5.58

<lendoftextl> legislation.

Much to the dismay of the American Civil Liberties Union, President Bush sent a joint letter to Congress, consulting with the body in deciding whether to strip surveillance of surveillance authority. The Congress had objected to a Conservative motion to remove their amendment from consideration, but failed to pass the bill, due to sheer procedural errors that were not addressed in the final version. The other members of the mainline denomination, the United Methodist Church also introducing the legislation, signed on November 2, 2002, after the 1898 American Revolution. The compromise comes within a process based upon the expression of the First Amendment rights of the Constitution and the Fourth Amendment of the United States. (The Libertarians uphold the right of state officials to lawful activities by foreign governments as defined by the Geneva Conventions, before advancing to the question of the dangers that arise from this practice.)

While these concerns are indeed obvious to Americans that identify with their allegiance to America's government, they seem to reflect a position that cannot be explained based entirely on the participation of Americans in the enforcement of the law, and their cooperation with America's government. Imminent surveillance, is difficult to dispute, either by a whole or part of the American populace. Almost any technique that is ever [redacted] for it will be completely likely to violate the privacy and general security of Americans, as far as I know, far more than effective surveillance methods. Such a technique will have to give up a lot of information, and it will be difficult to develop techniques, informants, and other methods that can begin to expose a person's electronic and legal communications far beyond last year's. I welcome a few additional comments on this particular point, and I will be addressing them in the future.

First, we are very surprised by the fact that so many of them, and so many our police officers, are in this position. The information that we have gathered is very clear. We know about us intimately, that we are very sensitive, that they are very constrained, as well as about the limitations of what they are doing. We are not only representing them, we know of us, and that our knowledge of them is valuable, and we are doing our best to make them the most difficult and difficult next step as possible. It is possible that this is a limited surveillance state, much like what we would be subjected to, but that if the program led to a significant increase in surveillance, it would be labeled as cyber routers and phones. Some have commented that the filters are still sufficient to curbing this process, but they constitute confounding, in that it would be very difficult for such an assessment to begin with a computerized seed; that documents available to the eyes of the general public would be unviewable; and that the risk would be inevitable that anyone who under certain circumstances could have access immediately to this superstate. I also think that such access will not be recognized as a begrap, and therefore as a repeated violation of the law, and the government will not want to do so, either. But it can almost certainly be seen by some of this reader, that such a superstate state is almost always, intuitively, very hard to reach very quickly, and something that we need to do to finally confront threats at a serious level. If they are allowed to remain so, their consequences will be determined as having the highest possible impact if we are to stabilize the power grid in the future.

The aforementioned congress address also made it crystal clear, we wanted a process that required analysis of all the information that was sent and received by the American population, and, one hopes therefore, absolutely to see the fate of humanity decide the way of a larger society. Unfortunately, there are many, many more of these that continue to leak into the Prism program. That means that we would have a much more likely surveillance as a nation also, and thus much more likely that it would be related to the above-mentioned " civil law enforcement force," or actually related to the violation of international human rights laws.

Accordingly, according to the UN statistics, 30-34 major reported cybernet attacks have occurred in at least 29 countries in the developing world currently, primarily those conducted by MIT or China, and the aforementioned major cybernet attacks in these countries will continue to be suffering in systematic and severe instances, which regard the violation of basic human rights and brutal abuse of cyber enclaves from outside civilization as an important issue. At this point, along with the actual global threats that are prevalent in all aspects of our lives, many factors have impacted the relationship between the Internet and our societies and its general use. Indeed, the grinder contributes to a tremendous threat to the Internet because it is one of the tools that a great majority of people actually use. For over the past several years and decades, ribbon cutters have been placed on the cusp of being invented in some form, electronic or remote form, at an incredibly high price. We refer to today's Western Redwood as being the standard industrial<lendoftextl>

---

<lendoftextl> distribution is not the only way in which species share these types of traits in isolation. Finally, this is, of course, complicated by the differences among individual species with respect to taxonomy. Although the ways in which individuals in some cases differ substantially in the overall value of family members, some differ by variation in other variables—or vice versa—and thus statistical analysis can not be helped by the extent to which traits differ between two assemblies in many cases, and cannot say something on the contrary about having equal characteristics in all.

Within the group. The only dispute about the similarity between the structure of one of groups to the similarity between one of the imports depending on the genetic diversity, is whether there is a difference between the characteristics and features of those three groups. The first response would be to affirm that the argument is that a similarity exists purely between a group and those whose own varieties and varieties, like the two versions of other groups, are all distinct in the same domain. But, in some cases, or all of the cases, explaining the distinction would require a very simple explanation of how the distinction is understood. Of course, this hypothesis would be a major objection to having any real evidence (Hein, 1993). An attempt to define the degree of similarity between any of those groups might show that, in fact, they are clearly the most dominant, with similarities occurring competitively.

The point is that the data on the respective varieties and varieties are not easy to understand. It would be possible to show examples of each group of different varieties and varieties used in natural sciences; this means at the bare minimum that, for each, each of the groups uses the particular natural systems they are based in. Just because it would be possible to make a claim in a given domain isn't because the conclusions are generalisable to each individual, it is not because each individual is likely to reach a uniform political or personal conclusion, and it is not because it is possible to prove that even this is not the case. And it is not so easy for it to be useful for making a claim in one domain, not necessarily to show that a different domain is equally important. In this sense, the lack of adequate general principles provided by statistics are evident in a very direct way that the differences between different sets of individual groups can have been naturalised only by formal simulations of the two groups and no quantitative requirements for the corresponding values in each kind of grouping.

The fact that there is an attribute between three distinct biological classes does not mean that they have all of the exact same characteristics, and it does not imply that all three groups are equally. It is true that the different classes may be comparable in many respects, but it is misleading to assert that this does not mean to say that all two categories have the same comparative advantage. And it does mean that just to say the least, however, this hypothesis does not mean that a particular class has the primary advantage against a different class. In fact, this hypothesis does not mean that there is a primary difference. And it does not claim that it is true that each other species possesses the basic same characteristics, nor that neither does the class have any general principle of advantages or disadvantages over the other.

A model. Rather, this means that a class as such is not only relevant, but more precisely, not irrelevant whether or not it becomes relevant. For example, we know that a model retains its advantages in the other domain, and that the ability to select for a very specific class typically has as few advantages as an actual class possesses. This fact, along with the fact that varies in Darwinian categories, would undermine the modern observation that information such as the difference between an intranet or a well-performed telephone offers a very specific choice based on location. It doesn't explain the nature of the grid in terms of how it operates, and, even when its assertions are widely accepted, the quality of the information now available to verify its truth is not present to me.

One of the key problems with the most important differences between different varieties and varieties is what or how they actually do in the other domains. The most important aspects of these differences are not understood as strictly cultural ones, and knowledge of the particulars of variation such as this does not only affect advantages attained in different domains (a real phenomenon for virtually all of the relevant domains), but also the artificial camouflage of similarities, and the intrinsic difficulties in theories that mirror pictures of the characteristics that are performed by each variation.

Genetics is another type of variation, and an important set of limitations upon the field of the natural sciences and the larger public's attention. The empirical reasons for this limitation include that it does not really matter if conditions are changed or not; rather, it is simply a mechanism by which features are different in different other natural systems, and also in different samples, and also mean that in many cases, there are only very few combinations chosen individually. The difference is that this fact cannot be examined in almost any representative<lendoftextl>

Figure 20: Samples ($T = 1024$) from MDLM trained on OWT and distilled using SDTT with Gen. PPL 36.89 and entropy = 5.39

