# OpenReview forum: "The Diffusion Duality"
_ICML.cc/2025/Conference — ICML 2025 poster_

### Official Review · Reviewer_oihg · 2025-03-12

**Overall Recommendation:** 4

**Summary:**

This paper finds that discrete diffusion models are just a transformed version of continuous Gaussian diffusion using an argmax operation. This lets them borrow techniques from continuous diffusion, like curriculum learning for faster training and distillation for super-fast sampling. The result is training that’s twice as fast, sampling that takes way fewer steps, and even better zero-shot performance than some autoregressive models. It’s a mix of solid theory and real practical improvements.


## Update after rebuttal

After reading the authors’ rebuttal, my overall evaluation has not changed， and I still tend to accept this paper with a score of 4.

**Claims And Evidence:**

The paper makes some bold claims, where the below are some points that I think facing some weakness from my point of view:

(1)  First, the theoretical connection between discrete and continuous diffusion is a cool idea, but I don’t think the paper fully proves that discrete diffusion is fundamentally just a transformed version of Gaussian diffusion. The mapping they derive using argmax feels more like an approximation rather than a deep equivalence. They don’t really show that the key properties (e.g., transition dynamics, loss landscapes) are preserved—just that the marginals line up, which isn’t enough to claim a fundamental link.

(2) The 2× faster training claim also feels a bit shaky. I was expecting a breakdown of different factors—like, does reducing variance actually translate to better generalization, or just faster convergence to a similar end result? Also, the use of softmax annealing[1][2] is a known trick in discrete optimization, so I’m not convinced this is a novel contribution.

(3) Then there’s the "two orders of magnitude" speedup in sampling—that’s a massive claim, but the evaluation doesn’t fully back it up. The perplexity numbers are nice, but where’s the qualitative analysis? If you’re cutting sampling steps from 1000 to 10, how does that affect fluency and coherence in generated text? I was expecting some real-world comparisons—maybe human evaluations or even error cases where their method struggles.

---
Reference
---
[1] Chen, Binghui, Weihong Deng, and Junping Du. "Noisy softmax: Improving the generalization ability of dcnn via postponing the early softmax saturation." Proceedings of the IEEE conference on computer vision and pattern recognition. 2017.

[2] Gu, Jiuxiang, et al. "Exploring the frontiers of softmax: Provable optimization, applications in diffusion model, and beyond." arXiv preprint arXiv:2405.03251 (2024).

**Essential References Not Discussed:**

As far as I know,  there are several important related works that the paper does not cite including within the aspect of the Softmax Relaxation[1][2],  and sample accelerate[3], list as below.


---
Reference
---
[1] Jang, Eric, Shixiang Gu, and Ben Poole. "Categorical reparameterization with gumbel-softmax." arXiv preprint arXiv:1611.01144 (2016).

[2] Maddison, Chris J., Andriy Mnih, and Yee Whye Teh. "The concrete distribution: A continuous relaxation of discrete random variables." arXiv preprint arXiv:1611.00712 (2016).

[3] Luo, Simian, et al. "Latent consistency models: Synthesizing high-resolution images with few-step inference." arXiv preprint arXiv:2310.04378 (2023).

**Experimental Designs Or Analyses:**

I checked the experimental design and analyses, and while the results look promising, there are several issues with the evaluations including the bechmarks and the SOTA method selections:

(1) This paper evaluates on LM1B and OpenWebText (OWT) for training, and then tests zero-shot generalization on 7 datasets (PTB, Wikitext, etc.). This is reasonable for a language modeling paper, but the OpenWebText dataset is outdated and is not representative of modern large-scale LMs trained on more diverse internet-scale corpora. Comparing to stronger baselines like models trained on The Pile, C4, or real GPT training datasets would have been more informative.

(2) The baselines are weak. The autoregressive models they compare to (e.g., Transformer-XL, OmniNet) are not the best available. They should have compared against modern SOTA models like GPT-3.5, PaLM, or LLaMA. Beating old baselines doesn’t mean diffusion models are actually competitive.

**Methods And Evaluation Criteria:**

I think the methods are well-motivated but not entirely convincing in execution, and the evaluation criteria have some gaps that make it hard to fully trust the conclusions.

For the method part, the core idea of leveraging Gaussian diffusion for discrete diffusion models makes sense conceptually, and mapping it through argmax is an interesting perspective. But I’m not sure if it’s the best approach in practice. The claim that this transformation allows discrete models to directly benefit from continuous techniques (like curriculum learning and consistency distillation) is reasonable, but it still feels like a bit of a shortcut rather than a truly fundamental reformulation. I would have liked to see more analysis of when this approximation holds and where it might fail. Right now, it’s mostly taken at face value.

As for the evaluation, I also think the comparison baselines could be stronger. While they do compare against prior discrete diffusion models (SEDD, UDLM, MDLM), the autoregressive models they use for comparison aren’t the latest state-of-the-art. For example, modern transformer-based LMs like GPT-style models could set a stronger baseline, and without that, it's hard to say if diffusion is actually competitive for practical language modeling.

**Other Comments Or Suggestions:**

See the Weakness

**Other Strengths And Weaknesses:**

Strengths:

- I think this paper proposed a novel connection between discrete and continuous diffusion models, framing discrete diffusion as a transformed version of Gaussian diffusion via an argmax operation. It offers an interesting way to think about discrete generative modeling and provides a new lens for improving training and sampling efficiency.
- The experiment results shows that the model can achieves lower perplexity on several benchmarks, including cases where it outperforms autoregressive baselines which I think it with great practical usage consider the ill nature of low speed for diffusions.

Weakness:

- The paper claims that diffusion models can compete with autoregressive models, but the baselines used for comparison are outdated. More modern transformers, such as GLaM or GILL, should be included. Beside, I hope the author provide a  runtime comparison between diffusion models and autoregressive models. Even if diffusion models achieve lower perplexity, they may still be much slower at inference.
-  The training dataset is mostly OpenWebText, which is outdated. Stronger large-scale benchmarks like The Pile, C4, or GPT-style training datasets should be used for a more comprehensive evaluation.

**Questions For Authors:**

（1）For the method part, the author  claims that discrete diffusion naturally arises from a Gaussian diffusion process via an argmax transformation. However, the derivation only shows marginal distribution alignment and does not establish that the transition dynamics and Markov properties of the discrete process are fully preserved. Could you provide a formal proof or additional empirical validation that supports this claim beyond marginal alignment?

（2）As for the evaluation,  I think the Perplexity is promising. since the paper focuses on sampling efficiency while preserving sample quality, why is there no qualitative evaluation of generated text at different sampling steps? Perplexity is a useful metric but does not fully capture fluency, coherence, and grammaticality. Would you consider adding human evaluations or qualitative examples in an updated version?

I am glad to see the author could resolve my concerns, Thank you.

**Relation To Broader Scientific Literature:**

I think this paper makes meaningful contributions to the broader scientific literature on discrete diffusion models, efficient training strategies, and sampling acceleration techniques. It builds on and extends several key ideas from prior work while offering a novel theoretical perspective on the connection between discrete and continuous diffusion.

**Theoretical Claims:**

I checked the core theoretical claims in the paper, particularly the connection between discrete and continuous diffusion and the evidence lower bound (ELBO) comparisons.

One Issue I am a little conern about is that the curriculum learning trick involves annealing argmax to softmax over training steps, supposedly reducing variance and leading to faster convergence. They cite prior work on softmax approximations of discrete gradients, which makes sense.while the proof of variance reduction is missing. They argue that higher τ (temperature) leads to lower variance, but there’s no mathematical analysis quantifying how variance scales with τ. This is just assumed based on intuition from Gumbel-Softmax-like tricks. It would have been more convincing with a variance-bound derivation rather than just empirical loss curves.

---

> ### Author Rebuttal · Authors · 2025-04-01
>
> We want to thank the reviewer for their constructive and detailed feedback.
> # Concern 1: Transition kernel and the loss landscape
> We emphasize that while the $\arg \max$ operator maps Gaussian marginals to discrete marginals, it also preserves the transition dynamics and Markov property. In `anon. link [3]`, we show the marginal evolution follows: $\frac{d}{dt} q_t = -\frac{\mathcal{T}'(\tilde{\alpha}_t)}{K \mathcal{T}(\tilde{\alpha}_t)} \left[{\mathbf{1}} {\mathbf{1}}^\top - K \mathbf{I}\right] q_t$, where $\mathcal{T}'(\tilde{\alpha}_t)$ is the time derivative of  $\mathcal{T}(\tilde{\alpha}_t)$. This implies$\frac{d}{dt} q_t = Q_t q_t$, with  $Q_t = -\frac{\mathcal{T}'(\tilde{\alpha}_t)}{\mathcal{T}(\tilde{\alpha}_t)} \left[{\mathbf{1}} {\mathbf{1}}^\top - K \mathbf{I}\right] \in \mathbb{R}^{n \times n}$ representing the transition kernel for a **Markovian discrete diffusion process** (Anderson, 2012). Schiff et al., 2024 (Supp C, Eq. 50) confirm this is the transition dynamics of USDMs with diffusion parameter $\mathcal{T}(\tilde{\alpha}_t)$.
>
> **Loss Landscape**:  However, transforming a Gaussian diffusion into a discrete one **does not preserve the loss landscape**. In fact, the loss landscape for the discrete diffusion process is much more desirable because it induces a tighter bound on the likelihood as stated in Theorem 3.1.
>
> We stress that the equivalence betweeen Gaussian diffusion and the USDMs is **deep and fundamental without any approximations** whatsoever.
> # Concern 2: This method is an approximation
> As established above, the connection between Gaussian diffusion and USDMs is absolute and without any approximations. The softmax approximation to argmax introduced in Eqn(16), the training loss, leverages this fundamental relationship to design a low variance curriculum learning training scheme.
> # Concern 3: Two orders sampling speedup
> After distillation, DUO achieves a Gen. PPL of $79.24$ and entropy of $5.25$ with $T = 8$, closely matching the undistilled DUO with $T = 1024$, which has a Gen. PPL of $72.05$ and the same entropy ($5.22$). This reduction in steps comes without sacrificing entropy or degradation in sample quality (see `anon. link [3]`).
> # Concern 4: Novelty of Softmax annealing trick
> The core **novelty of this work is establishing a fundamental connection between discrete and Gaussian diffusion** via the argmax operator. We show that the softmax annealing trick, originally proposed for backpropagating through argmax [1, 2], can be repurposed to design a low-variance training curriculum for USDMs.
> # Concern 5: 2x faster training
> Variance reduction leads to better generalization, as shown by DUO achieving lower val. PPL with Curriculum Learning compared to without it. See our response to Concern 1 for Reviewer 1i8M.
> # Concern 6: Curriculum Learning
> Please refer to discussions in the response for concern 2 for the Reviewer 1i8M
> # Concern 7: Weak AR Baselines
> The comparison with the AR baseline **is fair**—DUO, diffusion baselines [5, 7, 8], and the AR Transformer in Tabs. 1–3 all use the same architecture and dataset, with the only difference being causal attention for AR and bidirectional for diffusion models. Notably, in Tab. 3, where **DUO outperforms AR on 3 out of 7 benchmarks, both use the same Transformer architecture**. Omninet and Transformer-XL results in Tab. 2 are included for reference only.
> # Concern 8:  Experimental Design
> We’d like to emphasize that our experimental setup is consistent with the current literature in the field of diffusion modeling [5, 7, 8].  Our experiments isolate the effect of the training method by fixing the Transformer architecture. While training with modern architectures like GPT-3.5, PaLM, or LLaMA on datasets like Pile or C4 would be valuable, it would require retraining all baselines—an infeasible task for an academic lab.
> # Concern 9: Sampling speed
> Sampling from a distilled **DUO model is significantly faster than an AR model**. See responses for concern 2 for Reviewer J6SU.
>
> # Other concerns:
> **Missing citations**: We’ll cite [4] and add a detailed discussion for [1, 2, 4] in the next version of the paper.
>
> **Qualitative Analysis**: We provide more samples in the `anon. link [3]` where we observe that the distilled DUO model outputs **significantly better quality samples** than the distilled MDLM model at lower sampling steps.
>
> ---
>
> ### References
> [1] Jang et al., 2016
>
> [2] Maddison et al., 2016
>
> [3] `Anon. link`: https://docs.google.com/document/d/e/2PACX-1vR0uKDuQHl4bBuC8KokEQhHNMvGdxbIskJm_SfXO_L6haSzWEqjPtL9wkVmg_yacNzMei2DAk21J5XX/pub
>
> [4] Anderson, Continuous-time Markov chains: An applications-oriented approach.
>
> [5] Schiff et al., 2025 “Simple Guidance Mechanisms for Discrete Diffusion Models”
>
> [6] Luo, Simian, et al. "Latent consistency models"
>
> [7]  Sahoo et al., 2024 “Simple and Effective Masked Diffusion Language Models”
>
> [8] Lou et al., 2024 “Discrete Diffusion Modeling by Estimating the Ratios of the Data Distribution”

---

> > ### Comment · Reviewer_oihg · 2025-04-03
> >
> > Thank you for the rebuttal, which resolves most of my concerns, therefore, I will keep my score. good luck!

---

> > > ### Author Response · Authors · 2025-04-03
> > >
> > > Thank you for your prompt and thoughtful engagement. We greatly appreciate your detailed feedback and will incorporate these discussions into the next revision of the paper. In particular, your comments on the transition kernels are especially valuable, as they are central to the paper's narrative. Addressing these points will significantly strengthen both the story and the overall clarity of the work.

---

### Official Review · Reviewer_J6SU · 2025-03-12

**Overall Recommendation:** 3

**Summary:**

This work presents a new training scheme for a uniform state discrete diffusion model based on the correspondence between Gaussian diffusion in continuous state and uniform state discrete diffusion. The authors state that while uniform state discrete diffusion and Gaussian diffusion are two separate Markov chains, the discrete process can be understood as a continuous process through the argmax operator. Based on this connection, the paper proposes a curriculum learning training that leads to faster convergence and low variance, and also a dual consistency distillation method. The experimental results show 2 times speed-up in training convergence and two orders of magnitude improvement in sampling speed.

**Claims And Evidence:**

Yes, the claims are supported by experimental results.

**Essential References Not Discussed:**

To the best of my knowledge, most of the relevant works were discussed in the paper.

**Experimental Designs Or Analyses:**

Yes, the experimental design seems valid.

**Methods And Evaluation Criteria:**

Yes, the evaluation criteria including LM1B and OWT datasets seem reasonable.

**Other Comments Or Suggestions:**

Please see the question below.

**Other Strengths And Weaknesses:**

**Strength**

- This work finds a new training algorithm effective for training uniform state discrete diffusion model from the connection between the discrete diffusion and Gaussian diffusion.

- The proposed training algorithm improves the performance of the uniform state discrete diffusion model that reduces the gap between the masked discrete diffusion model.

**Weakness**

- The performance of the uniform state discrete diffusion model even with the complex training/distillation is still underperforming the masked discrete diffusion model. As they both work on discrete space while the difference comes only from how the transition kernel is designed, the strength of the proposed method diminishes.

- The reason for using USDM is not clearly addressed in this paper (or I may have missed it). What is the advantage of using a uniform state instead of the masked discrete diffusion model? [Schiff et al., 2025] state that using a uniform state is advantageous for conditional generation but doesn't seem to apply for large language benchmarks such as LM1B. In this sense, this work could benefit from adding a controllable generation task where USDM would excel.

Schiff et al., Simple Guidance Mechanisms for Discrete Diffusion Models, ICLR 2025

**Questions For Authors:**

- Why is the reported performance of MDLM on the LM1B dataset different from the number from the original paper?

**Relation To Broader Scientific Literature:**

This work improves the performance of the uniform state discrete diffusion model with a new training scheme.

**Theoretical Claims:**

The claim on the connection between uniform state discrete diffusion and Gaussian diffusion has been explained.

---

> ### Author Rebuttal · Authors · 2025-04-01
>
> We want to thank the reviewer for their constructive feedback. We address the reviewers comments and questions below.
> # Concern 1: USDMs still lag MDMs
> As mentioned in the response to Concern 1, USDMs lag MDMs only when measured in terms of perplexity which isn’t necessarily the best metric for comparisions with MDMs. **USDMs are preferred over MDMs on few-step generation** (Fig. 3) where a distilled version of DUO significantly outperforms a distilled version of MDLM at $T=8$
>
> As mentioned in lines 16-27 (right), the design space of Discrete Diffusion models remains largely under explored as compared to Gaussian Diffusion models. In this work we establish a core property of USDMs — they emerge from Gaussian diffusion. This **opens up new avenues of future research which would leverage this connection** to improve USDMs by borrowing techniques from Gaussian diffusion— a connection that doesn’t exist for MDMs.
>
> # Concern 2 : Motivation for using USDMs over MDMs
> USDMs are preferable to MDMs for tasks like **guidance** (Schiff et al., 2025) and **fast sampling** (this work). The reason USDMs allow for faster sampling than MDMs is that they can fix their mistakes. This allows USDMs to make mistakes fast, but fix them later.
>
> We also note that while perplexity is a useful sanity check, it does not account for speed. Perplexity only captures how sample quality with a high number of sampling steps.
>
> In the table below, we present **new experiments** that show that USDMs can achieve strong sample quality quickly: Either with low latency, the time to produce a whole sequence, or high throughput, the rate of parallel generation.
>
> | **Model** | **Non embedding parameters** | **Latency $(\downarrow)$(BS=1)** | **Throughput $(\uparrow)$ with MAX BS (tok/sec)**  | **MAX BS** | **Gen. PPL  $(\downarrow)$ <entropy>** | **Gen. PPL  $(\downarrow)$ <entropy> (nucleus sampling p=0.9)** |
> | --- | --- | --- | --- | --- | --- | --- |
> | AR | 17M | 11.70 $\pm$ 1.15 | 926.80 $\pm$ 10.78 | 80 | 92.43 <5.61> | 32.13 <5.19> |
> | AR  | 110M | 14.70 $\pm$ 0.16 | 471.16 $\pm$ 3.54 | 32 | 35.93 <5.58> | **13.44** <5.26> |
> | Distilled DUO ($T=8$) | 110M | **0.21** $\pm$ 0.01 | **9938.00** $\pm$ 4.14 | 32 | **78.73** <5.24> | - |
>
> In this table, we train AR models on OWT for 1M steps with a context length of $1024$, varying the number of Transformer layers to control parameter count; exact architecture details can be found in `anon link [2]`. We do not use a KV cache for the AR baselines. We measure latency (sec) for generating batch size (BS) = 1 and find that DUO is significantly faster than even the smallest AR model. We also measure throughput (tokens/sec) using the largest BS (multiple of 16) that fits in memory. **The distilled DUO outperforms the 17M-parameter AR model in tems of speed**. All experiments were run on a single A100-SXM4-80GB. The mean and std are computed over 100 batches.
>
> # Concern 3: MDLM number different from original in Tab. 2
> Please refer to our response for the reviewer maKg, Concern 4.
>
> ---
>
> References
>
> [1]  Schiff et al., 2025 “Simple Guidance Mechanisms for Discrete Diffusion Models”
>
> [2] `Anon. link` : https://docs.google.com/document/d/e/2PACX-1vR0uKDuQHl4bBuC8KokEQhHNMvGdxbIskJm_SfXO_L6haSzWEqjPtL9wkVmg_yacNzMei2DAk21J5XX/pub

---

> > ### Comment · Reviewer_J6SU · 2025-04-02
> >
> > I appreciate the authors for the detailed response. Most of my concerns are addressed, and I raise my score to 3.

---

> > > ### Author Response · Authors · 2025-04-03
> > >
> > > Thank you for your prompt and thoughtful engagement. We sincerely appreciate your detailed feedback and will incorporate these insights into the next revision of the paper. In particular, your suggestion to elaborate on the motivation for USDMs over MDMs will meaningfully enhance both the narrative and the overall clarity of the work.

---

### Official Review · Reviewer_maKg · 2025-03-14

**Overall Recommendation:** 4

**Summary:**

This paper presents the first theoretical connection between discrete and continuous diffusion models, showing that discrete diffusion emerges from an underlying continuous Gaussian diffusion process. Building on this insight, the authors propose a curriculum learning strategy to improve training efficiency. Furthermore, they develop distillation techniques inspired by the relationship between discrete and continuous diffusion. Finally, extensive experiments are conducted to validate their theoretical findings.

**Claims And Evidence:**

The experimental results generally support the authors' claims.

**Essential References Not Discussed:**

What is the relationship between the proposed method and Bayesian flow networks [2, 3]?

[2] Graves et al. Bayesian flow networks. arXiv 2023.

[3] Xue et al. Unifying Bayesian Flow Networks and Diffusion Models through Stochastic Differential Equations. ICML2024.

**Experimental Designs Or Analyses:**

Yes, the authors’ experimental setup is reasonable.

**Methods And Evaluation Criteria:**

The evaluation metrics used in the paper are generally comprehensive and appropriate for the task.

However, generative perplexity (Gen PPL) has been criticized as an imperfect measure, as it can be artificially lowered by repeated words [1]. Despite this limitation, Gen PPL remains a widely used metric in this research area. Additionally, the authors mitigate this concern by including entropy as a supplementary evaluation metric.

[1] Zheng et al. Masked Diffusion Models are Secretly Time-Agnostic Masked Models and Exploit Inaccurate Categorical Sampling. ICLR2025.

**Other Comments Or Suggestions:**

In Lines 768–769, is there a missing part of the sentence? It seems incomplete.

**Other Strengths And Weaknesses:**

#### **Strengths**
1. The contributions of this paper are highly novel (as detailed in *Relation to Broader Scientific Literature*).
2. The paper is well-written, with clear logic and concise explanations.
3. Techniques such as curriculum learning are simple yet effective.

#### **Weaknesses**
1. In Section 4.2, the authors claim that deterministic trajectories can be obtained in the continuous space using DDIM and then mapped to the discrete space. However, this is confusing because the continuous and discrete processes follow different trajectories. Based on the loss function, Duo is trained on discrete trajectories (i.e., the neural network receives discretized inputs after the *argmax* operation). Given this, how can we obtain a score function in the continuous space that allows deterministic trajectory generation via DDIM? Am I missing something here?

2. The authors do not address a key question: While they establish a theoretical connection between continuous and discrete diffusion, which enables the transfer of techniques such as distillation (despite my earlier concerns about DDIM), the loss function used (i.e., Eq. (14)) is equivalent to the original discrete diffusion model’s loss function (i.e., Eq. (5)). If the two are mathematically equivalent, why does Eq. (14) yield better PPL results than Eq. (5)?

3. I have some doubts about the comparison with the UDLM baseline. First, Table 1 and Table 3 do not include results for UDLM. I understand that the original UDLM paper did not report results on the OWT dataset or zero-shot PPL, but the authors retrained SEDD Absorb and Plaid for these comparisons. Why did they not retrain UDLM under the same conditions? Second, in Table 2, the reported result for retrained UDLM (36.71) is significantly worse than the result reported in the original UDLM paper (31.28). The original UDLM result is actually better than the proposed method’s result (33.68). Why is there such a large discrepancy?

4. I also have concerns about the generative perplexity experiments. First, Figure 3 does not report entropy values. Second, in Table 4, Duo's entropy is significantly lower than that of MDLM, which raises concerns about whether the improved generative perplexity is simply due to Duo generating low-diversity sentences. According to Section 6.1 in [1], the normal entropy range is around 5.6–5.7, while Duo’s entropy is noticeably lower than this range.

**Questions For Authors:**

Please refer to the  Weaknesses.

**Relation To Broader Scientific Literature:**

Continuous diffusion models have achieved significant breakthroughs in image and video generation. More recently, discrete diffusion models have gained increasing attention. However, key questions—such as the theoretical connection between discrete and continuous diffusion and whether widely used distillation techniques from continuous diffusion can be applied to discrete diffusion—have remained open. This paper makes a novel contribution by theoretically establishing the link between continuous diffusion and discrete diffusion based on the uniform forward process. Additionally, it introduces distillation techniques to discrete diffusion, further advancing the field.

**Theoretical Claims:**

No.

---

> ### Author Rebuttal · Authors · 2025-04-01
>
> We want to thank the reviewer for their constructive feedback. We address the reviewers comments and questions below.
>
> # Concern 1 : Generating trajectories using DDIM.
> As noted in lines 250–252 (left), the deterministic Gaussian trajectories assume an optimal denoiser: given clean data $\mathbf{x}$ and a latent  $\mathbf{x}\_t \sim q_t(. | \mathbf{x})$*,* the optimal denoiser is defined as $\mathbf{x}^*_\theta(\mathbf{x}_t, t) = \mathbf{x}$  for all $t \in [0, 1]$. These DDIM trajectories (Eq. 17), under an optimal denoiser, preserve the Gaussian diffusion marginals (Song et al., 2022; Sec. C.1 in Zhou et al., 2025) and, when mapped to discrete space via $\arg \max$, align with the marginals of uniform state diffusion, as shown in Sec. 3.
>
> # Concern 2: Training loss vs Eval loss
> We'd like to clarify that while Eq. 14 and Eq. 6 are equivalent, **Eq. 16 which corresponds to a biased estimate of the true ELBO is used for training**. The $\arg \max$ in Eq. 14 is approximated by a tempered-softmax in Eq.16. This approximation reduces training loss variance and improves generalization. For a detailed explanation, please see our response to Concern 2 from Reviewer 1i8M.
>
> # Concern 3: Perplexity for UDLM in Tab 2 and Tab 3.
> The numbers for SEDD Absorb and Plaid are taken from Sahoo et al., 2024 and Lou et al., 2024. At the request of the reviewer, we conduct **new experiments** by training UDLM [3] on OWT as follows:
>
> |  | Val PPL (OWT) $(\downarrow)$ |
> | --- | --- |
> | UDLM | 27.43 |
> | DUO (Ours) | **25.20** |
>
> Zero shot perplexities:
>
> |  | Wikitext $(\downarrow)$ | AG News $(\downarrow)$ | LM1B $(\downarrow)$ | Lambada $(\downarrow)$ | PTB $(\downarrow)$ | Pubmed $(\downarrow)$ | Arxiv $(\downarrow)$ |
> | --- | --- | --- | --- | --- | --- | --- | --- |
> | UDLM | 39.42 | 80.96 | 77.59 | 53.57 | 112.82 | 50.98 | 44.08 |
> | DUO (Ours) | **33.57** | **67.81** | **73.86** | **49.78** | **89.35** | **44.48** | **40.39** |
>
> The conclusions remain unchanged—DUO is the state-of-the-art among USDMs.
>
> # Concern 4: Reported numbers for MDLM and UDLM in Tab. 2
> Lines 258–264 (right) clarify the **discrepancy caused by differences in dataset preprocessing**. The LM1B dataset contains short sentences (~30 tokens each with the GPT-2 tokenizer). In prior work [1, 3, 4], each datapoint in a batch is a single sentence padded to 128 tokens. This results in each sentence being considered in isolation.
>
> In contrast, we follow the sentence-packing scheme from Austin et al. (2021) [5], where sentences are concatenated before batching. This results in sentences potentially being split between batches, with additional and  potentially noisy context added. As a result, the packed data has a higher (worse) perplexity than padded data.
>
> At the reviewer’s request, we conduct **new experiments** where we trained our model using the preprocessing from [1, 3, 4], and report the results below. The conclusions remain unchanged—**DUO is state-of-the-art among USDMs** and approaches MDM performance.
>
> |  | Val PPL (LM1B) $(\downarrow)$ |
> | --- | --- |
> | UDLM | 31.28 |
> | DUO (Ours) | 29.95 |
> | MDLM | 27.04 |
>
> # Concern 5: Low sample diversity
> To make the curves in Fig. 3 more readable, the corresponding entropy values are provided separately in Tab. 4. As clarified in lines 351–358, the entropies of MDLM, SEDD Absorb, and SEDD Uniform align with that of an autoregressive model without nucleus sampling (approximately 5.6). In contrast, **DUO’s entropy matches that of an AR model with nucleus sampling (p = 0.9)**, around 5.2. Qualitative samples (`anon. link [6]`) further show that DUO produces significantly higher-quality outputs than both distilled and undistilled MDLM at lower sampling steps $T=8$.
>
> # Comments: Clarification on lines 768-769 in the appendix
> We meant to say the following:
> "We empirically verify the equivalence of the USDM ELBO with discrete latents (Eqn. 6) and Gaussian latents (Eqn. 14). To do this, we trained UDLM [3] on LM1B using the true ELBO from (6). We then evaluated the model using Gaussian latents (Eqn. 14), and
> recovered the same perplexity (36.71) as when using discrete latents. For each datapoint $\mathbf{x}$, we used $1000$ Monte Carlo samples for $t$ sampled using antithetic-sampling, with a linear schedule for $\tilde{\alpha}_t = 1 − t$."
>
> ---
> References
>
> [1]  Sahoo et al., 2024 “Simple and Effective Masked Diffusion Language Models”
>
> [2] Zhou et al., 2025 “Inductive Moment Matching”
>
> [3] Schiff et al., 2025 “Simple Guidance Mechanisms for Discrete Diffusion Models”
>
> [4] Lou et al., 2024 “Discrete Diffusion Modeling by Estimating the Ratios of the Data Distribution”
>
> [5]  Austin et al., 2021 “Structured denoising diffusion models in
> discrete state-spaces”
>
> [6] `Anon. link`: https://docs.google.com/document/d/e/2PACX-1vR0uKDuQHl4bBuC8KokEQhHNMvGdxbIskJm_SfXO_L6haSzWEqjPtL9wkVmg_yacNzMei2DAk21J5XX/pub

---

> > ### Comment · Reviewer_maKg · 2025-04-03
> >
> > I appreciate the author’s response and have raised my score from 2 to 4 accordingly.

---

> > > ### Author Response · Authors · 2025-04-03
> > >
> > > Thank you for your prompt and thoughtful engagement. We greatly appreciate your detailed feedback and will incorporate these discussions into the next revision of the paper. In particular, your suggestion to clarify the distinction between training loss and evaluation loss will help improve the overall clarity of the work.

---

### Official Review · Reviewer_1i8M · 2025-03-17

**Overall Recommendation:** 3

**Summary:**

This paper proposed a continuous parametrization of uniform-state discrete diffusion models. The key finding is a noise schedule (Eq. 11) that ensures the equivalence of the marginal distributions of the uniform-state discrete diffusion process and a Gaussian continuous diffusion process transformed by an argmax operator. The authors demonstrate two applications of this finding: to introduce a relaxed version of the training objective where a softmax operator is gradually annealed into the argmax operator, and to introduce a distillation method based upon the consistency loss of the underlying Gaussian diffusion. Their experiments show that the proposed method leads to improved perplexity in uniform-state diffusion models and faster sampling after distillation.

### After rebuttal ###
Thank the authors for their response. My concern on the gradient variance is resolved, but the others remain. I stand my initial rating.

**Claims And Evidence:**

The major claim is equivalence of the uniform-state discrete diffusion process and a Gaussian continuous diffusion process transformed by an argmax operator, for which the authors provide a formal proof.

On top of that, the authors further claim that a special curriculum that gradually anneal the soft to the hard max can be introduced to reduce the variance of in training, which is supported by the training curves with less perturbation than the baseline, as well as an improvement in the validation perplexity and the zero-shot perplexity.

The authors also claim that the distillation method built on top of the aforementioned equivalence is effective, which is supported by the curves of Gen perplexity and entropy in Fig3 and Fig8.

**Essential References Not Discussed:**

N/A

**Experimental Designs Or Analyses:**

The experiments and the associated analyses are quite standard so I don't feel any fundamental challenges. However, I am curious why the reduction of variance is only demonstrated with the training loss, instead of the gradients.

**Methods And Evaluation Criteria:**

The proposed method makes sense for improved training of uniform-state Diffusion models.
The evaluation criteria all commonly used in the field.

**Other Comments Or Suggestions:**

The name of the proposed method DUO appears abruptly in the experiment section.

**Other Strengths And Weaknesses:**

+ The writing is clear and very easy to follow.

- Although the equivalence between the uniform-state discrete diffusion process and a Gaussian continuous diffusion process transformed by an argmax operator is established formally, the training loss appears to be less rigorous. The network takes in the soft version but predict the hard version. This may render the training loss harder to interpret. Reflected in the empirical observation, the curves in Fig2 shows clear margin between the proposed method and MDLM, but the result in Table 2 reports the opposite.
\\

**Questions For Authors:**

Apart from my concerns mentioned above, I am also curious to know how is the distilled network parametrized. If the student model is similar to the teach in terms of the factorization of dimensions as articulated in Xu et al. 2024, why wasn't it an issue for the proposed method?

Xu et al. 2024, ENERGY-BASED DIFFUSION LANGUAGE MODELS FOR TEXT GENERATION

**Relation To Broader Scientific Literature:**

Uniform-state discrete diffusion model is an important family, for it naturally induce error correction with more sampling steps. However, it does not work as well as the masked diffusion. This paper is an attempt to improve the training.

**Theoretical Claims:**

I have checked the proofs and they look sensible to me.

---

> ### Author Rebuttal · Authors · 2025-04-01
>
> We want to thank the reviewer for their constructive feedback. We address the reviewers comments and questions below.
>
> # Concern 1: Interpreting training loss (Fig 2) vs Tab 2
>
> We will clarify Fig 2 by highlighting the key takeaway: DUO’s training loss exhibits significantly lower variance than both MDLM and UDLM’s. We thank the reviewer for pointing out that the biased training loss of DUO distracts from this key takeaway, and will revise this figure to emphasize the empirical variance of the loss, as well as gradient variance (more details in Concern 2).
>
> We note that in the current Fig. 2 (paper), DUO’s curve lies below MDLM’s because its training loss (Eq. 16) is a biased approximation of the true ELBO (Eqs. 6 and 14). This bias causes the training loss to be lower. More formally, for the true ELBO (Eq. 6), the discrete diffusion parameter $\tilde{\alpha}_t$ must lie in $[0, 1]$. However, during training with the approximation (Eq. 16), it suffices to choose the Gaussian diffusion parameter $\tilde{\alpha}_t \in [0.85, 0.97]$, which maps to  $\alpha_t = \mathcal{T}(\tilde{\alpha}_t) \in [\approx 0, \approx 1]$ for $\tau \to 0^+$ (see Fig. 4, appendix). Since, **Gaussian latents contain more signal than noise**, while training with Eq. 16 with $\tau > 0$ the denoising model finds reconstructing the input easier, **resulting in a lower training loss**.
>
> Tab. 2 (paper) reports the validation perplexity computed using the true ELBO (Eq. 6 for DUO). For clarity, we report the validation perplexities of various methods below at different training steps and observe that DUO consistently outperforms UDLM and approaches the performance of MDLM. We thank the review for pointing out the potential conflicting interpretations of Fig 2 and Tab. 2, and believe revising Fig 2 to focus on variance will alleviate this.
>
> |  | 10K | 100K | 250K | 500K | 750K | 1M |
> | --- | --- | --- | --- | --- | --- | --- |
> | MDLM | 65.75 | 40.01 | 36.37 | 33.41 | 32.66 | 32.03 |
> | UDLM | 91.95 | 74.71 | 43.11 | 39.70 | 37.63 | 36.71 |
> | DUO (Ours) | 69.34 | 57.11 | 37.44 | 35.20 | 34.22 | 33.68 |
>
> **Note:** UDLM is equivalent to DUO w/o Curriculum Learning
>
> # Concern 2: Gradient Variance
>
> As requested by the reviewer, we report gradient variance across model weights. We compute the variance of each weight’s updates over 10 steps on LM1B at different training steps. **With Curriculum Learning (CL), gradient variance is significantly lower**—especially in early training (RTab 1; below). Among the 100 weights with the highest variance (RTab 2; below), **CL reduces variance by an order of magnitude** early on, which gradually diminishes over time. This reduction appears beneficial, as it accelerates learning in the initial phases, reflected in improved validation loss; see above.
>
> **(RTab 1) Sum of all variances:**
>
> |  | 10k | 20k | 50k | 100k | 500k |
> | --- | --- | --- | --- | --- | --- |
> | CL | **2815.36** | **2471.65** | **1890.76** | **1469.85** | **947.98** |
> | w/o CL | 10852.9 | 7811.04 | 6315.7 | 5454.7 | 1678.47 |
>
> **(RTab 2) Sum of highest 100 variances:**
>
> |  | 10k | 20k | 50k | 100k | 500k |
> | --- | --- | --- | --- | --- | --- |
> | CL | **0.30** | **0.85** | **1.21** | **0.86** | **1.15** |
> | w/o CL | 11.7 | 20.09 | 34.2 | 55.1 | 1.92 |
>
> We observe a similar trend in the variances of the loss as well; see below.
>
> **(RTab 2) Sum of highest 100 variances:**
>
> |  | 10k | 20k | 50k | 100k | 500k |
> | --- | --- | --- | --- | --- | --- |
> | CL | **7.09** | **6.29** | **5.33** | **4.97** | **4.76** |
> | w/o CL | 9.19 | 7.72 | 6.85 | 6.32 | 5.47 |
>
> We will replace Fig 2 (the loss curves) with these findings.
>
> # Other comments
>
> During distillation, the student and teacher networks share the same architecture and differ only in weights—the teacher uses EMA weights, while the student uses the current model parameters. Both networks factorize the joint distribution independently as $p_\theta(\mathbf{x}\_0 | \mathbf{z}\_t)  = \prod_i p_\theta(\mathbf{x}^i_0 | \mathbf{z}\_t)$. Although Xu et al. (2024) propose a more expressive factorization with an energy term,$p_\theta(\mathbf{x}\_0 | \mathbf{z}\_t)  = \prod_i p_\theta(\mathbf{x}^i_0 | \mathbf{z}\_t) e^{E_\phi} / Z_\phi$, which leads to a better fit to the data, the independent assumption performs well in practice [2, 3, 4]. Modeling the energy function $E_\phi$ and the partition function $Z_\phi$ is also non-trivial, often requiring an additional network and complicating both training and sampling. We believe incorporating this approach could enhance our model’s expressiveness and benefit both DUO and MDLM, which we leave for future work.
>
> ---
>
> References
>
> [1] Xu et al., 2024, ENERGY-BASED DIFFUSION LANGUAGE MODELS FOR TEXT GENERATION
>
> [2] Sahoo et al., 2024 “Simple and Effective Masked Diffusion Language Models”
>
> [2] Schiff et al., 2025 “Simple Guidance Mechanisms for Discrete Diffusion Models”
>
> [3] Lou et al., 2024 “Discrete Diffusion Modeling by Estimating the Ratios of the Data Distribution”

---

### Decision · Program_Chairs · 2025-05-01

**Decision:**

Accept (poster)

**Comment:**

The manuscript establishes a connection between uniform-state discrete diffusion and Gaussian diffusion and uses the property to improve discrete diffusion. The reviewers found the theoretical contributions novel, the evaluation setup reasonable, and the empirical evaluation supports the claim about improving the performance of USDM. The reviewers requested several clarifications, particularly regarding the loss (including interpreting the training loss and the distinction between training and evaluation loss) and gradient variance.  The relationship between USDM and MDM also raised some concerns, particularly regarding the remaining performance gap between USDM and MDM despite the proposed improvements. The authors addressed most of the reviewers’ concerns during the rebuttal period, and since all reviewers recommended acceptance, I also recommend accepting the paper for ICML. However, given the extension discussion and the number of clarifications, it is crucial that the authors incorporate the promised changes into the final version of the paper.